# Polycomb deficiency drives a FOXP2-high aggressive state targetable by epigenetic inhibitors

Fan Chen[1,2], Aria L. Byrd[1], Jinpeng Liu[3], Robert M. Flight [4,5], Tanner J. DuCote[1], Kassandra J. Naughton[1], Xiulong Song[1], Abigail R. Edgin [1], Alexsandr Lukyanchuk[1], Danielle T. Dixon[1], Christian M. Gosser[1], Dave-Preston Esoe[1], Rani D. Jayswal[6], Stuart H. Orkin [7], Hunter N. B. Moseley[4,5], Chi Wang[3,5] & Christine Fillmore Brainson [1,5] ✉

Inhibitors of the Polycomb Repressive Complex 2 (PRC2) histone methyltransferase EZH2 are approved for certain cancers, but realizing their wider utility relies upon understanding PRC2 biology in each cancer system. Using a genetic model to delete *Ezh2* in KRAS-driven lung adenocarcinomas, we observed that *Ezh2* haplo-insufficient tumors were less lethal and lower grade than *Ezh2* fully-insufficient tumors, which were poorly differentiated and metastatic. Using three-dimensional cultures and in vivo experiments, we determined that EZH2-deficient tumors were vulnerable to H3K27 demethylase or BET inhibitors. PRC2 loss/inhibition led to de-repression of FOXP2, a transcription factor that promotes migration and stemness, and *FOXP2* could be suppressed by BET inhibition. Poorly differentiated human lung cancers were enriched for an H3K27me3-low state, representing a subtype that may benefit from BET inhibition as a single therapy or combined with additional EZH2 inhibition. These data highlight diverse roles of PRC2 in KRAS-driven lung adenocarcinomas, and demonstrate the utility of three-dimensional cultures for exploring epigenetic drug sensitivities for cancer.

Enhancer of zeste homolog 2 (EZH2), the enzymatic component of the Polycomb Repressive Complex 2 (PRC2), is best known for trimethylation of histone H3 at lysine 27 (designated as H3K27me3) to silence chromatin regions[1]. PRC2 plays key roles in embryonic development and maintenance of cell identities, and may act as an oncogene or tumor suppressor in different contexts[2]. Fitting with an oncogenic role, high EZH2 expression in the lung is oncogenic, drives sensitivity to pharmacological inhibition of EZH2[3] and correlates with worse lung cancer prognosis[4,5]. However, decoupling of

EZH2 from H3K27me3 levels has been widely observed[6–9], suggesting that high EZH2 does not necessarily translate to high levels of PRC2 catalytic function. In line with PRC2 function being anti-tumor, deletion of the PRC2 component *Eed* enhanced malignancy in KRAS+/*Trp53*-null lung cancers, drove aggressive mucinous tumors, and changed the tumor immune microenvironment[10,11]. Likewise, deletion of *Ezh2* in the context of oncogenic KRAS and intact *Trp53* drove aggressive lung cancers and exacerbated immune cell infiltration[12]. These data indicate that PRC2 plays

[1]Department of Toxicology and Cancer Biology, University of Kentucky, Lexington, KY 40536, USA. [2]Department of Medical Oncology, Sun Yat-sen University Cancer Center, State Key Laboratory of Oncology in South China, Collaborative Innovation Center for Cancer Medicine, Sun Yat-sen University, 510060 Guangzhou, P. R. China. [3]Department of Internal Medicine, University of Kentucky, Lexington, KY 40536, USA. [4]Department of Molecular & Cellular Biochemistry, University of Kentucky, Lexington, KY 40536, USA. [5]Markey Cancer Center, University of Kentucky, Lexington, KY 40536, USA. [6]Markey Cancer Center Biostatistics and Bioinformatics Shared Resource Facility, Lexington, KY 40536, USA. [7]Department of Hematology/Oncology, Boston Children's Hospital, Boston, MA 02115, USA. ✉e-mail: cfbrainson@uky.edu

divergent roles in different cancer contexts, each requiring comprehensive characterization.

Recently, tazemetostat (EPZ6438), an inhibitor of the catalytic activity of EZH2, was approved for the treatment of advanced epithelioid sarcoma and follicular lymphoma[13], and numerous clinical and preclinical trials are focused on combining EZH2 inhibitors with other drugs. One consequence of EZH2 inhibition is a dramatic shift in the immune cell microenvironment, and this is one focus of current drug combinations[11,14,15]. Other efforts focus on determining epigenetic vulnerabilities associated with low PRC2 activity. For example, PRC2-deficient malignant peripheral nerve sheath tumors are sensitive to inhibition of H3K27ac-reader bromodomain and extra-terminal motif (BET) proteins when combined with MAPK/ERK kinase (MEK) inhibition[16]. Similarly, H3K27M mutations in diffuse intrinsic pontine glioma predispose to sensitivity to both EZH2 inhibition and BET inhibition through the drug JQ1[17–19]. Lastly, GSK-J4, which inhibits histone demethylases and stabilizes H3K27me3, inhibits chemo-resistant lung cancer cells and mammosphere-derived breast cancer stem cells, both of which harbor decreased H3K27me3[20–22]. Although these studies suggest possible synergy of PRC2 loss with other epigenetic drugs, this idea has not been tested in lung cancer.

In addition to gene expression control through epigenetic mechanisms, transcription factors are master regulators of important cellular phenotypes, including stemness, proliferation, and migration[23,24]. Forkhead box protein P2 (FOXP2) is a member of the FOX transcription factor family that share conserved forkhead or "winged-helix" DNA-binding domains[25]. In addition to its role in the development of speech, FOXP2 is also known to be expressed in developing embryonic lung and can be used to mark induced pluripotent stem cell-derived alveolar lung progenitors[26,27]. Knockout of *Foxp2* led to defective postnatal lung alveolarization in murine models[28]. In the adult, FOXP2 is expressed in the distal lung epithelium and inhibits lung cell differentiation genes[29,30]. Despite clues suggesting the importance of FOXP2 in maintaining stemness in lung epithelium, the roles it plays in lung adenocarcinoma have yet to be explored.

## Results

### *Ezh2* haplo- and full Insufficiency drive distinct phenotypes in KRAS+/*Trp53*-null lung cancer

We generated the LSL:*Kras*^G12D/+; *Trp53*^flox/flox (LSL: lox-stop-lox) genetically engineered mouse model with either zero, one, or two floxed alleles of *Ezh2* and induced autochthonous lung tumors by intranasal adeno-Cre (AdCre) virus administration (Fig. 1a). We analyzed the resulting KRAS+/*Trp53*-null lung tumors at a variety of timepoints. Mice with one floxed allele of *Ezh2* (*Ezh2* heterozygous; Δ/+) had significantly lower tumor burden when compared with *Ezh2* wild-type (WT) mice (Fig. 1b, c and Supplementary Fig. 1a). In contrast, mice with two floxed alleles of *Ezh2* (*Ezh2* null, Δ/Δ) had significantly higher tumor burden, significantly shorter survival, and significantly higher nuclear grades when compared to mice with *Ezh2* WT or heterozygous tumors (Fig. 1d, e). Strikingly, *Ezh2* null tumor cells had a significantly higher propensity to metastasize to areas including lymph nodes, pleural space, chest wall, diaphragm, and liver than *Ezh2* WT or heterozygous tumor cells (Fig. 1f). Immunohistochemical (IHC) staining showed that *Ezh2* null tumor cells had significantly lower EZH2 expression than *Ezh2* WT, and that *Ezh2* heterozygous cells had intermediate levels of EZH2 expression (Fig. 1g and Supplementary Fig. 1b, c). Confirming that PRC2 activity in *Ezh2* null tumors was deficient, H3K27me3 staining was significantly lower in *Ezh2* null tumors than in the other two genotypes (Supplementary Fig. 1d). Unlike *Eed* null tumors, *Ezh2* null tumors had rare areas of alcian blue (mucin) stain, but did sometimes show more collagen staining, consistent with an epithelial-to-mesenchymal (EMT) phenotype (Supplementary Fig. 1e).

To further understand the mechanisms at play in the immune microenvironment, we performed RNA-sequencing (RNA-seq) on bulk tumor tissue containing immune cells. We first utilized TIMER2.0[31,32] and specifically used the CIBERSORT algorithm[33] to estimate the immune cell populations. *Ezh2* heterozygous tumors contained significantly more tumor-infiltrating CD8+ T cells than the other two *Ezh2* genotypes, and significantly more monocytes than *Ezh2* WT tumors (Supplementary Fig. 1f). Gene set enrichment analysis (GSEA)[34] demonstrated that *Ezh2* heterozygous tumors were significantly less enriched for cold tumor gene signatures, but had enrichment for lymphocyte signatures, indicating an anti-tumor immune microenvironment (Supplementary Fig. 1g and Supplementary Table 1). We next used HALO® artificial intelligence software to analyze the immune cell infiltration (Supplementary Fig. 1h). Analysis showed that *Ezh2* heterozygous tumors had significantly more tumor-associated lymphocytes/monocytes when compared with *Ezh2* null tumors (Supplementary Fig. 1i). We confirmed by IHC that *Ezh2* heterozygous tumors had significantly higher infiltration of CD3+ T cells when compared with the other two genotypes (Supplementary Fig. 1j, k). Collectively, these data suggest that full and partial insufficiency for *Ezh2* drive distinct phenotypic outcomes in KRAS+/*Trp53*-null lung adenocarcinomas, with full insufficiency driving more aggressive and metastatic tumors, while partial deficiency generates an anti-tumor immune microenvironment.

### Model systems differ in cell cycle and gene expression

Next, we aimed to understand the epigenetic and transcriptional differences that underlie the in vivo tumor phenotypes, but prior to doing this, we characterized four model systems that can be used for this purpose: (1) tumor-derived 2D cell lines that are easy to grow and manipulate; (2) 3D tumoroids in Matrigel transwells that can be expanded, but less easily than 2D cells; (3) freshly sorted tumor cells that represent tumor cells in vivo, but are difficult to obtain in large numbers (Supplementary Fig. 2a); and (4) total (bulk) tumors that have the caveat of containing numerous stromal cell types. Consistent with previous reports[10], 2D *Ezh2* null cultures exhibited more elongated mesenchymal-like cells than 2D *Ezh2* WT cultures, whereas in cross sections of 3D tumoroids and in vivo tumors, the mesenchymal phenotype was less obvious (Fig. 2a). To confirm a functional EMT in 2D *Ezh2* null cells, we performed transwell migration assays, and observed significantly more migration of *Ezh2* null cells than *Ezh2* WT cells (Supplementary Fig. 2b). As expected, we observed that in both 2D and 3D states, *Ezh2* null cells expressed lower levels of *Ezh2* than the other two genotypes (Supplementary Fig. 2c). To confirm excision of the *Ezh2* gene, we used endpoint PCR of the *Ezh2* genomic DNA to verify the WT, floxed intact and floxed excised *Ezh2* alleles in the tumoroids with the corresponding genotypes (Supplementary Fig. 2d).

To understand the transcriptional landscapes in each model systems, we compared the four models (independent of *Ezh2* genotype) by RNA-sequencing. Unsupervised hierarchical clustering of samples using distances derived from weighted correlations showed that 2D cell lines form their own cluster that is distinct from all other samples, while in vivo samples clustered closely together and tumoroids were clustered between in vivo and 2D samples, regardless of *Ezh2* phenotype (Fig. 2b). Principal component (PC) analysis performed on all samples determined that PC3 showed a stratification of the 2D samples away from the other three groups (Supplementary Fig. 2e). This PC is enriched for genes associated with epithelial cell proliferation, lung development, wound healing, cell–cell interactions, and immune response (Supplementary Table 2). Similarly, GSEA showed that when compared with 2D cultures, tumoroids and sorted tumor cells had enrichment for transcriptional signatures related to immunity and lung lineage determination, while 2D cells were enriched for cell cycle gene signatures (Fig. 2c and Supplementary Table 3).

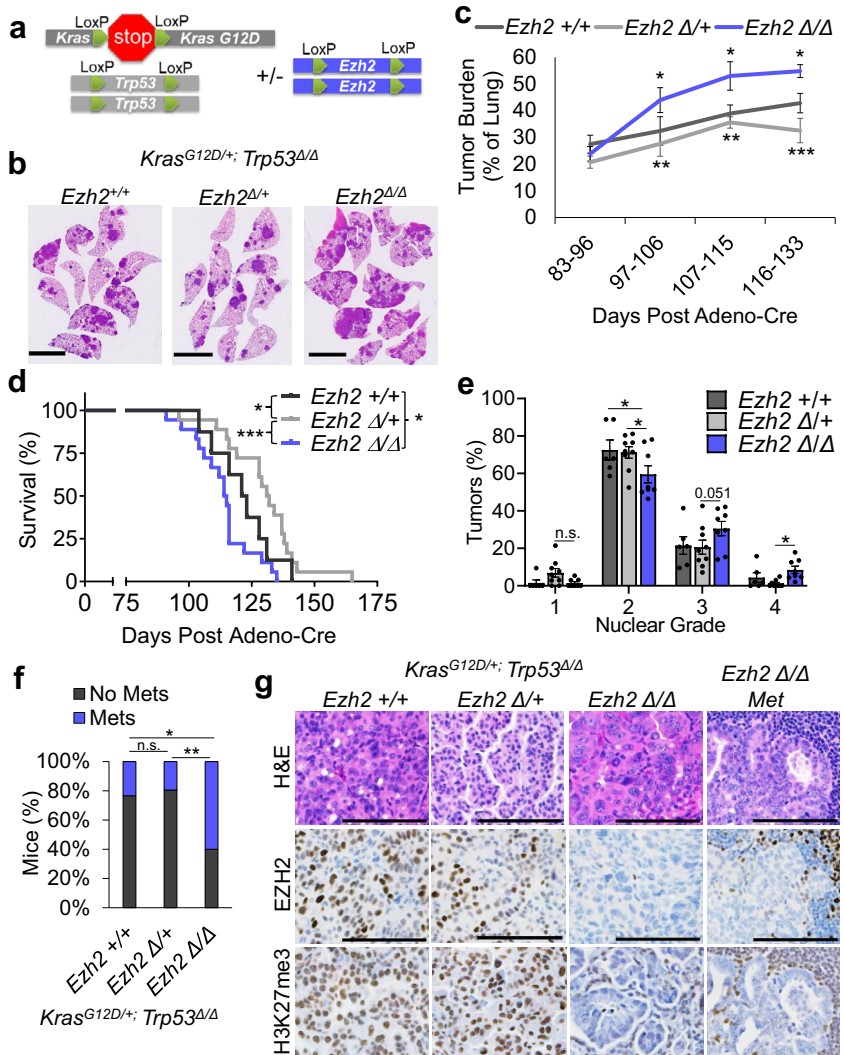

**Fig. 1 | *Ezh2* haplo- and full insufficiency drive distinct phenotypes in KRAS +/*Trp53*-null lung cancer. a** Schematic for LoxP-mediated deletion of *Ezh2* to generate EZH2 WT, heterozygous, and homozygous null KRAS/*Trp53-null* GEMMs. **b** Representative H&E-stained cross sections from lungs of closely related mice that were induced and sacrificed at the same timepoint, scale bar = 5 mm. **c** Tumor burden relative to the whole lung at the different timepoints post Adeno-Cre graphed as mean values +/− SEM. * indicates *P* = 0.0461 *Ezh2* WT vs. null Bin 2, *P* = 0.028 Bin 3, *P* = 0.0437 Bin 4, ** indicates *P* = 0.005, *** indicates *P* = 0.0001 *Ezh2* heterozygous vs. null by one-way ANOVA with multiple comparisons. Bin 1 *n* = 4, 9, 6, Bin 2 *n* = 10, 10, 9, Bin 3 *n* = 11, 14, 6, Bin 4 *n* = 9, 11, 9 for *Ezh2* WT, heterozygous, null individual mice, respectively. **d** Percentage survival of mice of indicated genotypes. * indicates *P* = 0.0376 *Ezh2* WT vs. *Ezh2* heterozygous, *** indicates *P* = 0.0003 *Ezh2* heterozygous vs. *Ezh2* null, and * indicates *P* = 0.0397 *Ezh2* WT vs.

*Ezh2* null by log-rank test. *n* = 8, 18, 18 for *Ezh2* WT, heterozygous, null individual mice, respectively. **e** Nuclear grade was scored on sectioned tissue by a blinded pathologist and graphed as mean values +/− SEM. * indicates *P* < 0.042 by one-way ANOVA with multiple comparisons, *n* = 6, 9, 8 for *Ezh2* WT, heterozygous, null individual mice, respectively. **f** Mice of indicated genotypes were scored as having metastasis (Met) or not by gross examination and histology at days 91–133 post tumor induction and percentage of each are graphed. * indicates *P* = 0.012 and ** indicates *P* = 0.0024 by Fisher's exact test. *n* = 30, 36, 25 for *Ezh2* WT, heterozygous, null individual mice, respectively. **g** Representative images of tumor sections stained with H&E, EZH2 or H3K27me3, scale bar = 100 μm. *n* = 26 or greater individual mouse tumor sections were stained and data are summarized in Supplementary Fig. 1a, b, d. Genotypes are signified as: WT, +/+; heterozygous, Δ/+; null, Δ/Δ. Source Data are provided for this figure.

To functionally verify differences in cell cycle, we next compared the cell cycle profiles of 2D cell lines and 3D tumoroids and found that they were dramatically different. 2D cultures showed much higher percentages of cells in S-phase and much lower percentages of cells in G0/G1 than 3D tumoroids for every *Ezh2* genotype (Fig. 2d). Intriguingly, *Ezh2* heterozygous tumoroids had increased G0/G1, while *Ezh2* null tumoroids had more G2/M when compared to both *Ezh2* WT and *Ezh2* heterozygous, and less G0/G1 compared with mono-allelic deletion. These results may partially explain the opposing phenotypes driven by *Ezh2* heterozygous and homozygous deletion in vivo. Consistent with the cell cycle results, 2D cultures had significantly higher levels of proliferating cell nuclear antigen (PCNA) and phospho-H3 than tumoroids (Fig. 2e, f). Intriguingly, *Ezh2* null tumoroids expressed

significantly higher levels of PCNA than *Ezh2* WT and heterozygous tumoroids, while all 2D cells expressed comparable levels of PCNA. These data demonstrate critical differences in transcriptional states and growth patterns among culture systems, which should be considered when designing and testing novel therapeutic strategies.

### *Ezh2* deficiency leads to decreased PRC2-chromatin occupancy and diverse gene expression in different models

To further study the molecular consequences of *Ezh2* deletion in vitro and in vivo, we used RNA-sequencing to investigate gene expression differences between the *Ezh2* genotypes in each model system (Supplementary Fig. 2f). GSEA comparing *Ezh2* null to *Ezh2* WT cells, or *Ezh2* null to *Ezh2* heterozygous cells demonstrates the level of complexity of

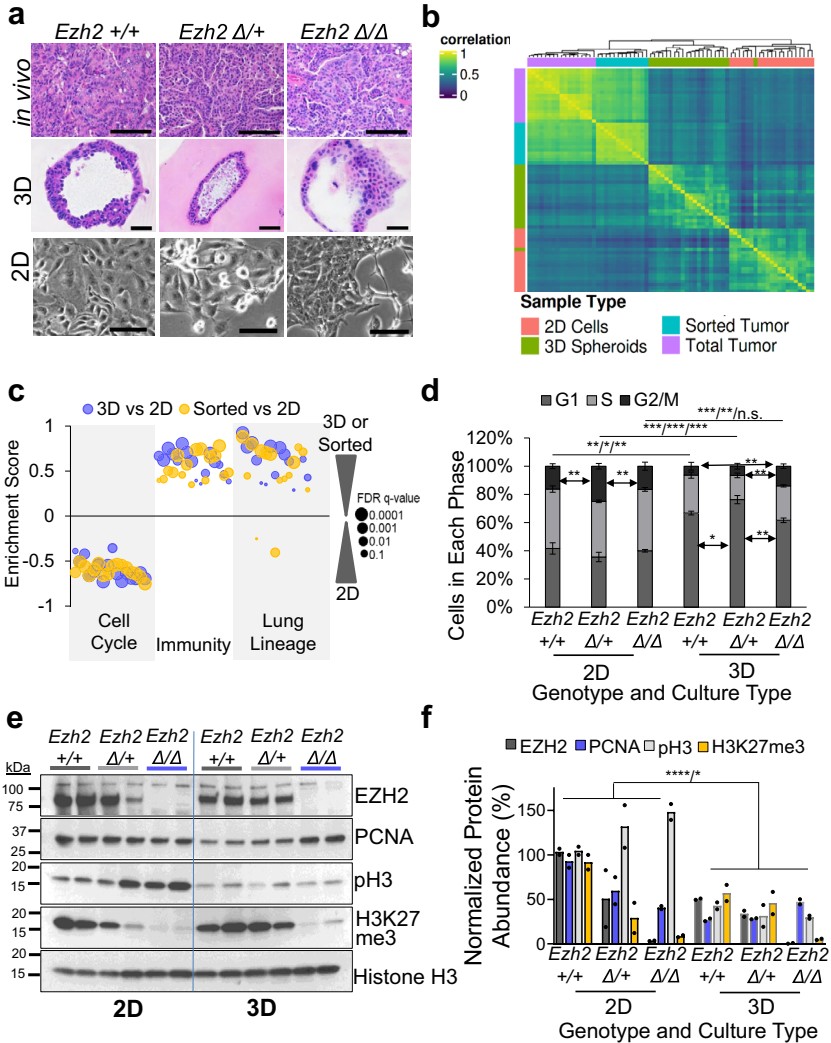

**Fig. 2 | Model systems differ in cell cycle and gene expression. a** Representative images of in vivo tumor histology, in vitro 3D tumor spheroid histology and 2D cell cultures of the indicated *Ezh2* genotypes, scale bar = 100 μm. *n* = 3 or greater individual mouse tumor-derived cultures were examined for each genotype and culture system. **b** Heatmap of sample-to-sample correlations using highest loading genes from PC3 analysis, where the Pearson correlations are weighted by sample-sample information and sample-sample consistency (see "Methods"). The 1 − correlation value was then used as a pseudo distance for hierarchical clustering and ordered using default argument of dendsort. *n* = 17, 13, 20, 21 for total tumor, sorted tumor, 2D cell cultures, 3D tumoroid individual mouse tumor-derived samples, respectively. **c** Rank-ordered gene lists were queried against the MSigDB databases and enrichment scores of selected gene signatures for 2D, 3D, or in vivo sorted tumors were plotted. Dot size reflect FDR estimates. Data are shown in Supplementary Table 3. **d** Percentages of cells in each cell cycle phase were measured by 7AAD cell

cycle flow cytometry in 2D cells and 3D tumoroids of the indicated *Ezh2* genotypes plotted as mean values +/− SEM. * indicates *P* < 0.035, ** indicates *P* < 0.009, *** indicates *P* < 0.0009 by one-way ANOVA with multiple comparisons and Holm–Šídák's post hoc test for comparing genotypes within culture system, and by two-tailed *t* test for comparing the same genotype in the two culture systems. 2D: *n* = 4, 3, 3; 3D: *n* = 3, 4, 3 for *Ezh2* WT, heterozygous, null individual mouse tumor-derived cultures, respectively. **e** 2D cells and 3D tumoroids were examined for expression of the indicated proteins by immunoblot. **f** Normalized protein abundance of the indicated proteins in 2D cells and 3D cultures by immunoblot. **** indicates *P* = 0.0000086 for pH3 in 2D vs. 3D and *P* = 0.024 for PCNA in 2D vs. 3D for all *Ezh2* genotypes combined by two-tailed *t* test. *n* = 2 individual mouse tumor-derived cultures per genotype/culture system. Genotypes are signified as: WT, +/+; heterozygous, Δ/+; null, Δ/Δ. Source Data are provided for this figure.

PRC2-mediated gene regulation in the different systems (Fig. 3a and Supplementary Table 4). When comparing *Ezh2* null to *Ezh2* WT cells, Polycomb targets genes, *HOX* genes, EMT, and immunity-related signatures were all enriched in *Ezh2* null 2D cultures. In contrast, many of these same signatures were enriched in *Ezh2* WT (depleted in *Ezh2* null) in both sorted cells and tumoroids. These results suggest that compared to 2D cells, the tumoroids more faithfully recapitulate gene expression patterns of in vivo samples. Interestingly, Polycomb targets genes were more uniformly enriched (expressed) in both the *Ezh2* null and *Ezh2* WT cells when compared to *Ezh2* heterozygous cells, including in the sorted cells and tumoroids (Supplementary Fig. 3a). This result suggests that *Ezh2* heterozygous tumors may have stabilization of canonical PRC2 activity rather than a partial loss, and this is

consistent with the observation that genes involved in histone methylation, including Polycomb complex genes, are more highly enriched in *Ezh2* heterozygous cells when compared with the other two genotypes. Lastly, relative to *Ezh2* null tumoroids, *Ezh2* heterozygous tumoroids had enrichment of immunity gene signatures, predominately involving type-I interferon response pathways, suggesting that *Ezh2* heterozygous cells may have hyperactivation of the damage-associated molecular pattern (DAMPs) and STING signaling[35].

We next queried significantly differentially expressed genes (DEGs) between the *Ezh2* null and *Ezh2* WT cells in the different systems (Fig. 3b). Consistent with the GSEA, *Hox* genes were upregulated in *Ezh2* null 2D cells, but *Hox* genes were not expressed in tumoroids and sorted tumors. In tumoroids, genes relating to lung cell

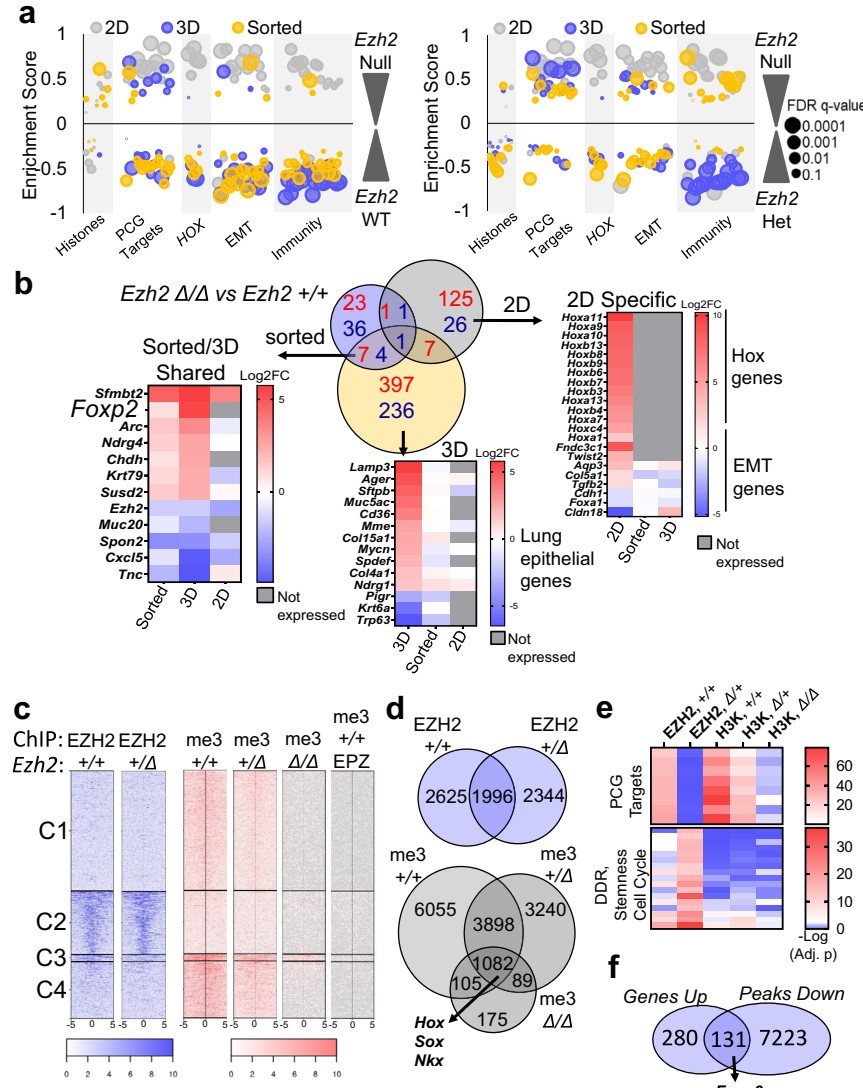

**Fig. 3 | *Ezh2* deficiency leads to diverse gene expression in different models. a** Rank-ordered gene lists were queried against the MSigDB databases and enrichment scores of selected gene signatures enriched in the *Ezh2* null or *Ezh2* heterozygous compared to *Ezh2* WT for the indicated models were plotted. Dots size estimates FDR. Data are summarized in Supplementary Table 4. **b** Venn diagram and heatmaps depict significant differentially expressed genes in *Ezh2* null vs. WT cells for the indicated culture systems, with log-fold change LFC > 0.5 or LFC < −0.5, FDR < 0.05. Genes higher in *Ezh2* null are in red, genes lower in *Ezh2* null are in blue. Heatmaps depict LFC of selected genes identified by the indicated contrasts. **c** Heatmap representation of EZH2 and H3K27me3-bound chromatin peaks centered across a ± 5-kb window that showed decreased occupancy of *Ezh2*

heterozygous, *Ezh2* null, and EPZ6438-treated *Ezh2* WT mouse tumor spheroids compared to *Ezh2* WT. **d** Venn diagram showing EZH2 and H3K27me3 peaks significantly called by SICER at FDR < E-10 for genes in the indicated *Ezh2* genotypes or EZH2 inhibitor-treated organoids. **e** GREAT analysis heatmaps depicting the H3K27me3 peaks of mouse 3D organoid ChIP-seq samples enriched in the indicated gene sets. Data are summarized in Supplementary Table 6. **f** Venn diagram showing overlap of shared DEGs from (**b**) with H3K27me3 ChIP peaks found in *Ezh2* WT tumoroids that were decreased by more than twofold or absent in *Ezh2* null tumoroids, with *Foxp2* indicated as a shared gene. Genotypes are signified as: WT, +/+; heterozygous, Δ/+; null, Δ/Δ. Source Data are provided for this figure.

differentiation such as *Spdef*, *MucSac*, *Lamp3*, and *Trp63* were differentially regulated in *Ezh2* null cells, whereas many of these same genes were too lowly expressed to assess in 2D cell lines. We closely examined the seven genes that were significantly upregulated in both tumoroids and sorted tumor *Ezh2* null cells. Of these genes, Forkhead Box Protein 2 (*Foxp2*) encodes for a transcription factor vital in embryonic lung development and was significantly higher in *Ezh2* null tumoroids and in vivo sorted tumors than in *Ezh2* WT counterparts.

In order to understand the chromatin occupancy in PRC2-deficient cells, we performed chromatin immunoprecipitation and sequencing (ChIP-seq) for EZH2 in *Ezh2* WT and heterozygous tumoroids, and for H3K27me3 in *Ezh2* WT, heterozygous, null, and EPZ6438-treated WT tumoroids. Analysis of chromatin peaks showed

EZH2 occupancy was comparable between the *Ezh2* heterozygous and *Ezh2* WT tumoroids, while the H3K27me3 occupancy on the *Ezh2* heterozygous tumoroids was 75% of that in the *Ezh2* WT tumoroids (Fig. 3c and Supplementary Fig. 3b–d). In *Ezh2* null tumoroids, H3K27me3 peaks were 13% of the Ezh2 WT sample and nearly completely depleted in EPZ6438-treated *Ezh2* WT tumoroids. Interestingly, despite similar numbers of EZH2 peaks in *Ezh2* WT and heterozygous cells, only 45% of the peaks were shared between the genotypes (Fig. 3d). Similarly, there were a large number of H3K27me3 peaks that were unique to either *Ezh2* WT and *Ezh2* heterozygous cells. However, the majority of H3K27me3 peaks retained in *Ezh2* null cells were also found in the other two genotypes and corresponded to genes encoding master transcriptional factor gene families, including the

*Hox*, *Sox* and *Nkx* families (Supplementary Table 5 and Supplementary Fig. 3e). Further analysis of H3K27me3-bound regions showed that the majority of peaks retained in the *Ezh2* null and EZH2 inhibitor-treated tumoroids were in gene promoter regions, while peaks in other genomic locations were lost, suggesting that the small amount of H3K27me3 remaining in cells is preferentially retained at promoter regions to best control gene expression (Supplementary Fig. 3f). GREAT analysis revealed that although *Ezh2* heterozygous cells had lower enrichment of EZH2 at Polycomb targets genes, they retained H3K27me3 at Polycomb targets (Fig. 3e and Supplementary Table 6). Relative to *Ezh2* WT, *Ezh2* heterozygous tumoroids showed increased enrichment of EZH2 at genes associated with DNA damage repair, stemness and cell cycle, although these same gene signatures had only modest increases in H3K27me3 enrichment. Lastly, we examined H3K27me3 peaks in the *Ezh2* WT and null tumoroids and discovered that 32% of the DEGs we previously identified had peaks that were reduced by at least twofold in *Ezh2* null cells, including all seven of the DEGs shared between sorted and 3D cells, and notably, *Foxp2* (Fig. 3f and Supplementary Fig. 3g).

### Loss of PRC2 activity drives increased FOXP2 expression in normal and malignant lung cells

Given its critical roles in embryonic lung stem cells[26,27], we next sought to characterize FOXP2 expression in the murine samples. First, we confirmed the expression of FOXP2 in 2D and 3D cells at the mRNA and protein levels. *Foxp2* was elevated in *Ezh2* null cells relative to *Ezh2* WT cells in both 2D and 3D cultures (Fig. 4a). Furthermore, *Foxp2* mRNA levels were significantly higher in *Ezh2* null 3D tumoroids than in *Ezh2* null 2D cells. FOXP2 protein expression in *Ezh2* null tumoroids was also much higher than *Ezh2* WT and heterozygous tumoroids and all the 2D cells (Fig. 4b). To confirm the correct band for FOXP2 using this antibody, we used small interfering RNAs to inhibit the transcription of *Foxp2*, and observed a decrease only in the band between 75 and 100 kDa (Supplementary Fig. 4a, b). To test whether EZH2/PRC2 regulates FOXP2 expression through its enzymatic activity, we re-expressed EZH2 WT and enzymatic-incompetent F667I mutant EZH2 in the *Ezh2* null mouse tumor cells. We observed a significant reduction in FOXP2 protein expression only in EZH2 WT re-expression cells, but not in the F667I-mutant EZH2 re-expression cells (Fig. 4c, d). To support these in vitro findings, we verified that the FOXP2 expression in the mouse autochthonous tumors was significantly higher in the *Ezh2* null tumors than in the other two genotypes by immunostaining (Fig. 4e). ChIP-seq showed that H3K27me3 and EZH2 were both enriched at the 5′ TSS of *Foxp2* in the *Ezh2* WT cells, but these peaks were substantially reduced in the *Ezh2* null and EPZ6438-treated tumoroids (Fig. 4f and Supplementary Fig. 4c). ChIP-qPCR also showed reductions in H3K27me3 enrichment at the *Foxp2* 5′ transcription start site (TSS), regulatory element, and 3′ TSS, and significantly reduced enrichment at the sequence corresponding to the *Foxp2* translation start site (ATG) in *Ezh2* null cells compared with *Ezh2* WT (Supplementary Fig. 4d).

Next, to test if EZH2 inhibition leads to derepression of *FOXP2* in human samples, we measured FOXP2 expression in human cell lines treated with EPZ6438. Inhibition of EZH2 significantly elevated *FOXP2* mRNA expression in human cancer lines H460, H2030, A549, and H2009, and immortalized pulmonary epithelial cells BEAS2B and BEAS2B with additional *KRAS*^G12V and *TP53*^R175H mutations (BEAS-KP) (Fig. 4g). EZH2 inhibition also upregulated the protein expression of FOXP2 in human cancer lines H460, H2030, A549, and BEAS-KP, but not in immortalized epithelial cell lines HBEC3KT and BEAS2B (Fig. 4h, i). ChIP-qPCR showed a loss of H3K27me3 enrichment at a putative regulatory site 5′ of the FOXP2 gene in H460 and BEAS-KP cells treated with EPZ6438 (Supplementary Fig. 4e, f). To understand if the 2D vs. 3D expression differences observed in the murine cells could also be observed in human cells, H460 and H2030 were grown in both conditions and treated with EPZ6438. While EPZ6438 was able to drive

increased FOXP2 expression in both systems, 3D human cells exhibited much higher *FOXP2* mRNA levels than their 2D counterparts (Supplementary Fig. 4g). Together, these data suggest that EZH2/PRC2 directly regulates FOXP2 expression mainly through its EZH2 enzymatic activity, and that loss or inhibition of EZH2 increases FOXP2 expression in malignant lung cells of both mouse and human origin.

### FOXP2 dictates stemness and migratory capacity of lung epithelial cells

Having established FOXP2 as a direct target of PRC2 in lung epithelial cells, we sought to understand if FOXP2 could drive the phenotypic differences observed in *Ezh2* null tumor cells. We overexpressed FOXP2 in two immortalized pulmonary epithelial lines: HBEC3KT and BEAS2B, and two KRAS-driven lines: H460 and H2030 (Fig. 5a and Supplementary Fig. 5a). When grown in Matrigel cultures, FOXP2 overexpressing BEAS2B cultures yielded more, and significantly larger, organoids than the control cultures (Fig. 5b). Given the increased metastatic potential of *Ezh2* null tumor cells, we tested if FOXP2 overexpression changed migratory capability by transwell migration assays. We observed that FOXP2 overexpressing cultures had significantly increased migratory capacity when compared to control cultures for all the cell lines tested (Fig. 5c, d). Furthermore, FOXP2 overexpression conferred BEAS2B culture's resistance to etoposide and carboplatin (Supplementary Fig. 5b, c), consistent with the theory that cells in more stem-like states are relatively resistant to chemotherapy[36].

To discover the transcriptional programs changed by FOXP2 overexpression, we performed RNA-sequencing on FOXP2 overexpressing and control cultures. GSEA revealed that FOXP2 overexpression drove transcriptional amplification of many genes associated with EMT, cell adhesion, TNFα, RAS signaling, and lung development (Fig. 5e and Supplementary Table 7). The finding of RAS signaling enrichment was not unique to the H460 and H2030 KRAS mutated lines, but also observed in the HBEC3KT and BEAS2B lines. Lastly, to link FOXP2 overexpression in human cell lines to gene expression changes seen in *Ezh2* null spheroids, we examined *NDRG4* and *MYCN* in FOXP2 overexpressing or EZH2 inhibited H460 and H2030 tumoroids (Supplementary Fig. 5d, e). These two genes were highly upregulated in *Ezh2* null tumoroids, and are potential PRC2/FOXP2 downstream genes based on ENCODE data[37]. In these cultures, the combination of FOXP2 overexpression and EZH2 inhibition led to significant increases in *NDRG4* and *MYCN* compared to control cultures, confirming that these genes may be regulated by both EZH2 and FOXP2. Next, we tested whether the reduction of FOXP2 could reduce aggressive phenotypes of lung cancers. Knocking down FOXP2 with two different short hairpins in the human cell lines H460, A549, and H2009 remarkably impeded cell growth in 2D cultures (Fig. 5f, g and Supplementary Fig. 5f–i). FOXP2 knockdown also led to significantly smaller 3D tumoroids compared with the control *shGFP* in H460 cultures (Fig. 5h). Moreover, the H460 cells with FOXP2 knockdown migrated significantly less than the control, indicating a key role of FOXP2 in cell migration (Fig. 5i, j). Altogether, these discoveries identify a previously un-appreciated role for FOXP2 in the stemness and migration of normal and malignant adult lung epithelial cells.

### Polycomb deficiency drives BET and histone demethylase inhibitor sensitivity

We next aimed to identify epigenetic therapeutic vulnerabilities of PRC2-deficient KRAS+/*Trp53*-null lung adenocarcinomas. We hypothesized that because *Ezh2* null tumors were the most aggressive, methods to stabilize H3K27me3 or prevent the reading of H3K27ac could re-normalize the aberrant epigenetic state and reverse the aggressive phenotypes. Therefore, we focused our efforts on testing the histone demethylase inhibitor GSK-J4, and the BET inhibitor JQ1 (Fig. 6a). In 2D cultures, there were no differences in half-maximal inhibitory (IC$_{50}$) values among the *Ezh2* WT, heterozygous, and null cultures treated with

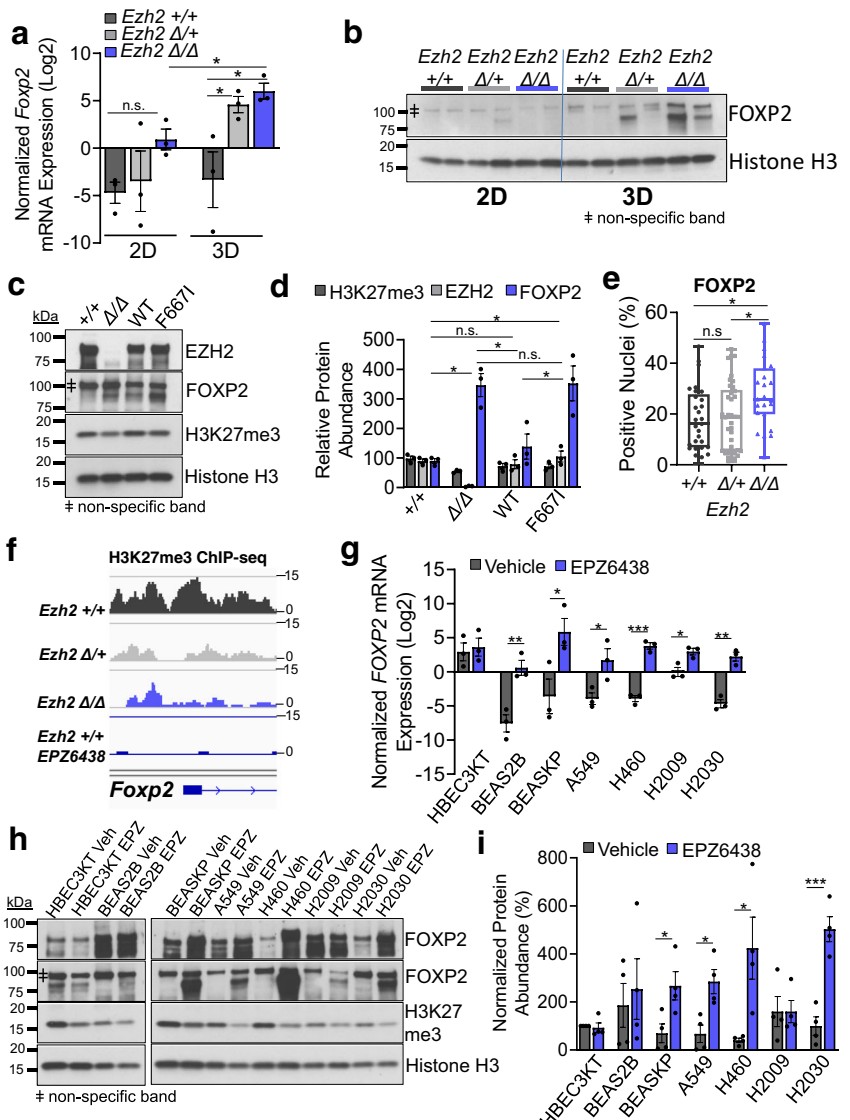

**Fig. 4 | Loss of PRC2 activity drives increased *FOXP2* expression in normal and malignant lung cells. a** Relative expression of *Foxp2* mRNA in mouse primary 2D tumor cells and 3D tumor spheroids with the indicated *Ezh2* genotypes graphed as mean values +/− SEM. * indicates $P < 0.043$ by one-way ANOVA with multiple comparisons and Holm−Šídák's post hoc test between genotypes and two-tailed *t* test between conditions. $n = 3$ individual mouse tumor-derived cultures per genotype/culture system. **b** FOXP2 protein expression in 2D cells and 3D tumoroids of the indicated *Ezh2* genotypes was examined by immunoblot. Total histone H3 is the same as Fig. 2e. ‡ indicates the non-specific band seen with FOXP2 antibody from Cell Signaling Technologies. **c** Immunoblot and **d** the normalized abundance of the indicated proteins in *Ezh2* WT, *Ezh2* null, *Ezh2* null with WT *Ezh2* re-expression (WT), and *Ezh2* null with F667I-mutant *Ezh2* re-expression (F6771). Quantification data are plotted as mean values +/− SEM. * indicates $P < 0.0261$ by one-way ANOVA with multiple comparisons and Holm−Šídák's post hoc test. $n = 3$ individual blotting experiments from two cell cultures. **e** Percentages of positively stained tumor nuclei for FOXP2 in the tumors of the indicated genotypes plotted as box-and-whisker plots, error bars are min−max, box bounds are 25th and 75th percentiles and center line is median. * indicates $P = 0.0233$ by one-way ANOVA

with multiple comparisons and Holm−Šídák's post hoc test for both comparisons. $n = 29, 38, 23$ for *Ezh2* WT, heterozygous, null individual mice, respectively. **f** Visualization of H3K27me3 ChIP-seq peaks enriched in the enhancer and promoter regions of *Foxp2* in the indicated 3D tumor spheroid samples. **g** Expression of *FOXP2* mRNA in human lines HBEC3KT, BEAS2B, BEAS-KP, H460, H2030, A549, and H2009 with or without 5 μM EPZ6438 treatment for 10 days plotted as mean values +/− SEM. * indicates $P < 0.043$, ** indicates $P < 0.009$, *** indicates $P = 0.0003$ by two-tailed *t* test. $n = 3$ individual cell cultures per cell line/condition. **h** Immunoblot and **i** the normalized abundance of the indicated proteins in human lines HBEC3KT, BEAS2B, BEAS-KP, H460, H2030, A549, and H2009 with or without 5 μM EPZ6438 treatment for 10 days. Top FOXP2 panel is SIGMA antibody, bottom FOXP2 panel is Cell Signaling Technologies antibody. Quantification data are plotted as mean values +/− SEM. * indicates $P < 0.032$, *** indicates $P = 0.000778$ with two-tailed *t* test. $n = 4$ with two individual blotting experiments derived from one cell culture per cell line/condition each probed with two distinct FOXP2 antibodies. Genotypes are signified as: WT, +/+; heterozygous, Δ/+; null, Δ/Δ. Source Data are provided for this figure.

GSK-J4 or JQ1 (Fig. 6b). However, *Ezh2* null tumoroids were significantly more susceptible to GSK-J4 or JQ1 when compared with *Ezh2* WT tumoroids (Fig. 6c and Supplementary Fig. 6a). *Ezh2* heterozygous tumoroids were similarly sensitive to JQ1 as *Ezh2* null, but only modestly sensitive to GSK-J4. The lack of consensus between 2D and 3D cultures prompted us to investigate drug vulnerabilities in vivo. Tumoroids were grafted subcutaneously onto immunocompromised mice. Intriguingly, consistent with the 3D culture system, the growth of *Ezh2* null grafts was inhibited by GSK-J4, and significantly inhibited by JQ1 treatments, when compared to *Ezh2* WT grafts, while the therapeutic effect of JQ1 on *Ezh2* heterozygous grafts was between the other two genotypes (Fig. 6d and Supplementary Fig. 6b).

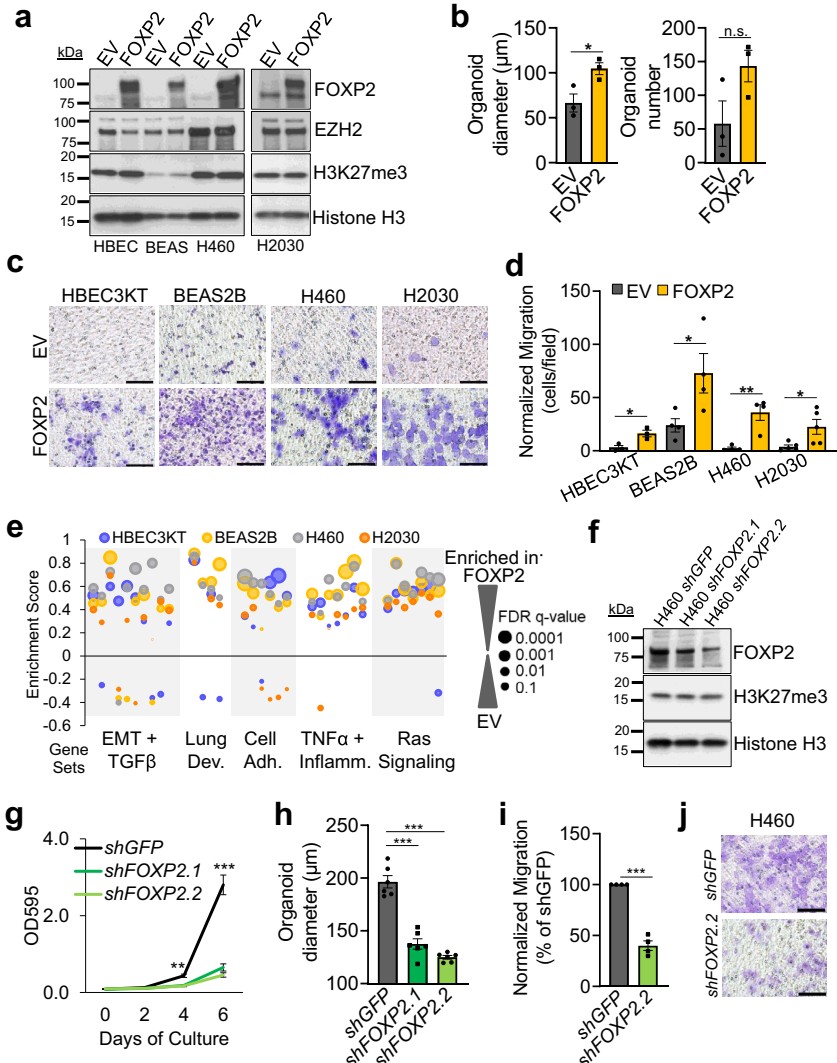

**Fig. 5 | FOXP2 drives stemness and migration in lung epithelial cells. a** FOXP2 protein expression in HBEC3KT, BEAS2B, H460, and H2030 cells transduced with FOXP2 lentivirus was examined by immunoblot. EV is the empty vector control line. **b** Average diameters and organoid counts of BEAS2B 3D organoids with or without FOXP2 overexpression plotted as mean values +/− SEM. * indicates *P* = 0.034 by two-tailed *t* test. *n* = 3 individual cell cultures per group. **c** Representative images, scale bar = 100 μm, and **d** average numbers of migrated cells normalized to growth rates in HBEC3KT, BEAS2B, H460, and H2030 cells with or without FOXP2 overexpression plotted as mean +/− SEM. * indicates *P* < 0.047, ** indicates *P* = 0.0041 by two-tailed *t* test. *n* = 3, 4, 5, 5 for HBEC3KT, BEAS2B, H460, H2030 individual cell cultures per group, respectively. **e** Rank-ordered gene lists were queried against the MSigDB databases and enrichment scores of selected gene signatures enriched in FOXP2 overexpressing cell lines relative to EV lines were plotted. Dot size estimates FDR. Data are summarized in Supplementary Table 7. **f** FOXP2 protein expression in 2D H460 cells transduced with the indicated small hairpins was examined by

immunoblot and representative of *n* = 4 blotting experiments from one cell culture per group. *shGFP* is the control cell line. **g** Crystal violet growth assays were performed on 2D H460 cells with or without FOXP2 knockdown at the indicated days of culture and mean values +/− SEM were graphed. ** indicates *P* < 0.0024, *** indicates *P* < 0.001 *shFOXP2* compared to *shGFP* by two-tailed *t* test. *n* = 6 individual cell cultures per group. **h** Average diameters of 3D H460 spheroids with or without FOXP2 knockdown plotted as mean values +/− SEM. *** indicates *P* = 0.000017 *shGFP* vs *shFOXP2.1* and *P* = 0.00000038 *shGFP* vs. *shFOXP2.2* tumoroids by two-tailed *t* test. *n* = 6 individual cell cultures per group. **i** Average percentages of migrated cells normalized to *shGFP* and relative cell growth rates graphed as mean values +/− SEM. *** indicates *P* = 0.000014 compared to *shGFP* cells by two-tailed *t* test. *n* = 4 individual cell cultures per group. **j** Representative images of migrated cells in H460 with or without FOXP2 knockdown. Source Data are provided for this figure.

To test if JQ1 could target tumors growing in the native lung microenvironment, we treated autochthonous tumor-bearing mice with JQ1 and measured changes in tumor volume by magnetic resonance imaging (MRI). Seven days of JQ1 treatment reduced the volumes of lung tumor nodules in *Ezh2* null and heterozygous GEMMs, while tumors continued to grow in the placebo group (Fig. 6e, f and Supplementary Fig. 6c). An additional seven days of JQ1 administration was sufficient to maintain tumors in their regressed state, though no additional tumor burden reduction was observed. To further explore the clinical significance of the drug vulnerability discoveries from murine tumors, we investigated several *KRAS* mutated human

non-small cell lung cancer lines. EZH2 knockdown sensitized the human line H2030 to JQ1 (Supplementary Fig. 6d, e), and such sensitization was more observable in 3D spheroid culture than 2D culture in H460 (Supplementary Fig. 6f). Additionally, drug matrix experiments demonstrated synergy when combining JQ1 with EPZ6438 in H2009 and BEAS-KP cells (Fig. 6g).

The differential response to BET inhibitor inspired us to interrogate whether FOXP2 was involved in this therapeutic effect in EZH2-deficient cells. First, we performed RNA-sequencing on the 3D tumoroids and subcutaneous grafts treated with JQ1, which revealed that BET inhibition led to significantly decreased *Foxp2* expression in *Ezh2* null cells

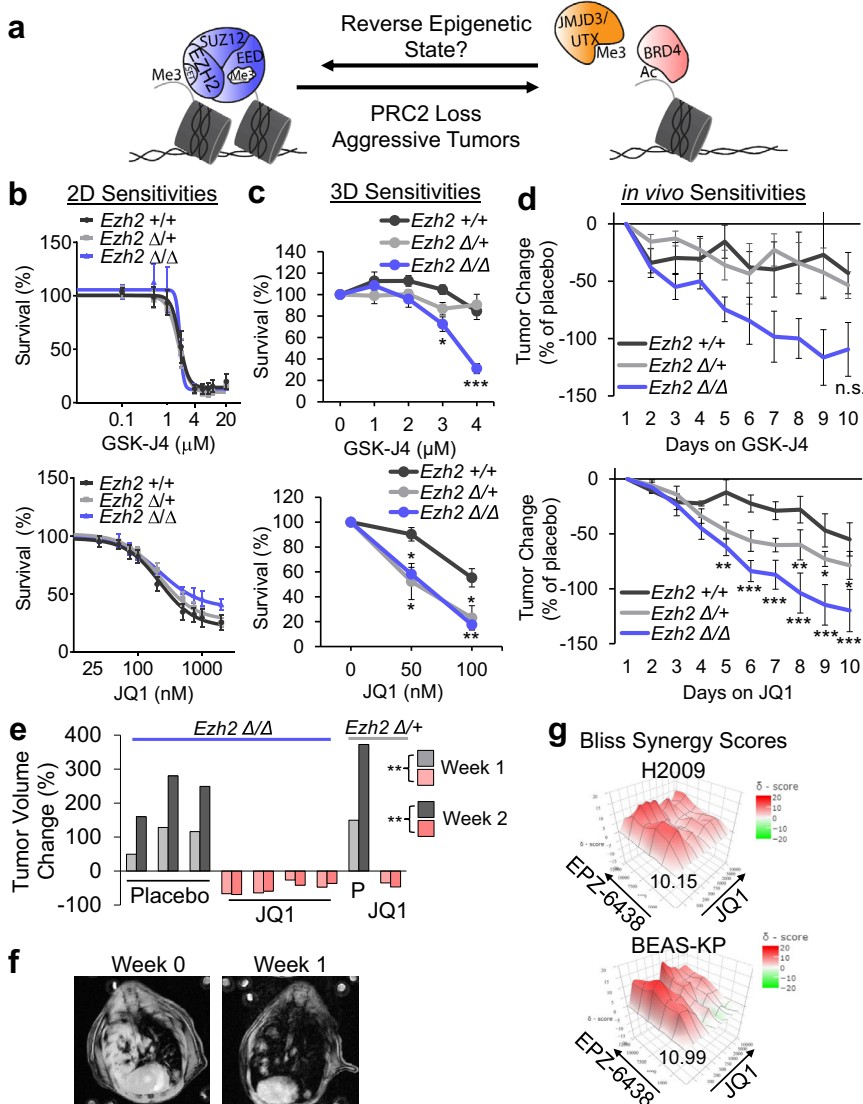

**Fig. 6 | Polycomb deficiency drives BET and JMJD3/UTX inhibitor sensitivity.**
**a** Schematic depicting the mechanisms to restore epigenetic states lost in PRC2-deficient tumors. **b, c** GSK-J4 (upper) and JQ1 (lower) percentage survival of (**b**) 2D cultures after 4 days culture with drugs plotted as mean values +/− SEM. GSK-J4: $n = 7, 7, 7$; JQ1; $n = 5, 6, 5$ for *Ezh2* WT, heterozygous, null individual mouse tumor-derived cultures, respectively. **c** 3D tumoroids after 12–14 days culture with drugs plotted as mean values +/− SEM. GSK-J4: * indicates $P = 0.0158$ *Ezh2* WT vs. *Ezh2* null, ***$P = 0.0006$ *Ezh2* WT or heterozygous vs. *Ezh2* null; JQ1: * indicates $P < 0.039$ and **$P = 0.0084$ for *Ezh2* WT vs. *Ezh2* null (lower) or *Ezh2* WT vs. heterozygous (upper) by one-way ANOVA with multiple comparisons and Holm-Šídák's post hoc. For both drugs, $n = 7, 5, 5$ for *Ezh2* WT, heterozygous, null individual mouse tumor-derived cultures, respectively. **d** Changes in volumes of grafts in mice treated with GSK-J4 and JQ1 relative to changes in placebo-treated mice plotted over time as mean values +/− SEM. *** indicates $P < 0.0008$, **$P < 0.0091$, *$P < 0.0472$ for *Ezh2* WT vs.

*Ezh2* null (lower) or *Ezh2* heterozygous vs. *Ezh2* null by one-way ANOVA with repeated measures and multiple comparisons. GSK-J4: $n = 5, 5, 4$; JQ1: $n = 5, 5, 6$ for *Ezh2* WT, heterozygous, null individual mouse tumor-derived grafts, respectively. **e** Quantification of relative tumor volume change after 1 week and 2 weeks of placebo or JQ1 treatment of *Ezh2* null or *Ezh2* heterozygous tumor-bearing mice, ** indicates $P = 0.0021$ for week 1, $P = 0.0003$ for weak 2 by two-sided *t* test on log2-transformed values. **f** Representative magnetic resonance images of mouse lungs at baseline (week 0) and week 1 of JQ1 treatment for an *Ezh2* null mouse. **g** Heatmap of Bliss synergy scores for EPZ6438 combined with JQ1 in H2009 and BEAS-KP cultures. The most synergistic areas had Bliss scores of 10.15 (H2009) and 10.99 (BEAS-KP), indicating synergy. $n = 3, 4$ for H2009 and BEAS-KP individual cell cultures, respectively. Genotypes are signified as: WT, +/+; heterozygous, Δ/+; null, Δ/Δ. Source Data are provided for this figure.

(Supplementary Fig. 6g). *c-Myc*, a gene often changed by JQ1 treatment[38], was also significantly decreased in expression in the JQ1-treated graft, confirming the drug's efficacy. RT-qPCR on mouse 3D tumoroids showed that *Foxp2* was significantly inhibited by 100 nM JQ1 in *Ezh2* heterozygous and null tumoroids, but not in *Ezh2* WT tumoroids (Supplementary Fig. 6h), which may explain the difference in drug sensitivity among the genotypes. In the JQ1-treated H2030 cultures, *FOXP2* levels were significantly decreased in the *shEZH2* lines, but not in the *shGFP* line, again suggesting that regulation of *FOXP2* correlated to the increased sensitivity of cells to JQ1 (Supplementary Fig. 6i). Similarly,

in EZH2 WT rescued *Ezh2* null tumoroids, which we demonstrated had lowered FOXP2 levels, JQ1 was significantly less effective than in *Ezh2* null tumoroids re-constituted with the F677I EZH2 mutant that retain high levels of FOXP2 (Supplementary Fig. 6j). Lastly, we observed H460 overexpressing FOXP2 were more sensitive to JQ1 than the control cells, and again the increase in sensitivity was more apparent in 3D cultures than in 2D (Supplementary Fig. 6k, l). Taken together, these data indicate reduction in PRC2 activity confers sensitivity to BET inhibition and demethylase inhibition in lung adenocarcinoma, and FOXP2 could be a promising target of BET inhibition.

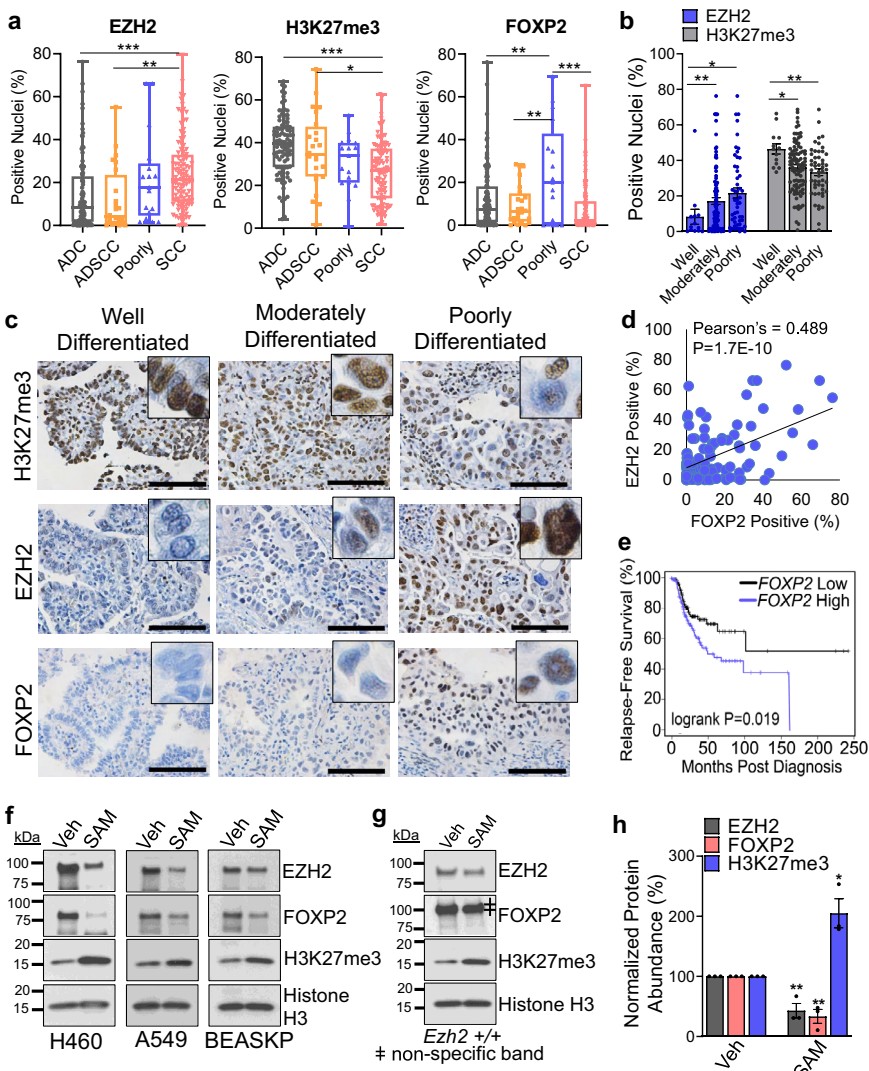

**Fig. 7 | Decoupling of EZH2 from PRC2 activity may explain co-expression of EZH2 and FOXP2 in patient samples. a** Positively stained tumor nuclei in ADC, ADSCC, poorly differentiated tumor and SCC specimens for the markers EZH2, H3K27me3 and FOXP2 plotted as box-and-whisker plots, error bars are min–max, box bounds are 25th and 75th percentiles and center line is median. * indicates $P = 0.0164$, ** indicates $P < 0.0084$, *** indicates $P < 0.0006$ by one-way ANOVA with multiple comparisons and Holm–Šídák's post hoc test. $n = 92, 24, 17, 104$ for ADC, ADSCC, poorly differentiated and SCC individual tumors, respectively. **b** The percentage of EZH2 and H3K27me3 positive nuclei in lung adenocarcinoma and poorly differentiated cancer specimens. * indicates $P < 0.0425$, ** indicates $P < 0.0086$ by one-way ANOVA with multiple comparisons and Holm–Šídák's post hoc test. $n = 13$, 51, 97 for well, moderately, poorly differentiated individual tumors, respectively. **c** Representative images of lung adenocarcinoma specimens defined as well, moderately, and poorly differentiated stained for the indicated markers, scale bar = 100 μm. The number of tumors stained are summarized in (**b**). **d** Correlation between EZH2 and FOXP2 based on the percentage of positively stained tumor nuclei in primary lung cancer specimens. Pearson's correlation coefficients and p values are shown. $n = 161$ individual tumors. **e** Kaplan–Meier relapse-free survival curves for *FOXP2*-high and *FOXP2*-low tumors as measured by RNA-sequencing 250 months post diagnosis; positive and negative groups were determined by best cut-off. The $P$ value was calculated by log-rank test. $n = 366$ individual tumors. **f, g** Immunoblot in 2D (**f**) H460, A549, BEAS-KP, and **g** *Ezh2* WT UK803 cells treated with or without 500 μM SAM for 6 days. **h** Normalized protein abundance in H460, A549, and BEAS-KP cells treated with or without 500 μM SAM for 6 days plotted as mean values +/− SEM. * indicates $P = 0.012$, ** indicates $P = 0.009$ for EZH2 and $P = 0.0045$ for FOXP2 by two-tailed $t$ test. $n = 3$ different cell lines. Genotypes are signified as: WT, +/+; heterozygous, Δ/+; null, Δ/Δ. SAM, S-adenosyl methionine. Source Data are provided for this figure.

## Decoupling of EZH2 from PRC2 activity may explain co-expression of EZH2 and FOXP2 in patient samples

To extend our findings to human lung cancer patient samples, we performed immunohistochemical staining for EZH2, H3K27me3, and FOXP2 on lung cancer tissue microarray. Consistent with previous research[6,9], the H3K27me3 stain was significantly higher in ADC than SCC, while EZH2 had the opposite pattern (Fig. 7a). FOXP2 expression was variable in ADC samples, but found to be significantly higher in poorly differentiated tumors than in the other three groups. Because the mouse model we used was adenocarcinoma, and became more poorly differentiated when *Ezh2* was deleted, we next focused only on ADCs and poorly differentiated lung tumors. In this cohort, we

observed that both well and moderately differentiated tumors had significantly lower EZH2, and significantly higher H3K27me3 marks, than poorly differentiated tumors, which had lower H3K27me3, but paradoxically had higher EZH2 levels (Fig. 7b, c). Intriguingly, EZH2 expression showed a strong positive correlation to FOXP2 expression (Fig. 7d). In tumors that were FOXP2 positive (>10% of nuclei) that mirror the high expression found in some *Ezh2* heterozygous and most *Ezh2* null tumors, there was a significant inverse relationship between EZH2 and H3K27me3 (Supplementary Fig. 7a). To study the correlation of EZH2/FOXP2 expression with patient prognosis, we queried published RNA-sequencing data from The Cancer Genome Atlas using KM-Plotter[39] on 366 lung adenocarcinoma samples. Consistent with

previous reports, higher *EZH2* expression correlated with significantly worse relapse-free survival in lung adenocarcinoma (Supplementary Fig. 7b). In our tissue microarray, EZH2 expression also correlated with a worse overall survival, but it is important to note that in our dataset, SCC patients had a significantly shorter overall survival than did lung ADC patients, which coincides with the fact that SCC tumors usually have higher EZH2 expression (Supplementary Fig. 7c, d). Importantly, higher *FOXP2* expression correlated with significantly shorter relapse-free survival up to 250 months post diagnosis (Fig. 7e).

Given that EZH2 is rarely mutated in NSCLC, one possible reason for observing decoupling in tumors is that in highly proliferative tumors EZH2 is upregulated, but there may be a limited availability of the methyl donor, S-adenosyl methionine (SAM)[10]. To examine the hypothesis that SAM levels influence PRC2 stability and activity, 500 μM SAM was added to the culture media of human lines H460, A549, and BEAS-KP for 6 days. As expected, these human lines treated with SAM had higher H3K27me3, and interestingly had significantly lower EZH2 and FOXP2 compared to the controls (Fig. 7f, h). Treating an *Ezh2* WT murine 2D culture with SAM led to a similar decoupling of EZH2 and the H3K27me3 mark (Fig. 7g and Supplementary Fig. 7e). In summary, EZH2 expression decouples from PRC2 activity, and a pattern of high EZH2, low H3K27me3 and high FOXP2 is enriched in poorly differentiated advanced lung cancers.

## Discussion

Here, we uncovered un-appreciated roles for EZH2 heterozygosity and full deficiency in regulating tumor status, whereby *Ezh2* heterozygous tumors had lower tumor burden, lower nuclear grade and were less lethal than *Ezh2* null tumors. Although the *Ezh2* heterozygous tumoroids had a 25% reduction in H3K27me3 marks, they had the strongest suppression of canonical PRC2 target genes, suggesting that increased canonical PRC2 activity, rather than partial loss, drives *Ezh2* heterozygous phenotypes. These cells also had changes in cell cycle, increased type-I interferon response pathways, and in vivo had an enrichment for T cell infiltration, and these mechanisms may explain the lower tumor burden in *Ezh2* heterozygous mice. In contrast, *Ezh2* null tumors were more metastatic and higher grade. Despite the fact that EZH2 inhibitors may drive more aggressive phenotypes in cancer, there still may be clinical utility for EZH2 inhibitors due to effects on the tumor microenvironment, and the ability to drive additional therapeutic vulnerabilities of tumor cells. Our data suggest that in KRAS-driven lung cancers, a PRC2-deficient state, either occurring naturally or through EZH2 inhibition, may be vulnerable to BET inhibition. We observed that *Ezh2* null tumoroids, subcutaneous grafts and autochthonous lung tumors were sensitive to the BET inhibitor JQ1. Mechanistically, we determined that FOXP2, which was previously identified as a key regulator of embryonic lung progenitor cells[28], was downregulated in several cases where BET inhibition was effective, and in the H460 cell line, overexpression of FOXP2 conferred increased sensitivity to JQ1. We confirmed that FOXP2 was a direct target of PRC2-mediated gene silencing in malignant lung epithelial cells, and that enforced expression of FOXP2 increased stemness and migration of lung epithelial cells.

It is notable that our drug response findings and the identification of FOXP2 were discovered in vitro only due to the use of 3D tumoroid cultures. We found that murine 2D cultures had extremely high levels of S-phase and down-regulation of genes involved in lung lineage determination when compared to 3D tumoroids, which showed retention of lung specification transcriptional programs. This finding is consistent with the belief that 3D cultures are able to retain more characteristics of the native tissue from which they were derived[40], and demonstrates the vast differences between the same cells cultured in 2D versus in 3D. The outcomes of loss of PRC2 activity in 3D and 2D cells were also divergent. In the *Ezh2* null genotype, 2D cell lines had marked derepression of *Hox* genes, but in 3D tumor spheroids, *Ezh2* null cells were able to continue to repress *Hox* genes and instead showed derepression of *Foxp2*. Together, these findings underscore the power of using 3D systems for genetic, epigenetic and drug validation experimentation in cancer research.

Finally, while the co-expression of FOXP2 and EZH2 in patient samples may seem contradictory to the results from our mouse model, we believe these results support each other. Several papers, including our own, have shown an inverse correlation between EZH2 and its catalytic mark, H3K27me3 in vivo[6–9,41–43]. This abundant EZH2 may be part of PRC2-independent complexes[44–46], or unable to complete histone methylation for a variety of reasons (one reason postulated is lack of SAM). Therefore, high EZH2 can indicate lower PRC2 function, and our mouse model and in vitro data show that one result of lowering PRC2 function in KRAS-driven adenocarcinomas is to de-repress FOXP2. In this way, the mouse model, with EZH2 deletion to drive PRC2 dysfunction, and the human samples, with high EZH2 indicating PRC2 dysfunction, do match. Furthermore, retrospective analysis of patient sample RNA-sequencing data showed that high *FOXP2* expression predicted significantly shorter relapse-free survival. Our data suggest that when SAM is abundant, which may be the case in more well-differentiated tumors and normal tissues, PRC2 activity is high and EZH2 is downregulated. However, in aggressive tumors that often have low levels of H3K27me3[7,47], EZH2 may be upregulated, but unable to perform its catalytic function due to lack of available SAM. Future research focuses on understanding the physiological or pathophysiological metabolic mechanisms by which the SAM cellular pool are regulated, especially in vivo, and whether this process can be reversed for therapeutic intervention.

## Methods

### Cell lines

Cell culture experiments were done in accordance with Institutional biosafety regulations. All human cell lines except HBEC3KT were maintained in RPMI 1640 media (Gibco, #11875-093) with 8–9% fetal bovine serum (VWR), glutaMAX (Gibco, #35050-079) and penicillin/streptomycin (Gibco #15140-122) at 37 °C, 5% $CO_2$. A549, H460, H2030, and H2009 cells were obtained from the laboratory of Dr. Carla Kim at Boston Children's Hospital, BEAS2B cells were a kind gift from Dr. Chengfeng Yang at University of Kentucky, and HBEC3KT cells[48] were a kind gift from Dr. David Orren at University of Kentucky. HBEC3KT cells were seeded onto gelatin pre-coated plate and maintained in PneumaCult-Ex Basal media (StemCell Tech, #05009) with PneumaCult-Ex Supplement (StemCell Tech, #05019), hydrocortisone (Sigma, #H0888) and penicillin/streptomycin at 37 °C, 5% $CO_2$. All human lines were verified by STR genotyping at IDEXX laboratories and no mycoplasma was detected in cultures by MycoAlert Mycoplasma Detection Kit (Lonza). Plasmocin (InvivoGen, #ant-mpt) was added into cultures regularly. Mouse primary 2D cells were obtained by plating freshly dissociated mouse tumors on tissue culture plates and passaging cells that adhered and grew. These cultures were maintained in DMEM F-12 media (Gibco, #11330-032) with 8–9% fetal bovine serum, 0.9% insulin–transferrin–selenium (Corning or Gibco), 4 mM L-glutamine, penicillin/streptomycin and plasmocin at 37 °C, 5% $CO_2$. For three-dimensional Matrigel culture of primary cells, see below "3D tumoroid culture".

### Animals

All care and treatment of experimental animals were in accordance with Boston Children's Hospital and University of Kentucky institutional animal care and use committee (IACUC) approved protocols. Cohorts of LSL:*Kras*^G12D/+^; *Trp53*^flox/flox^; *Ezh2*^flox/flox^ (LSL: lox-stop-lox) genetically modified mouse models[49,50] were maintained in virus-free conditions on a mixed C57BL/6x129SVj background. Mice were bred predominately as *Ezh2*^fl/+^ so that littermates and close relatives of the three *Ezh2* genotypes could be analyzed for differential survival,

metastasis, and tumor burden studies. For survival, mice that had healthy body characteristics at time of sacrifice were excluded from analysis. Both male and female mice were used for all experiments and no sex differences were noted. Adult mice received $2.5–2.9 \times 10^7$ PFU adeno-Cre by intranasal instillation. Mice were evaluated by Magnetic resonance imaging (MRI) 12 weeks after adeno-Cre infection for baseline scans. For *Kras/Trp53/Ezh2* mouse models, JQ1 in aqueous suspension were made by diluting DMSO stock dropwise with 1:1 w/w Captisol in sterile saline while vortexing and was i.p. injected at 50 mg/ kg q.d. for 2 weeks. For Nude mouse allograft experiment, *Kras/Trp53/ Ezh2* mouse tumor spheroids were dissociated into single cells, counted, and resuspended at $1 \times 10^5$ cells per 200 µL of 1:1 media/ Matrigel (Corning, #356231). Female Athymic Nude *Foxn1^Nu/Nu^* mice (Envigo) were injected subcutaneously with $1 \times 10^5$ cells in 2–4 spots on flanks. Tumors were allowed to grow for 21 days to a mean size of 60 mm³. Mice were then randomized into groups that received GSK-J4, JQ1 or vehicle. 100 mg/kg GSK-J4 or 50 mg/kg JQ1 dissolved with 100% DMSO were administered by i.p. injection q.d. for 10 consecutive days. Tumors were measured by electronic caliper and tumor diameters were used to calculate tumor volumes (tumor volume = (length × width²)/2).

## Patients samples

Patient slides were provided by the Markey Cancer Center Biospecimen Procurement & Translational Pathology Shared Resource Facility (BPTP SRF) of University of Kentucky. Leftover clinical specimens were obtained by BPTP with patient informed consent or waiver under a protocol approved by the University of Kentucky Institutional Review Board (IRB), and when transferred to our group, the samples had been de-identified and therefore exempt from further Institution Review Board approval. The tissue microarray included 83 lung adenocarcinomas (ADCs), 14 lung adenosquamous carcinomas (ADSCCs), 102 lung squamous cell carcinomas (SCCs), and 17 poorly differentiated tumors including one sarcomatoid tumor, one giant cell carcinoma, two large cell neuroendocrine and one pleomorphic carcinoma. For these cases, three core biopsies were removed from a larger tumor piece and arrayed on slides. All of the cases were confirmed for cancer subtype and differentiation state by a board-certified pathologist. To increase sample size, we also used whole-tumor slides from an additional 9 ADCs, 10 ADSCCs, and 2 SCCs.

## Flow cytometry analysis and sorting

Tumors were dissected from the lungs of primary mice and tumor tissue was finely chopped with surgical scissors. Tumor chunks were incubated with 1 mL 1× PBS with 60 µL Collagenase/Dispase (Sigma, #10269638001) in a rotator at 37 °C for 30 min. After washing with 1× PBS, chunks were resuspended in 100 µL 0.25% Trypsin-EDTA (Gibco) for 2 min, then neutralized with 900 µL PBS containing 10% FBS (PF10). The digested tissue was passed through a 100-µm filter (VWR), then a 40-µm filter (VWR) into a 50cc conical tube. Cell pellets were resuspended in 100 µL Red Cell Lysis buffer (0.15 M NH4Cl, 10 mM KHCO3, 0.1 mM EDTA, in 1 L distilled H₂O; filtered with 0.45-µm filter and stored at RT) for 2 min at RT and neutralized by 800 µL PF10. Single-cell suspensions were stained using rat-anti-mouse antibodies for: anti-mouse-EpCAM-PECy7 (BioLegend, 1:100); anti-mouse-CD31-APC (BioLegend; 1:100); and anti-mouse-CD45-APC (BioLegend; 1:100). Live cells were gated by the exclusion of 4′,6-diamidino-2-phenylindole (DAPI) positive cells (Sigma, #D9542l 4 µg/mL). All antibodies were incubated for 10–15 min. Cell sorting was performed with a Sony iCyt with a 100-µm nozzle.

## 7AAD cell cycle flow cytometry analysis

Cells propagated in 2-dimensional dishes or 3-dimensional transwells were separated into single cells by 0.25% Trypsin. In all, $1–10 \times 10^5$ cells were spun down in a 1.5 mL microtube and resuspended in cold 300 µL PF10. The resuspended cells were added dropwise to 700 µL cold 70%

ethanol with slow vortexing and were incubated at 4 °C for least 1 h. Cells were pelleted by pulse spin and resuspended in 250 µL/tube of 1 mg/mL RNase A (Thermo Fisher, #EN0531) diluted in PBS for 30 min at room temperature. Cells were washed by 1 mL PBS, pelleted and resuspended in 250 µL/tube of 4 µg/mL 7-aminoactinomycin D (Invitrogen, #A1310). 30,000 events/sample were collected on the BD LSRII and analyzed with the ModFit LT software and results were averaged for three or four biological replicates.

## MRI of genetically engineered mouse models

A Brucker ClinScan system that has 12 cm actively shielded gradients with a maximum strength of 630 mT/m and slew rate of 6300 T/m/s was used. The tumor-bearing mice were anesthetized by isoflurane inhalation, kept warm by a heating pad and imaged on the 7 T system with a 2 × 2 array coil with a 2D gradient echoT1-weighted sequences with parameters: 18 slices, TR = 170 ms, TE = 2.4 ms, α = 38°, Navg = 3, FOV 26 × 26 mm², 1 mm thickness, matrix size 256 × 256, for a voxel size of 0.102 × 0.102 × 1.0 mm. Images were gated to the animal's respiratory cycle using a pneumatic respiratory monitor (SA Instruments) to remove breathing motion artifacts. The cardiac cycle and temperature were monitored using SA instruments hardware and software. The tumor burden volume and quantification were reconstructed on 3D Slicer software (http://www.slicer.org).

## 3D tumoroid culture

Mechanically and enzymatically dissociated tumor cells were resuspended in DMEM/F-12 media containing penicillin/streptomycin, glutaMAX, 8–9% fetal bovine serum, 10 µg/mL Insulin (SIGMA, #I-6634), 0.9× Insulin/transferrin/selenium mixture (Corning), 12.5 µg/mL bovine pituitary extract (Invitrogen, #13028-014), 0.1 µg/mL cholera toxin (SIGMA, #C8052), 25 ng/mL mEGF (Invitrogen, #53003-018) and 25 ng/mL rmFGF2 (R + D Systems, #3139-FB/CF), mixed 1:1 with growth factor-reduced Matrigel (Corning), and pipetted into a 24-well 0.4-µm Transwell insert (Corning, #CLS3470). A 3D air–liquid interface Matrigel culture was created by placing tumor chunks or roughly 5000 tumor cells into each well. MTEC/Plus (described above) medium (500 µL) was added to the lower chamber and refreshed every other day. Tumoroids usually form in 10–14 days. Upon second and third passage, more homogeneous tumoroids arose with little residual debris remaining. To passage tumoroids, media in the lower chamber was aspirated, 100–150 µL of Dispase (Corning, #42613-33-2) was added over the top of the well, and the Matrigel was dislodged to float. After incubation at 37 °C for 1–2 h, the Matrigel was liquefied and the spheroids were pipetted into a tube, trypsinized to passage, or used for assays including CellTiter-Glo readings, RNA isolation or ChIP. Large differentiated tumoroids with dense squamous-like structures were removed. EPZ6438 (MCE, #HY-13803), GSK-J4 (MCE, #HY-15648B) and JQ1 (MCE, #HY-13030) were stocked as 10 mM solution in DMSO and were diluted according to the designated concentration into tissue culture media for use.

## Cell viability assay

For the treatment of 2D cultures, cell lines were dissociated, counted, and plated as 50 µL media with 2000 cells (for human cells) or 5000 cells (for mouse cells) per well in 96-well plates. Edge wells were filled with 135 µL PBS for hydration. In all, 50 µL media with different concentrations of drugs EPZ6438, GSK-J4, JQ1 or carboplatin (Sigma, #C2538) were added after 24 h. After 96 h, 50 µL CellTiter-Glo (Promega, #G7573) was added into each well and the chemical luminescence was measured using a BioTek Cytation5. Values were normalized to vehicle for each experiment to yield percent survival, and drug doses were converted to log scale for graphing as log(inhibitor) vs. response-variable slope (four parameters). Vehicle wells were set to 1E-10. Results from independent experiments were entered into the GraphPad Prism software to extrapolate IC₅₀ and *P* values. For synergy

assays, combinations of different doses of EPZ6438 with JQ1 were added into 96-well plates as a synergy matrix and measured by CellTiter-Glo. The percent survival for each well compared to the control well from three to four independent experiments were uploaded to https://synergyfinder.fimm.fi/ (version 2.0) to calculate Bliss synergy score (readout: viability; baseline correction: yes) and display 3D synergy heatmap (>10 means synergism). For 3D cultures, organoids were seeded as described in "3D tumoroid culture", and into the lower media drugs were added. Organoids were cultures for 9–14 days, at which point Matrigel was dissolved with dispase, solution with cells was moved to 96-well-opaque plates, incubated with Cell-Titer-Glo, and luminescence was read on the Cytation5. Relative luminescence was calculated based on each experiment's vehicle control cultures. For Supplementary Fig. S6j, in order to query more JQ1 doses, cells were seeded in 20 μl of 50% Matrigel into 384-well Corning Spheroid microplates at 500–1000 cells/well. An additional 10 μl of medium was applied to the top of Matrigel after it solidified. Three days later, 20 μl of drug solutions were added to the wells. After treatment for 3 days, cell viability was measured by adding 25 μl of CellTiter-Glo 3D (Promega), shaking for 5 min, incubating at room temperate for 25 min, and reading luminescence on the Cytation5. For Supplementary Fig. S6I, we followed the protocol as for Fig. 6j but with JQ1 treatment starting the day after seeding cells and lasting for 4 days. Values were normalized to vehicle for each experiment (six replicate wells) to yield percent survival, and drug doses were converted to log scale for graphing as log(inhibitor) vs. response-variable slope (four parameters). Vehicle wells were set to 1E-10 and tops were constrained to 100% while bottoms were constrained to zero.

## Crystal violet growth assay

Cell lines were dissociated, counted, and 1 mL media with 1000 cells per well was seeded into four 12-well plates per sample. The four plates were marked as day 0, day 2, day 4, and day 6. Then, 6–8 h after seeding, media was aspirated and cells in the day 0 plate were fixed with 10% neutral-buffered formalin for 30 min RT. After removal of formalin, the plates were washed with water 2–3 times, air-dried and inverted. The same procedures were followed on day 2, day 4 and day 6 plates. After day 6, all the four plates were stained with 400 μL crystal violet solution (1 g crystal violet (0.5%, VWR #97061-850), 150 mL ddH$_2$O, 50 mL methanol) for at least 30 min. Then the crystal violet solution was removed, and the plates were washed with water three times, dried and inverted. The stained crystal violet was then released into solution with 400 μL 10% glacial acetic acid (in water, Sigma #A-6283). In total, 100 μL of the dissolved crystal violet solution was aliquoted into a 96-well plate and OD$_{595}$ absorbance was obtained with a Cytation5 (Biotek).

## Quantitative RT PCR and RNA-sequencing

RNA from treated cell lines was extracted using Absolutely RNA kits (Agilent) and cDNA was generated using the SuperScript III kit (Invitrogen). Relative gene expression was assessed with Taqman probes (Thermo Fisher, #4318157) on the Quant Studio 3™ Real-Time PCR System (Applied Biosystems). Relative expression was calculated by Gene of Interest $(Ct_{reference} - Ct_{exprimental})$-House-keeping gene$(Ct_{reference} - Ct_{experimental})$ and graphed on the log$_2$ scale. Housekeeping genes were either *GAPDH/Gapdh* or *ACTB/Actb* (B-actin) (Applied Biosystems). Statistics were performed on log$_2$ data. For all experiments except Supplementary Fig. S6A, the reference Ct value was the average of Ct values from each culture and treatment for a given experiment, which allowed plotting of relative mRNA levels among cultures. For Supplementary Fig. 6e, the reference Ct values were the average of Ct values from the EV control cells for each cell line. For RNA-sequencing, small bulk tumor pieces, 10,000–50,000 freshly sorted tumor cells, or 0.5–1 million of 2D or 3D cultured cells were suspended in lysis buffer for RNA extraction

(Agilent columns). RNA was extracted by Agilent Nanoprep (3D and sorted tumors, Agilent #400753) or Microprep (2D cells, Agilent #400805) kit. Library preparation and sequencing were sent to and performed by the University of Kentucky Oncogenomics Shared Resource Facility or by Beijing Genomics Institute (BGI Group). For sorted tumor cells, the Smart-SeqII low-input protocol was used. A total of $n = 17$ total tumor, $n = 13$ sorted tumor, $n = 20$ 2D and $n = 21$ 3D samples were used for clustering and further analysis. Several samples were run twice to test for any batch variations and to sequence more well-established cultures, and the later samples were retained for analysis. One *Ezh2* null tumoroid that was an outlier in both in vitro and in vivo drug treatment experiments was excluded from genotype comparison analyses, and the data are available upon request. Sequencing reads were trimmed and filtered using Trimmomatic (V0.39)[51] to remove adapters and low quality reads. Reads from human and mouse samples were mapped to Ensembl GRCh38 and GRCm38 transcripts annotation (release 82), respectively, using RSEM[52]. Gene expression data normalization and differential expression analysis were performed using the R package edgeR[53]. The "filterByExpr" function in edgeR package was used to keep genes with count-per-million (CPM) > = 10 in at least n samples, where n is the smallest group sample size. Significantly up/downregulated genes were determined as |LFC| >0.5, FDR < 0.05.

## Principal component analysis (PCA)

For PCA, the transcripts per million (TPM) read counts of each transcript were log-transformed using log of TPM + 1 (using the log1p function in R for numerical stability), and then PCA decomposition was performed using the prcomp R function[54]. Each sample was designated as either being from 2-dimensional cultures (2D) or not (other). For each principal component (PC), the sample scores between 2D and other were compared using ANOVA. To find transcripts with statistically significant loadings on a PC, each transcript's loadings in that PC were tested. All other transcript's loadings in all the other PCs were used to create a null distribution of loadings, and then the number of loadings greater than the tested transcript's loading value and in the same direction (positive or negative) counted, and divided by the number of null loadings in that direction. This fraction is the reported $P$ value for that transcript's loading in that PC. Transcript loadings values were considered significant if they had a $P$ value < = 0.05. The log-transformed TPM values of transcripts with significant loadings in PC3 were used to calculate an information-theoretically weighted correlation, where the raw Pearson correlation value is weighted by the information content between the two samples based on their Jaccard metric, and the fraction of entries that are consistently present (nonzero TPM) and missing (zero TPM) between the two samples. The weighted correlation was then used as a pseudo distance (1 − correlation) for hierarchical clustering and ordering using the dendsort package[55]. Heatmaps of the correlation were generated using ComplexHeatmap[56].

## Gene ontology enrichment

Gene ontology (GO) enrichments of the PC3-associated genes were done using a custom version of categoryCompare[57]. GO terms from all three of biological process, molecular function, and cellular component were tested simultaneously. GO annotations to genes are based on mapping the Entrez Gene IDs to GO terms in the Bioconductor packages org.Mm.eg.db and GO.db version 3.11.4. The false discovery rate correction was generated using the Benjamini−Hochberg method. Only those GO terms with an FDR < = 0.05 and annotated to two or more genes are reported. The top 40 GO terms by $P$ value are listed in Supplementary Table 1.

## Gene set enrichment analysis

GSEA was performed with GSEA version 4.0.3 (Broad Institute) with rank-ordered gene lists generated using all log-fold change values.

Mouse_ENSEMBL_Gene_ID_to_Human_Orthologs_MSIGDB.v7.1.chip was used to map mouse genes to human orthologs. Databases queried included Hallmarks (h.all.v7.1), Curated (c2.all.v7.1), BioCarta (c2.cp.biocarta.v7.1), KEGG (c2.cp.kegg.v7.1), Reactome (c2.cp.reactome.v7.1), GO (c5.all.v7.1), Oncogenic Signatures (c6.all.v7.1), and Immunologic Signatures (c7.all.v7.1). Gene signatures that converged on common themes were selected for further analysis and graphing of enrichment scores and FDRs for specific contrasts. Signatures are listed in Supplementary Tables 2–5 and 7.

### Chromatin immunoprecipitation and qPCR (ChIP-qPCR)

Tumor cells were resuspended in Buffer A from Covaris kit (Covaris, #520154). Then cells were cross-linked with 1% formaldehyde for 5 min at RT, quenched by Quenching buffer E, washed in cold PBS and pulse spun. Pellets were resuspended in Lysis buffer B and rotated for 10 min at 4 °C. Lysed nuclei were washed with Wash buffer C, resuspended in Sonication buffer D3 and sonicated with the Covaris M220 Focused-ultrasonicator to obtain chromatin fragment lengths of 200–1500 bp judged by a Bioanalyzer DNA High sensitivity kit (Agilent #5067-4626). Fragmented chromatin was diluted in Sonication buffer D3 and incubated overnight at 4 °C with antibodies including anti-GFP (Thermo Fisher MS-1288-P0, 5 µL per 300 µL chromatin), isotype IgG (Cell Signaling #3900 S, 5 µL per 300 µL chromatin), H3K27ac (Abcam, #ab4729, 5 µL per 300 µL chromatin) or H3K27me3 (Millipore, #07-449 for mouse, 5 µL per 300 µL chromatin, Cell Signaling #9733 s for human, 10 µL per 300 µL chromatin). Then 40 µL Protein A/G 1:1 mixed magnetic Dynabeads (Invitrogen) were added and rotated at 4 °C for 2 h. Immunoprecipitates were washed with cold ChIP buffer (1% Triton X-100, 0.1% deoxycholate, 50 mM Tris-HCl pH 8.1, 150 mM NaCl, 5 mM EDTA), High salt buffer (1% Triton X-100, 0.1% deoxycholate, 50 mM Tris-HCl pH 8.1, 500 mM NaCl, 5 mM EDTA), LiCl buffer (250 mM LiCl, 0.5% IGEPAL, 0.5% deoxycholate, 10 mM Tris-HCl pH 8.1, 1 mM EDTA) and finally TE buffer (10 mM Tris-HCl and 1 mM EDTA). Immunoprecipitated (or no IP input) DNA was incubated with 50 µL Elution buffer (1% SDS and 0.1 M NaHCO$_3$) in a 55 °C water bath overnight. The eluted DNA was treated with RNase A and Proteinase K, and the column was purified with DNA clean and concentrator kit (Zymo Research, #D4014). ChIP-qPCR was run with the Power SYBR reagents (Thermo Fisher, #100029284) on the Quant Studio 3™ Real-Time PCR System for the mouse Foxp2/human FOXP2 TSS, ATG, and enhancer regions. Primers are listed in Supplementary Table 8.

### ChIP sequencing

For chromatin immunoprecipitation (ChIP) analysis, three biological replicates of tumoroids of each Ezh2 genotype were expanded. The Ezh2 WT tumoroids were treated with 5 µM EPZ6438 or the same amount of DMSO for 12 days, and the Ezh2 heterozygous and null tumoroids were treated with DMSO for 12 days. Tumoroids were collected, briefly dissociated and counted to mix equal numbers of cells from each biological replicate with 5 × 10E6 cells for each ChIP. The protocol from Active Motif was followed for preparation of the cells. Briefly, the tumoroids were fixed by adding freshly prepared 11% formaldehyde (with 1 mM EDTA and 50 mM HEPES) to the cells in media with shaking at room temperature for 15 min. 1/20 volume of 2.2 M glycine was added to stop the fixation and incubated at room temperature for 5 min. Cells were pelleted at 800×g for 10 min and washed with in 10 mL cold PBS- IGEPAL® CA-630 (0.5%). Cells were again pelleted and washed in 10 mL cold PBS-IGEPAL® CA-630 (0.5%) and 100 µL PMSF (100 mM in ethanol). Finally, cells were pelleted and snap-froze on dry ice and sent to Active Motif for ChIP sequencing using the antibodies EZH2 (Active Motif 39901, 8 µL antibody per 40 µg chromatin) and H3K27me3 (Active Motif 39155, 4 µL antibody per 40 µg chromatin). The quality control and read alignment was performed by Active Motif. Briefly, the 75-bp single-end sequence reads were mapped to the mouse reference genome mm10 using the bwa samse with default settings. Reads that had >2 mismatches and multimapping reads were removed, followed by PCR deduplication. The resulting bam files were normalized to account for the differences in the sequencing depth. The EZH2 samples were reduced by random sampling to the number of unique alignments present in the smallest sample. Spike-in normalization method was performed for the H3K27me3 samples by down-sampling to result in the same number of spike-in Drosophila reads for each sample. Since the 5′-ends of the aligned reads represent the end of ChIP/IP-fragments, the tags were extended in silico using Active Motif software at their 3′-ends to a length of 200 bp. To identify the density of fragments along the genome, the genome was divided into 32-nt bins and the number of fragments in each bin was determined. The SICER version 1.1 peak finding algorithm was used to identify regions of ChIP-seq enrichment over the background, with window size 200 bp, fragment size 200 bp, gap size 600 bp, and FDR value threshold of enrichment 1E-10 for all datasets. Genomic regions known to have low sequencing confidence were removed using blacklisted regions defined by the ENCODE project. The selected peak intervals were annotated to the nearest transcription start sites (TSS) using the KnownGene mm10 TSS annotation. To compare peak metrics, overlapping intervals were grouped into merged regions, defined by the start coordinate of the most upstream interval and the end coordinate of the most downstream interval. In locations where only one sample has an interval, this interval defines the merged region. Peak distribution patterns were obtained using seqplots across all merged intervals from −5 kb to +5 kb to include distal promoters and regulatory regions. The heatmaps are used for visualization of tag distributions which are mapped across target regions and were clustered into 4 groups based on the tag densities using k-means algorithm. The average values for all target regions in heatmaps were calculated and plotted in histograms. Peaks unique to each genotype or conserved in multiple genotypes were annotated by GREAT[58] to associate each genomic region with all genes whose regulatory domain it overlaps. The resulting gene list was subjected to gene set enrichment analysis to identify significantly enriched gene signatures from GSEA-curated signature gene sets (Supplementary Table 6).

### Western blotting

Whole cell extracts were made in RIPA buffer (0.5% deoxycholate, 1% IGEPAL-CA-630, 0.1% sodium dodecyl sulfate, 150 mM NaCl, 50 mM Tris-8.1), lysates were cleared by centrifugation, and protein concentrations were quantified with the Pierce BCA Protein Assay Kit (Thermo Fisher, #23225). For western blots, 1–60 µg of protein extract per sample was denatured with heat and reducing agents, separated on a 4–15% acrylamide gel (BioRad) and transferred to nitrocellulose membranes (GE Healthcare, #10600002). Antibodies used for western blots were: EZH2 (Cell Signaling, #5246s, 1:1000), H3K27me3 (Cell Signaling, #9733s, 1:1000) and FOXP2 (Sigma #HPA000382 1:1000, or Cell Signaling #5337 1:1000), PCNA (Biolegend Cat# 307901, 1:1000), or Histone H3S10ph (pH3, GeneTex, #GTX128116, 1:1000) incubated overnight at 4 °C. Histone H3 (Abcam, Cat# ab1791, 1:5000-1:10,000) was used as loading control on blots loaded with ~1/4 of the total lysate from the same protein dilution run at the same time. Secondary anti-rabbit IgG, HRP-linked antibody (Novus #NB7160, 1:10,000) was incubated for 1 h at room temperature. After washing, chemiluminescence was visualized with West Pico PLUS Chemiluminescent Substrate (GE Healthcare) and exposure onto Hyperfilm (Amersham). After H3K27me3 visualization, some blots were stripped with an SDS/β-mercaptoethanol solution at 55 °C for 30 min and re-probed with total H3.

### Histology and immunohistochemistry

Mice were anesthetized, hearts were removed and the lung lobes were inflated with neutral-buffered 10% formalin and fixed overnight at

room temperature, and then transferred to 70% ethanol, embedded in paraffin and sectioned at 4−5 μm. Hematoxylin and eosin, alcian blue, and Masson's Trichrome stains were performed by the Biospecimen Procurement & Translational Pathology Shared Resource Facility (BPTP SRF) of the University of Kentucky Markey Cancer Center or at Harvard Medical Center. Nuclear grade was determined by Dr. Roderick Bronson at Harvard Medical School on a set of histology slides representing tumors harvested 111 days or more past adeno-Cre. Descriptions for nuclear grade criteria were previously published[49]. Grade 5 criteria were not used. Grade 4 included the criteria listed in the original paper and the appearance of tumor cells invading into normal tissue or into blood vessels. Immunohistochemistry was performed with either citrate, Dako High pH, or Ventana CC2 antigen retrieval in the Brainson Laboratory or the BPTP SRF, and staining was visualized with brown DAB substrate and counterstain was either Mayer's or Harris's hematoxylin. Slides were batch stained to reduce variability. Antibodies used for immunostaining were: EZH2 (Cell Signaling, #5246s, 1:300), H3K27me3 (Cell Signaling, #9733s, 1:500), FOXP2 (Sigma #HPA000382 1:200 for mouse, 1:125 for human), and CD3 (Dako IR503, ready-to-use dilution). Stained slides were scanned on an Aperio slide scanner and three areas of the largest tumors (10×−14.6× magnification) were selected for analysis. Regions of stroma (blood cells, vessels, fibroblasts and connective tissue) were deleted from the images by hand in Photoshop. Edited images containing predominantly tumor nuclei were analyzed using Nikon's dark spot detection module on the blue channel at eight different equally spaced intensity cutoffs, which represent increasing stringency for positive IHC stain. A moderately stained tissue section was used for calibration of the dot detection macro before being applied equally to all images for a given stain. Nuclei counted at intensities 5−8 were considered positive, while nuclei counted at intensity 1 represented all nuclei for a given image. The percentage of positive nuclei in bins 5−8, or moderately-strongly positive nuclei in bins 6−8 were calculated, with the exception of the murine H3K27me3 stain, which was lighter and therefore bins 4−8 were considered positive and bins 5−6 were moderately positive. Additional imaging was performed on a Nikon Ti-Eclipse inverted microscope. For HALO® image analysis (Indica Labs), H&E-stained tumor slides scanned with an Aperio slide scanner were loaded and a nuclear profiler was trained to recognize the cell types indicated with the help of Markey Cancer Center pathologists. For the final version, a total of 20,491 nuclei were selected from 23 different mouse and human tumor samples were selected and the training had completed over 400,000 iterations. Visual inspection of samples and running of control areas (lymph nodes, highly neutrophilic regions etc.) allowed for the selection of the final macro, which was confirmed by two Markey Cancer Center pathologists for accuracy. Regions of interest, including the tumors and approximately a 50 μm of edge space around the tumors to include tumor-infiltrating immune cells, were set on the 24 experimental slides. For the *Ezh2* null tumors, the regions identified to be EZH2 positive by IHC were excluded. Then the macro was run on all samples together. For CD3 stains, six areas with the highest levels of CD3 staining were selected from each mouse slide, non-tumor/lymphoid areas were masked, CD3 stain was quantified using the Nikon AR software, and the area of unmasked tissue was quantified using ImageJ.

## Tumor burden and survival analysis

H&E-stained slides were scanned with an Epson scanner at 3200 dpi resolution 48-bit color. Images were cropped, white space was masked and trachea/esophagus/lymph nodes were masked. Images were saved as 16-Bit grayscale, then opened in ImageJ software. Threshold settings were first set to cover all of the lung and measurements were taken, 0.0001-infinity (inch²). Then auto threshold was set, and modified if needed to exclude areas that were heavily infiltrated with immune cells but not tumor, and to exclude thickened bronchi, and measurements

were taken, 0.0001-infinity. The ratio of tumor to lung was calculated for each mouse. For some mice, a representative lobe or residual lung with many tumors intact or partially intact were used so that tumors could be harvested for other analyses. Mice were excluded from analyses when the majority of tumors were removed for other analyses or when lung was not collected for histology. For *Ezh2*^fl/fl^ lungs, if an EZH2 positive tumor area was noted, an image of the EZH2 immunohistochemistry slide scanned at 20× by an Aperio slide scanner was used for ImageJ analysis. The steps were the same as for the H&E-stained slides, except the measurement settings were from 500-infinity (pixel²) and then EZH2 positive areas were manually masked, measured and removed from the total tumor area. For the survival curves, mice that died or that were close to endpoint as judged by mouse health or observed tumor burden were included. Mice that appeared healthy or that were not noted to have extremely high tumor burden at sacrifice were not included as they may have been able to survive much longer.

## Transwell cell migration assay

Migration assays were performed with 24-well transwell plates with 8-μm Pore (Corning, #CLS3422). Briefly, cells were trypsinized into single cells by 0.25% Trypsin-EDTA and counted. Then $1-5 \times 10^5$ cells in 100 μl serum-free culture media was added into the top chamber insert and incubate for 10 min. The bottom chamber was filled with 600 μl media containing 10% FBS. After 24-h incubation, non-migratory cells were removed with a cotton swab, and the migratory cells were fixed with formalin for 10 min, 70% ethanol for 10 min, stained with 0.2% crystal violet for 10 min and washed off with tap water. Ten ×20 images of each transwell were taken on BioTek Cytation5, the numbers of crystal violet-stained cells were counted manually and the average number of cells per well was calculated. The data were then normalized to the relative growth rate of the cell lines assessed by plating cells for 24 h in opaque 96-well plates and reading luminescence by CellTiter-Glo.

## Viruses and siRNA

The pLKO.1 *EZH2* shRNA (TRCN0000040076 and TRCN0000040073) and *FOXP2* shRNA (TRCN0000426742, TRCN0000426178) construct clones were purchased from SIGMA and the *shGFP* plasmid #12273 and pENTR-*FOXP2* plasmid #47053 were available on Addgene[59]. The FOXP2 overexpression construct was flipped into pLenti6.3/V5-DEST vector (Invitrogen). The murine *Ezh2* expression vectors plasmids #24927 and plasmid # 24926 were purchased from Addgene[60]. For BEAS-KP cell generation, KRAS^G12V^ and p53R175H lentiviruses were used[61] and these cells were characterized in a previous publication[62]. KRAS^G12V^ was a kind gift from Aveo Pharmaceuticals and p53^R175H^ was a kind gift from the Kuperwasser laboratory at Tufts Medical Center. Viruses were packaged in HEK293T cells with above vectors together with the packaging construct pCMV DR8.2Dvpr and VSV-G-pseudotyped vectors by FuGENE 6 (Promega, #E2691), using established protocols[63]. Cell lines were infected with viral-containing supernatant for a period of 4−8 h. Infected cultures were selected with puromycin (hairpins) or blasticidin (FOXP2 and p53^R175H^) 4−5 days post infection. The siRNAs were purchased from Invitrogen and included Silencer™ Select Negative Control (#4390843) and Silencer™ Pre-Designed siRNAs (#AM16708, ID #85875 and ID #85870). Transfection of cells was performed in six-well plates using 100 pmol of each siRNA and Lipofectamine 2000 (Invitrogen) in additive-free media. Cells were collected 48 h after transfection for western blotting.

## Statistics and reproducibility

All graphed data were presented as mean values +/− standard error of the mean (SEM), except box-and-whisker plots, for which the center line shows the median value, the box extends from the 25th to 75th percentiles, and upper and lower whiskers extend to the highest and lowest values, respectively. For mouse experiments, the biological *n*

represented individual donor mice, and often several replicate experiments for each donor mouse were averaged to produce the biological replicates. Experimental replicates are all explicitly stated in the figure legends. Each experiment was performed in at least an experimental duplicate (i.e., individual cell cultures or RNAi sequences), biological duplicate (i.e., individual mouse or human donor) or both. As indicated, *P* values represent unpaired two-tailed *t* test with equal variance that were used to compare continuous outcomes between two experimental groups (e.g., 2D and 3D), or one-way ANOVA with pairwise comparisons and Holm–Šídák's post hoc test that were used for comparing multiple groups (e.g., *Ezh2* WT, heterozygous and null), with the exception of nuclear grades and tumor burden over time, which were one-way ANOVA with multiple comparisons raw *P* values. ANOVA with repeated measures was used to compare the tumor change at each timepoint in the tumor growth studies. Fisher's exact tests were used to compare binary outcomes between groups. Kaplan–Meier curves and log-rank tests were used for survival outcomes. Pearson's correlation coefficients were used to quantify correlations of marker expressions. A *P* value less than 0.05 was considered statistically significant. Statistical analyses were carried out using Microsoft Excel, GraphPad Prism or SAS version 9.4.

### Reporting summary
Further information on research design is available in the Nature Portfolio Reporting Summary linked to this article.

## Data availability
The RNA-sequencing data generated during this study are available at NCBI GEO database under accession number: GSE154689. The ChIP-sequencing data is available under accession number GSE182819. Five sorted samples of varied genotypes were excluded due to questions about sorter operation that day. Those additional data are available upon request. MSigDB database is available at https://www.gsea-msigdb.org/gsea/msigdb/. The summarized GSEA data generated in this study are provided in the Supplementary Information file. Source data are provided with this paper.

## Code availability
Principal components analysis and correlation analysis were performed using R (v 4.0.0), and functions available in visualizationQualityControl (v 0.3.6, https://github.com/moseleybioinformaticslab/visualizationQualityControl). Gene ontology enrichments used categoryCompare2 (commit 906721, https://github.com/moseleybioinformaticslab/categoryCompare2), as well as Bioconductor (v 3.11), org.Mm.eg.db (v 3.11.4), and GO.db (v 3.11.4). Other R packages used directly include: rmarkdown (v 2.1); dplyr (v 1.0.10); viridis (v 0.6.2); circlize (v 0.4.15); ComplexHeatmap (v 2.4.3); readr (v 2.1.3); ggplot2 (v 3.4.0); dendsort (v 0.3.3). All direct codes, as well as inputs and outputs necessary for the principal components analysis, are available from a figshare repository at https://doi.org/10.6084/m9.figshare.13179989.v1.

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

## Acknowledgements

The authors thank Dave Powell in the Markey Cancer Center Small Animal Imaging Facility (MCCSAIF) for extensive help with the MRI scanning, the Yang and Orren laboratories at the University of Kentucky for sharing cell lines, Aveo Pharmaceuticals and the Kuperwasser laboratory for sharing vectors, Dr. Roderick Bronson of Harvard Medical School for assessing tumor sections, and Joanne M. Berry and Yanming Zhao for assistance with mice. This work was supported in part by NCI K22 CA201036, Kentucky Lung Cancer Research Program, V Foundation Scholar Award, American Cancer Society Grants IRG-85-001-25 and 133123-RSG-19-081-01-TBG, NCI R01 CA237643, the American Institute for Cancer Research, and American Association for Cancer Research-Bayer Innovation and Discovery Grant (C.F.B.), NIGMS P20 GM121327-03 (C.F.B. and H.N.B.M.), NIEHS T32 5T32ES007266 (T.J.D. and A.L.B.), and NHLBI F31 HL151111 (A.L.B.). This research was also supported by the Biostatistics & Bioinformatics Shared Resource Facility, Oncogenomics Shared Resource Facility, Biospecimen Procurement & Translational Pathology Shared Resource Facility and Flow Cytometry & Immune Monitoring Shared Resource Facility of the University of Kentucky Markey Cancer Center (P30CA177558). Finally, the Markey Cancer Center's Research Communications Office assisted with manuscript preparation.

## Author contributions

Conceptualization: F.C., C.W., J.L., H.N.B.M., R.M.F., and C.F.B.; data curation: J.L., T.J.D., F.C., C.F.B., and C.W.; formal analysis: F.C., J.L.,

R.M.F., and C.F.B.; funding acquisition: H.N.B.M., C.W., and C.F.B.; data acquisition and processing: F.C., A.L.B., T.J.D., X.S., A.R.E., A.L., D.T.D., C.M.G., D.P.E., and C.F.B.; *Ezh2* conditional mouse model: SHO; Bioinformatics and biostatistics: R.M.F., R.D.J., H.N.B.M., J.L., and C.W.; writing—original draft: F.C. and C.F.B.; writing—review & editing: F.C., R.M.F., X.S., H.N.B.M., and C.F.B.

## Competing interests

The authors declare no competing interests.
