## [Peer Review File · Nature Communications]

Reviewers' comments:

Reviewer #1 (Remarks to the Author):

The manuscript by Chen et al entitled "Polycomb deficiency drives a FOXP2-high aggressive state targetable by epigenetic inhibitors" reported the characterization and functional investigation into heterozygous or homozygous deletion of Ezh2 in a Kras-driven lung adenocarcinomas models. The authors observed that Ezh2 haplo-insufficient tumors were less lethal and lower grade compared with Ezh2 fully insufficient tumors. In addition, the authors explored the culture conditions on gene expression, which identified Foxp2 as a target of EZH2 whose upregulation may account for certain aspects of observed phenotypes. Further, based on previous publications, the authors explored the possibilities of targeting EZH2 inhibited cells with BET or H3K27 demethylase inhibitors. Finally, the authors performed correlative studies in human lung cancer. While there are several interesting observations presented by the authors, the manuscript is fragmented without mechanistic understanding with various findings. In addition, there are many issues with interpretation of results. The contradiction between mouse models and human lung cancer further weakened the conclusion. Therefore, the manuscript is preliminary at this stage and pre-mature for publication.

1. The differences observed between heterozygous (oncogene) and homozygous (tumor suppressor) Ezh2 knockout are interesting. However, the underlying mechanism is completely not clear. One would expect EZH2, H3K27me3 and SUZ12/EED ChIP-seq to dissect EZH2 methyltransferase dependent vs. independent function in these settings. Along these lines, a rescue experiment with wildtype or methyltransferase inactive mutant EZH2 is essential for dissecting the relevant mechanisms. For all the analysis, heterozygous cells should be included (instead, most of comparisons were done only between wildtype and homozygous knockout, which will not address the differences).
2. There is a major disconnect between mouse models and human lung cancer data. Mouse models suggest that Ezh2 loss upregulates FOXP2 through unknown (and likely methyltransferase activity because EZH2 inhibitor increases FOXP2 expression) to promote a more aggressive phenotype. However, the data as presented in Figure 7 do not support this conclusion. For example, EZH2 expression instead positively correlate with FOXP2 expression and the survival data support both EZH2 and FOXP2 as oncogenes. Instead, Figure 7 appears to focus on the discrepancy between EZH2 and H3K27me3 expression observed in the tumors (through investigating SAM availability).
3. The comparison between 2D vs. 3D culture conditions is again interesting. However, the results as presented are descriptive with the exception of identification of FOXP2 as a potential EZH2 target gene. Please see specific comments below.
4. The use of JQ1 and GSK-J4 in the context of EZH2/H3K27me3 inhibition has been reported, which reduces the novelty of the relevant data. Although authors have a unique opportunity to dissect the mechanism using the mouse model generated, there is no attempt on this front (for example, to differentiate methyltransferase dependent vs. independent effects of EZH2 etc).
5. Many statements in the manuscript are simply not supported by results, which include but not limit to those without statistical significance.

Specific comments:

1. For all the migration assays, please normalize for cell growth rates that were clearly affected in 2D culture conditions where these assays were performed.
2. There is a disconnect between grade of tumors observed in Ezh2 homozygous knockout mice and Ezh2's potential role in differentiation status (e.g., Fig. S3D).
3. For FOXP2 westernblot, please clarify which band is FOXP2 in Fig. 4b and perhaps include FOXP2 knockdown cells as controls.
4. Ezh2 -/- increased FOXP2 expression in Fig. 4b. If upregulation of FOXP2 determines the sensitivity to JQ1 in Ezh2 -/- cells, the enhanced effect of JQ1 should be observed in Ezh2 -/- in Fig. 6b.
5. The authors claimed that the reason why FOXP2 was not identified in 2D condition is because its expression is too low to generate enough reads (page 8, Fig. 3a). The result showed in Fig. 4b would

argue against this conclusion if the band(s) are indeed FOXP2. A direct comparison of FOXP2 between 2D and 3D condition would be informative.

0. Please also examine EZH2, H3K27me3 and FOXP2 in Fig. 4b for comparison between 2D and 3D culture conditions.
1. For all the ChIP analysis, please clarify how the normalization was performed (over input or IgG controls). In addition, IgG controls should be included throughout if this is not the case.
2. For Fig.4h, please clarify whether the experiments were done in 2D or 3D condition. In addition, please clarify whether there is a difference in regulating FOXP2 by EPZ-6438 between 2D and 3D conditions.
3. Likewise, the author showed EZH2 KD had distinct effect to JQ1 sensitivity between 2D and 3D in Fig. S6. It would be interesting to check whether EZH2 KD differentially affect FOXP2 expression between 2D and 3D conditions.
4. Fig. 5a, please clarify which band is FOXP2 band.
5. Fig. 5h-k: as shown in Fig. S7j, FOXP2 expression is relatively low in H460 cells. Please clarify the reason why the authors chose this cell line for FOXP2 knockdown (and, in particular, the authors categorize H460 cell line as JQ1 resistant, suggesting that FOXP2 level in this line is not sufficient to confer the phenotype).
6. Fig. 7: it is interesting EZH2 and FOXP2 are positively correlated in patient samples, while the points of the mouse models were to demonstrate an upregulation of FOXP2 by Ezh2 inhibition. Likewise, patients with low EZH2 expression have a better prognosis. This is in contrast with phenotype observed in Ezh2 homozygous knockout mice.
7. The rationale for including only ADC and poorly differentiated tumors in the analysis in Fig. 7 is not clear.
8. Please include all the cell lines as shown in Fig. 5a for Fig. 7i.
9. Some of the data as presented are marginal at the best (despite the indicated significance, e.g., Fig. 5f and 7i)
10. Figure S5f and S5h, the knockdown of FOXP2 does not correlate with spheroid numbers. Also, the knockdown efficiency was quite low.
11. Results showed in Fig. S6b does not match those in Fig. 6d. Contrary to the statement in results section, there is no difference in Foxp2 expression among the three conditions in 100 nM JQ1 treatment in Fig. S6h. It is surprising that EZH2 knockdown alone did not affect cell growth in Fig. S6.
12. Statements related to Fig. S7e and 7h are not accurate. For Fig. S7i-j, please specify the experiment was performed in 2D or 3D condition.
13. Please include statistical analysis in all the analysis where applicable.

Reviewer #2 (Remarks to the Author):

This manuscript describes the phenotypes of EZH2 heterozygous and homozygous deletions in non-small cell lung models, finding divergent effects across genotypes as well as differences across model systems. The demonstration that loss of one copy of EZH2 may be tumor suppressive while loss of both copies may be tumor promoting and increase aggressive phenotypes is of general interest. The authors take both a genetic and a pharmacological approach which make the study comprehensive and their conclusions generally accurate and relevant. However, the main message of the study is not presented clearly since although the divergent phenotypes of the EZH2 heterozygous vs homozygous gets lost in the multiple comparisons of 2D vs 3D vs in vivo models, which in most cases only investigate the wt vs homozygous deletion models. Within this context, it is unclear how authors rationalize the use of EZH2 inhibitors since this would imply a functional 'total ablation' of EZH2 function, similar to the homozygous deletion, which has a pro-cancer aggressive phenotype in their models. In addition, there are multiple instances of "data not shown" and even an instance of "data to be published in a separate manuscript" which are not acceptable if the missing data is to be considered toward the conclusions of the present work. Also, statistically non-significant data is mentioned in multiple cases as following a 'trend' and used to drive conclusions. Although some of

these conclusions have other data to back it up, others do not and need to be addressed or conclusions altered. All these general concerns should be addressed. In addition, in more specific terms authors should comment on/address the following points:

1. What is the direct evidence of cell cycle 'arrest' in the models presented in Fig 2 and Supp Fig 2 vs a difference in cell cycle distribution/timing of cell division?
2. It is concerning that only one gene was shared across the three models systems in the EZH2 deleted models of Fig 3/Supp Fig 3, EZH2 itself. Can authors address this at some level?
3. Of the eleven genes shared by the 3D and the sorted tumors, what is the rationale for focusing mainly on FOXP2 other than the stated observations? The other 10 genes may have functional consequences as well. Please address.
4. The differences seen in immune cell infiltration and signatures seem key. Can authors expand on the differences across all three phenotypes seen in vivo and in 3D models? This seems important for really understanding the tumor suppressive vs promoting effects of the heterozygous vs homozygous tumors, respectively. More of this data could perhaps be included in main Fig 3 rather than supplemental. Similarly, the finding that heterozygous tumors were enriched in histone methylation signatures should be functionally explored at some level to make a connection to the phenotype of these tumors vs the homozygous deleted tumors.
5. What is the mechanism behind the immune repertoire differences observed? Is this transcriptionally driven by EZH2/PRC2 or non-transcriptional? Can any further insights be added?
6. Have authors truly established that FOXP2 is a "direct target" of EZH2/PRC2? The data presented is consistent with this possibility but no PRC2 component has been ChIPed on FOXP2 gene regulatory regions as far as I could tell.
7. What are the effects of FOXP2 k/o or o/e in the heterozygous setting?
8. Fig 7 and Supp Fig 7 appear weak as correlations are not robust and some are not even significant. Authors should carefully review all this data and only present conclusions that are fully warranted and significant. In addition, how each group was defined (ie, well, moderately or poorly differentiated or expression quartiles) should be more clearly defined in the fig and legends. Finally, it is hard to really related the genotypes studied in mice with the levels of expression in human tumors. Again, a more careful and conservative wording of conclusions is warranted in this regard.

More minor points:

1. EZH2 levels in the heterozygous deleted tumors appear lower than expected (Fig 1/Suppl Fig 1). Have Western blots been done on tumor tissue? Any comments?
2. How authors define "more poorly differentiated" tumors in both animal models and human data should be explained in more detail
3. Throughout the manuscript please add "n.s" for all non significant statistical differences and alter text so as not to include small non-significant small changes as 'trends' supporting any conclusion.
4. Authors may consider using more obvious differences in color schemes to distinguish wt vs heterozygous models throughout the figures. The two gray scales used are too close.
5. The SAM experiment is speculative. Has it been repeated and confirmed in some other independent way?

Reviewer #3 (Remarks to the Author):

The authors investigated the role of haplo- and full-insufficiency of EZH2 in a GEMM of lung adenocarcinoma (KRASG12D/+, Tp53-/-; herein referred to KP mice). EZH2 is the functional enzymatic component of the PRC2 complex. The PRC2 complex is a histone methyltransferase particularly at H3K27 to end up with tri-methylation of H3K27 (H3K27Me3), which acts as a gene silencer. Mutant EZH2 can render PRC2 inactive and lead to acetylation of H3K27, which activates gene transcription. While it would make sense if increased EZH2 expression leads to increased H3K27 methylation, it has been observed to have a reciprocal relationship in this manuscript and in the literature cited. Thus, the rationale for this study ultimately investigates the cellular conditions as to

whether EZH2 acts as a tumour suppressor or oncogene. The authors found significant differences in culture methods, where 3D organoid culture, but not 2D, recapitulated gene expression patterns found in in vivo tumours better. FOXP2 was identified as a direct target to de-repression by non-functional PRC2, assessed by ChIP-qPCR. High FOXP2 expression was seen with EZH2 null murine cancer cell lines, which showed "stemness" and capacity to migrate. These high FOXP2 cells were also resistant to carboplatin treatment. Inhibitors against histone demethylase (GSK-J4) and BET inhibitor (JQ1) showed the best efficacy in EZH2 null murine cancer cell lines, with JQ1 being the best at attenuating allograft tumours. In the clinic, a decoupling of EZH2 and H3K27Me3 levels are seen. The authors hypothesise that EZH2 proficient cells are highly proliferative non-differentiated cells, which contribute to low H3K27Me3 because of the non-abundance of SAM, a substrate for tri-methylation of H3K27. Supplementation of SAM to a human cell line reversed this decoupling, however, this effect was less evident in EZH2 proficient murine cell lines. Overall, the authors present a very interesting manuscript with different technical and theoretical ideas, however, some points need to be elaborated further.

Major comments:

1. The authors conclude that EZH2 acts as an oncogene based on analysis of KP mice proficient for Ezh2 or harboring loss of either one (D/+) of two (D/D) Ezh2 conditional alleles (Figure 1). Comparisons of survival, tumor burden and tumor grade, should be examined amongst all genotypes (+/+ vs D/+ vs D/D) for a fair conclusion to be drawn to support this statement. Specifically:
 - a. Is the survival of Ezh2+/+ mice vs Ezh2 D/+ mice significant?
 - b. In Figure 1e, the nuclear grade of Ezh2+/+ and Ezh2D/+ mice are largely comparable, albeit, Grade 4 lesions appear to be decreased in Ezh2D/+ mice, but comparable between Ezh2+/+ and Ezh2D/D mice. Statistics for all genotype comparisons should be performed.
 - c. The time-point analysis of tumor burden, as shown in Figure S1, is a more accurate representation of tumor burden at defined time-points, than what is shown in Figure 1c, which appears to be a combination of all time-points. The increased burden seen in Ezh2D/D mice and increased metastasis is consistent with Ezh2 acting as a tumor suppressor gene, rather than an oncogene.
 - d. Related to the above point, in Figure S1, the authors demonstrate that Ezh2 expression is still observed in some Ezh2D/+ and Ezh2D/D tumours. Are these "positive" tumours also included in the quantification of tumor burden (particularly important in the context of Ezh2D/D mice)?
 - e. IHC staining pattern for Ezh2D/+ mice shown in Figure 1g appears more similar to levels of positive cells observed in Ezh2D/D mice. How representative is this data, also given the results shown in Figure S1c, where almost all tumor nodules observed in Ezh2D/+ mice appear to show expression? Have the authors demonstrated loss of only one Ezh2 allele in heterozygous tumours i.e. by recombination PCR?
 - f. when comparing EZH2+/+ KP mice to EZH+/- KP mice (Page 4)?
2. The authors utilised 2D and 3D culture assays as a tool to functionally evaluate the mechanisms underlying hetero- vs homozygous loss of Ezh2 on tumorigenesis. While the authors beautifully confirm the absence of Ezh2 protein expression in 2D and 3D-derived cultures (Figure 2b and Figure S2b) Additional quantification of Ezh2 levels should be included to support haplo-insufficiency of Ezh2 in D/+ tumor cells.
 - a. Quantification of protein levels in Western blots shown in Figure 2b and Figure S2b. and/OR
 - b. PCR to confirm recombination of the Ezh2 allele(s).This is also required as mRNA expression is most variable in Ezh2D/+ cells propagated in 3D cultures (Figure S2a).
3. Did the authors evaluate the expression of EMT markers in 2D culture assays, given the published work of Serresi et al. Cancer Cell 2016. Moreover, how does this compare with 3D cultures. IHC/IF staining using markers of EMT + polarity/tight junctions e.g. E-cadherin would provide evidence to support the statement "'3D tumor spheroids....pseudo stratification and apical-basal polarity of the epithelial layers" (pg. 5).

4. On the hypothesis that higher proliferation rates lead to the decoupling of EZH2 and H3K27Me3, the data in Figure 2c shows higher proliferation (measured by increased in S-phase) in 2D cultures compared to 3D cultures. Both immunoblots in Figure 2B (2D culture) and Figure S2B (3D culture) show low EZH2 correlating with low H3K27Me3 in EZH2 null cultures. Can the authors please explain how these results relate to your hypothesis of decoupling of EZH2 and H3K27Me3 through high proliferation rates?
5. Figure S3C, what are the units for Immune Cell Infiltration on the y-axis? The observation of low macrophage infiltration (even in the setting of Ezh2 proficiency) is baffling, given that it is well recognised in the literature that Kras-mutant GEMMs exhibit high level of macrophage infiltration (Ji et al. Oncogene 2005; Best et al. Nature Communications 2019, among others). Moreover, how correlative were the RNA-seq findings related to immune infiltration to IHC staining of CD8 T cells, B cells and macrophages?
6. Examination of FoxP3 protein expression on spontaneous GEMMs tumors vs 3D tumor spheroids is recommended.
7. It would be nice to confirm FOXP2 expression in human cell lines treated with EPZ6438 to complement Figure 4h. Does a negative normalized FOXP2 mRNA expression value still reflect protein abundance?
8. Could the authors comment on whether the band at 75 kDa or 100 kDa is specific to FOXP2 (Figure 4b), and why this double band signature is not seen in the empty vector human cell lines (Figure 5a). High EZH2 protein expression in 2D EZH2 WT cultures produces a band at 100 kDa (Figure 4b), while high EZH2 protein expression in HBEC did not have this 100 kDa band.
9. The authors demonstrate the FoxP3 is more highly expressed in Ezh2-null tumor cells. If the authors were interested in determining whether FoxP3 played a role in the metastatic phenotype observed in Ezh2-KO tumor cells, have the authors KD FoxP3 in the murine tumor cell lines / spheroids and evaluated whether FoxP3 KD augmented the migration observed in Figure 5c? That would be definitive evidence of a role of FoxP3 in this process.
10. Have the authors tried other first-line chemotherapy agents besides carboplatin (e.g. cisplatin, etoposide)? Using more than one agent would be more compelling to the authors' idea of poorly differentiated cells with high FOXP2 being more chemoresistant.
11. Can the authors comment on the 3d organoid sphere sizes? There appears to be variability in 3D organoid sizes across the three EZH2 genotypes in Figure S6. Could the size of the 3D organoid structure have a confounding effect on JQ1 / GSK-J4 efficacy due to the possibility of the lack of drug penetrance into the centre of bigger organoid structures.
12. Can the authors comment on the observed decrease in survival of 3D EZH2 +/- organoids upon treatment with JQ1 (Figure 6C)?
13. Statistics on GSK-J4 day 10 values of EZH2 null compared to EZH2 haplo-insufficient and EZH2 proficient genotypes needed (Figure 6D). Please specify what allograft tumour volume was reached before drug treatment commenced, and at what range the tumour volumes were.
14. If high proliferation is the cause of decoupling EZH2 and H3K27Me3, have the authors considered using small molecular inhibitors or knock-down experiments targeting cell cycle. This would be additional compelling data alongside supplementation with SAM.
15. In the patient IHC, decoupling EZH2 loss and H3K27Me3 is observed. However, in many in vitro and in vivo systems presented in this manuscript, we observe high EZH2 being associated with high

H3K27Me3. This decoupling hypothesis is only explored with one human cell line H460 (Figure 7F) and less compellingly with a EZH2 WT murine line (Figure S7D). This begs the question as to whether this particular murine GEMM model is an appropriate model for decoupling.

Minor Comments:

1. Page 3 – spell out in full: MEK acronym.
2. Related to Figure 1b – no scale bar is included. Please add.
3. Figure 1G column 2 – missing label for EZH2 $\Delta/+$
4. There is no panel Figure 3c and Figure 3d that relate to what is written in the Figure Legend.
5. Figure legend 5d and 5k are missing scale bar measurement in the Figure legend.
6. There are two figure legends for Figure S5c.
7. Measurement for scale bar in Figure S5c missing in figure legend.

Rebuttal for *Nature Communications* manuscript “Polycomb Deficiency Drives a FOXP2-high Aggressive State Targetable by Epigenetic Inhibitors” by Chen et al.

We thank the reviewers for their comprehensive review of our manuscript. We have taken significant time and effort to address the concerns raised. Below are each of the reviewer’s comments followed by our responses in blue.

Reviewers' comments:

Reviewer #1 (Remarks to the Author):

The manuscript by Chen et al entitled “Polycomb deficiency drives a FOXP2-high aggressive state targetable by epigenetic inhibitors” reported the characterization and functional investigation into heterozygous or homozygous deletion of *Ezh2* in a *Kras*-driven lung adenocarcinomas models. The authors observed that *Ezh2* haplo-insufficient tumors were less lethal and lower grade compared with *Ezh2* fully insufficient tumors. In addition, the authors explored the culture conditions on gene expression, which identified *Foxp2* as a target of EZH2 whose upregulation may account for certain aspects of observed phenotypes. Further, based on previous publications, the authors explored the possibilities of targeting EZH2 inhibited cells with BET or H3K27 demethylase inhibitors. Finally, the authors performed correlative studies in human lung cancer. While there are several interesting observations presented by the authors, the manuscript is fragmented without mechanistic understanding with various findings. In addition, there are many issues with interpretation of results. The contradiction between mouse models and human lung cancer further weakened the conclusion. Therefore, the manuscript is preliminary at this stage and pre-mature for publication.

1. The differences observed between heterozygous (oncogene) and homozygous (tumor suppressor) *Ezh2* knockout are interesting. However, the underlying mechanism is completely not clear. One would expect EZH2, H3K27me3 and SUZ12/EED ChIP-seq to dissect EZH2 methyltransferase dependent vs. independent function in these settings. Along these lines, a rescue experiment with wildtype or methyltransferase inactive mutant EZH2 is essential for dissecting the relevant mechanisms. For all the analysis, heterozygous cells should be included (instead, most of comparisons were done only between wildtype and homozygous knockout, which will not address the differences).

To address this comment, we performed ChIP-sequencing on the tumoroids that were *Ezh2* wild-type (WT), heterozygous, null or wild-type treated with EPZ-6438 EZH2 inhibitor. We immunoprecipitated EZH2 in the wild-type and heterozygous and H3K27me3 in all four tumoroids. These new data are presented in new **Figures 3, S3, 4 and S4**. These data demonstrate that the number of both total EZH2 peaks, and EZH2 peaks associated with genes in WT and heterozygous tumoroids were similar, but that more than half of the gene-associated peaks in the heterozygous tumoroids were unique. These gained EZH2 peaks were at genes enriched in pathways including DNA damage response, Cell Cycle and Stemness (**Fig. 3e**). Together with the cell cycle data and the lower tumor growth, these results suggest that *Ezh2* heterozygous cells have changes in cell cycle and stemness regulation that leads to more indolent tumor cells. We also found that about 131/411 of the genes significantly up-regulated in *Ezh2* null tumoroids also had H3K27me3 peaks called at FDR=1E-10 in *Ezh2* wild-type organoids and were reduced by 2-fold or more in the *Ezh2* null tumoroids. These included all of the genes up-regulated in both the 3D and sorted cells (**Fig. 3b+f**), of which *Foxp2* was the key gene that we studied further.

We also rescued an *Ezh2* null cell line with either wild-type or mutant (F667I) EZH2, which have been classified as methyltransferase active and inactive (L. Wang, Jin, Lee, Su, & Ge, 2010). We were able to observe rescue (reduction) of FOXP2 protein in the EZH2-WT transduced cells, and the levels of FOXP2 in the *Ezh2* null and EZH2-F667I transduced cells were not statistically different (**Fig. 4c+d**).

For full analysis of the *Ezh2* heterozygous state, we performed every experiment in all three *Ezh2* genotypes, with the exceptions of the migration assay in **Fig. S2a** and ChIP-PCR in **Fig. S4b**. We did include the *Ezh2* heterozygous in all RNA-seq, ChIP-seq, western blot, *in vitro* drug response and subcutaneous graft assays. We also now include GSEA analysis of heterozygous vs wild-type and heterozygous vs. null in the different models. This GSEA analysis shows that *Ezh2* heterozygous cells, despite having some loss of H3K27me3 enrichment at Polycomb targets, have the lowest expression of Polycomb target genes. Instead of a haplo-insufficiency, this suggests that the heterozygous state has more canonical PRC2 gene repression and that EZH2 is acting as a tumor suppressor in both *Ezh2* heterozygous and *Ezh2* null cells. Importantly, we also analyzed the immune microenvironment in the *Ezh2* heterozygous tumors that suggest the tumors contain more lymphoid cells and this could explain the lower tumor burden and longer survival when comparing the *Ezh2* null phenotypes (**Fig. S1f-k**).

We hope that the reviewer agrees that with the above-described additions, along with the identification of FOXP2 as a bona-fide PRC2 target gene in lung adenocarcinoma, the data showing that FOXP2 can increase stemness and migration, and the careful analysis of gene expression and drug responses in varied model systems, our work is now suitable for publication in *Nature Communications*.

2. There is a major disconnect between mouse models and human lung cancer data. Mouse models suggest that *Ezh2* loss upregulates FOXP2 through unknown (and likely methyltransferase activity because EZH2 inhibitor increases FOXP2 expression) to promote a more aggressive phenotype. However, the data as presented in Figure 7 do not support this conclusion. For example, EZH2 expression instead positively correlate with FOXP2 expression and the survival data support both EZH2 and FOXP2 as oncogenes. Instead, Figure 7 appears to focus on the discrepancy between EZH2 and H3K27me3 expression observed in the tumors (through investigating SAM availability).

We are using the EZH2 knock-out mouse as a paradigm for modeling PRC2 dysfunction. In patient samples and certain mouse models, PRC2 dysfunction, as measured through loss of H3K27me3, is often accompanied by an increase in EZH2 (Chen et al., 2013; Holm et al., 2012; Pellakuru et al., 2012; H. Zhang et al., 2017) (**Fig. 7a**). In that way, high EZH2 expression may be considered a biomarker of reduction in PRC2 activity in some settings. With the common phenotype being reduction in PRC2 activity, the mouse model and human data do support each other. In both the *Ezh2*-null organoids and the patient tissues, H3K27me3 is lower, but not completely absent. Also, in both systems lower levels of H3K27me3 are correlated with more poorly differentiated tumors. Because we do observe higher EZH2 in the H3K27me3-low human tumors, we reasoned that there must be another mechanism other than EZH2 abundance that regulates PRC2 activity. This is why we explored whether availability of SAM could in part explain the de-coupling of EZH2 expression from its canonical enzymatic activity. We have clarified these points in the final figure.

3. The comparison between 2D vs. 3D culture conditions is again interesting. However, the results as presented are descriptive with the exception of identification of FOXP2 as a potential EZH2 target gene. Please see specific comments below.

To address this comment, we compared 2D to 3D by western blotting for several cell cycle proteins, supporting the finding that cell cycle is one of the major changes between 2D and 3D cultures (**Fig. 2e+f**). In addition to profiling cell cycle, we also observed that *Hox* genes, which are thought of as canonical PRC2 targets, are not de-repressed by loss of *Ezh2* in the 3D cultures, but are in 2D (**Fig. 3a**). This is consistent with the ChIP-seq data that show a retention of H3K27me3 marks at the *Hox* gene clusters in *Ezh2* null 3D tumoroids (**Fig. 3d+S3e**). Finally, as the reviewer suggests, we identified *Foxp2* as a target of PRC2 that may control stemness programs and migration, and is highly up-regulated (6-100 fold) in 3D cultures relative to 2D cultures.

4. The use of JQ1 and GSK-J4 in the context of EZH2/H3K27me3 inhibition has been reported, which reduces the novelty of the relevant data. Although authors have a unique opportunity to dissect the mechanism using the mouse model generated, there is no attempt on this front (for example, to differentiate methyltransferase dependent vs. independent effects of EZH2 etc).

We present data in human lines that show that EPZ-6438 and JQ1 have synergism (**Fig. 6g**), and given the mechanism of action of EPZ-6438, this should be a methyltransferase dependent effect. We further add to the previous observations of JQ1 sensitivity in PRC2-deficient settings (Huang et al., 2018; Y. Zhang et al., 2017) by demonstrating that the model system used to test the drug efficacy dramatically effects the results. In fact, both human and murine cells grown in 3D cultures are more sensitive to JQ1 than cells grown in 2D cultures (**Fig. S6j, Fig 6b+c**).

It would be very interesting in future studies to compare rescued EZH2 to a methyltransferase dead version of EZH2 in the organoids and explore what functions EZH2 may have outside of methylation activity. However, despite observing rescue of *Foxp2* expression, we did not observe the expected changes in H3K27me3 by western blot (**Reviewer Fig. 1, Fig. 4c+d**), and we believe we would need to perfect this system before attempting to use it to answer the reviewer's question. We will likely write a future R01 on this topic and the relationship of PRC2 activity to FOXP2 in lung cells.

Reviewer Figure 1: Plot of western blot quantification from **Fig. 4d** without the FOXP2 to better visualize EZH2 and H3K27me changes, n=3 blots. Analysis by one way ANOVA with multiple comparisons.

5. Many statements in the manuscript are simply not supported by results, which include but not limit to those without statistical significance.

To address this comment, we have carefully worded the conclusions to focus on statistically significant results that were observed in several models.

Specific comments:

1. For all the migration assays, please normalize for cell growth rates that were clearly affected in 2D culture conditions where these assays were performed.

For all migration assays, including those in new Fig. **S2a**, **5d** and **5i**, migration is now normalized to the relative cell proliferation in 2D at the 24-hour time-point as suggested.

2. There is a disconnect between grade of tumors observed in *Ezh2* homozygous knockout mice and *Ezh2*'s potential role in differentiation status (e.g., Fig. S3D).

The *Ezh2* null tumors had higher nuclear grades, indicating more poorly differentiated tumors. In the original Fig. **S3d**, *Ezh2* null sorted tumor cells had lower enrichment in the lung cell differentiation signature, supporting the hypothesis that *Ezh2* null tumors more stem-like, or less differentiated. However, we agree that this GSEA result was somewhat confusing, and therefore, we removed it from the manuscript to make room for new data.

3. For FOXP2 westernblot, please clarify which band is FOXP2 in Fig. 4b and perhaps include FOXP2 knockdown cells as controls.

According to the AbCAM website, this antibody recognizes FOXP2 around 75 kDa, and we believed that the band around 110 kDa was a non-specific band. To address this issue, we used two different pSilencer siRNA constructs from SIGMA and ran western blotting to examine which band was affected by the siRNA. The results shown in new Fig. **S4a+b** clearly show that the lower band is FOXP2 and that the upper band is unchanged by siRNA and is therefore likely a non-specific band. We have now labeled it as such in the figures using this evidence.

4. *Ezh2* ^{-/-} increased FOXP2 expression in Fig. 4b. If upregulation of FOXP2 determines the sensitivity to JQ1 in *Ezh2* ^{-/-} cells, the enhanced effect of JQ1 should be observed in *Ezh2* ^{-/-} in Fig. 6b.

The qPCR does show an increase in *Foxp2* in *Ezh2* null 2D cell relative to wild-type, but the overall level is lower in 2D than in 3D null cells (Fig. **4a** p=0.02, Fig. **S6c** p=0.0028). The original western blotting did show FOXP2 expression in 2D null cells; however, the 2D and 3D western blotting had been done separately. A new western blot is now shown with 2D and 3D samples that were isolated and run together (Fig. **4b**). This blot shows expression of FOXP2 in both *Ezh2* null 3D samples, and in one *Ezh2* heterozygous 3D sample, but very low levels of FOXP2 in the 2D samples

5. The authors claimed that the reason why FOXP2 was not identified in 2D condition is because its expression is too low to generate enough reads (page 8, Fig. 3a). The result showed in Fig. 4b would argue against this conclusion if the band(s) are indeed FOXP2. A direct comparison of FOXP2 between 2D and 3D condition would be informative.

For 2D and 3D RNA-seq samples, we sequenced to a depth of 25 million reads. For the 2D samples, *Foxp2* had very low expression (counts per million (CPM) < 10) in all 7/7 *Ezh2* wild-type samples, in 5/6 *Ezh2* heterozygous samples, and in 3/7 *Ezh2* null samples. In our differential expression analysis, we used the EdgeR "filterByExpr" function to filter out genes with very low expressions using the default threshold of CPM > 10 in at least 6 samples. *Foxp2* was not included because its expression was below the default threshold

defined by EdgeR. This is consistent with relatively low levels of *Foxp2* in 2D vs 3D by RT-PCR (**Fig. 4a**), and with the new FOXP2 WB showing 2D and 3D organoids on the same blot (**Fig. 4b**). On this blot, FOXP2 did not appear higher in 2D null cells, further supporting that FOXP2 is a specific PRC2 target in 3D and *in vivo* and may determine JQ1 sensitivity.

6. Please also examine EZH2, H3K27me3 and FOXP2 in Fig. 4b for comparison between 2D and 3D culture conditions.

The western blot directly comparing 2D to 3D lysates in **Fig. 2d** and **Fig. 4b** show that both EZH2 and H3K27me3 are higher in 2D *Ezh2* wild-type cells, while FOXP2 expression is highest in 3D *Ezh2* null cells.

7. For all the ChIP analysis, please clarify how the normalization was performed (over input or IgG controls). In addition, IgG controls should be included throughout if this is not the case.

In the original manuscript, we described in the methods that the ChIP normalization was to a *Gapdh* locus similar to the way RT-qPCR calculations are done. An IgG-anti-GFP antibody was used as a negative pull-down control, but was not depicted in the graphs. In the new manuscript, we show the mouse ChIP samples with the H3K27me3 and GFP antibodies graphed as 'normalized abundance', or 2^{-ΔdCt} relative to the 10% input sample in **Fig. S4d**. In addition to the new ChIP-seq data, these data show a loss of H3K27me3 mark at the 5'TSS in the *Ezh2* null tumoroids and a significant loss of H3K27me3 mark at the ATG. We also repeated the human cell line ChIP with an IgG antibody that matches the H3K27me3 antibody (we had previously used the anti-GFP control), using both BEASKP cells and the H460 cells and four rather than three experimental replicates. We observed that H3K27me3 mark is significantly reduced in the EPZ6438 treated cells at the regulatory site 5' of the gene. By normalizing both the to 10% input and to the vehicle control H3K27me3 ChIP, it can be observed that H3K27me3 is significantly lost at the regulatory region in H460 cells, and at all three sites probed in the BEASKP cells. The new data are depicted in new **Fig. S4e+f**.

8. For Fig.4h, please clarify whether the experiments were done in 2D or 3D condition. In addition, please clarify whether there is a difference in regulating FOXP2 by EPZ-6438 between 2D and 3D conditions.

The original **Fig. 4h** was performed on 2D cells. In the revised manuscript, we directly compared 2D to 3D cells treated with EPZ-6438 for *FOXP2* expression for the both the H2030 and H460 human cell lines. We found that in both cell culture models, EPZ-6438 significantly increases *FOXP2* expression. However, the overall levels in 3D cultures are significantly higher than in 2D cultures. These data are in new **Fig. S4g**.

9. Likewise, the author showed EZH2 KD had distinct effect to JQ1 sensitivity between 2D and 3D in Fig. S6. It would be interesting to check whether EZH2 KD differentially affect FOXP2 expression between 2D and 3D conditions.

While we did not further explore the *shEZH2* in 2D and 3D conditions, we did do a very similar experiment using the 2D, 3D and *in vivo* murine cells. We treated *Ezh2* wild-type, heterozygous and null 2D cell lines and 3D tumoroids with 100nM JQ1 and measured *Foxp2* expression relative to the vehicle control treated cells. We observed that *Foxp2* was not changed by JQ1 in *Ezh2* wild-type cells, that *Foxp2* was significantly down-regulated by JQ1 in *Ezh2* heterozygous cells in 3D, and that *Foxp2* was significantly down-regulated in *Ezh2* null cells in 2D, 3D (**Fig. S6d-f**). *In vivo* grafts from *Ezh2* placebo and JQ1 treated mice had similar transcriptional changes. It is important to note that for the *Ezh2* null cells, the 3D cells had a larger decrease in *Foxp2* expression (LFC= -1.6) than in 2D cells (LFC= -0.4) and that the relative levels of *Foxp2* were much

higher in 3D than in 2D cells (LFC= 9.2). We believe that this difference in baseline *Foxp2* expression is one of the key differences of the models that determines their differential sensitivity to JQ1.

0. Fig. 5a, please clarify which band is FOXP2 band.

The band between 75 kDa and 100 kDa is the endogenous FOXP2. The band at 100 kDa is the overexpressed FOXP2, which is consistent with an over-expressed FOXP2 construct (Moparthi & Koch, 2020). The construct we used is 2148bp plus an HA tag, while the construct used in the cited reference was 2220 including a flag tag. Therefore, our version runs a bit lower on the gel as expected.

1. Fig. 5h-k: as shown in Fig. S7j, FOXP2 expression is relatively low in H460 cells. Please clarify the reason why the authors chose this cell line for FOXP2 knockdown (and, in particular, the authors categorize H460 cell line as JQ1 resistant, suggesting that FOXP2 level in this line is not sufficient to confer the phenotype).

We chose H460 because it is a KRAS+ robustly growing cell line, therefore in part mimicking the mouse model. Additionally, H460 can form round and homogeneous tumoroids in 3D culture, so it was feasible for us to measure the diameter changes. Other human lines formed irregular spheroids or branching structures that made measuring the diameters very difficult. In the revised manuscript, we repeated the *FOXP2* knock-down on other two FOXP2-high cell lines, A549 and H2009. *FOXP2* knock-down inhibited the cell growth in both lines and the new data are shown in **Fig. S5f-h**.

2. Fig. 7: it is interesting EZH2 and FOXP2 are positively correlated in patient samples, while the points of the mouse models were to demonstrate an upregulation of FOXP2 by *Ezh2* inhibition. Likewise, patients with low EZH2 expression have a better prognosis. This is in contrast with phenotype observed in *Ezh2* homozygous knockout mice.

Yes, the data do appear to be contrasting in the EZH2 levels. However, we are using the *Ezh2* knock-out mouse model as a way to model Polycomb dysfunction. In the patient samples, Polycomb dysfunction, as measured by a loss of H3K27me3, appears to be often accompanied by higher EZH2 levels (Chen et al., 2013; Holm et al., 2012; Pellakuru et al., 2012; H. Zhang et al., 2017). Given that EZH2 is rarely mutated in lung cancers, we reasoned there must be another mechanism through which patient samples become PRC2-deficient. We explored SAM availability as this mechanism, and we expand on this idea in revised **Fig. 7**.

3. The rationale for including only ADC and poorly differentiated tumors in the analysis in Fig. 7 is not clear.

We chose to focus on these tumor types since the KRAS/*p53-null* mouse model is known to model lung adenocarcinoma, and with *Ezh2* deletion, we produced more poorly differentiated tumors. Squamous and adeno-squamous tumors are distinct in Polycomb levels, and we believe including them would confound the results. We have observed analyses in the literature where differences in EZH2 and H3K27me3 could be almost completely explained by the inherent differences between adenocarcinoma and squamous cell carcinoma and not other factors that they were attributed to, such as smoking. In our own dataset, we now show that EZH2 IHC does correlate with poor survival, but also that EZH2 is statistically higher in squamous tumors than in adenocarcinomas, and that patients with squamous tumors in this cohort had a significantly lower survival (**Fig. 7a, S7b+c**).

14. Please include all the cell lines as shown in Fig. 5a for Fig. 7i.

We show here in **Reviewer Fig. 2** the results for all cell lines in original **Fig. 5a** for JQ1 sensitivity. Given that only H460 had a robust change, we have edited this part.

Reviewer Figure 2: Results from 4 cell lines over-expression FOXP2 for changes in JQ1 sensitivity. The cell lines and n are indicated. For HBEC3KT, one run was the opposite of the other two and we cannot be sure if this was a real result or human error. In the revised manuscript, we show only the H460 data and have re-worded the conclusions.

15. Some of the data as presented are marginal at the best (despite the indicated significance, e.g., Fig. 5f and 7i)

We have removed original **Fig. 7i** and re-phrased this section. For the **Fig. 5f**, we repeated this experiment with the chemotherapy drug etoposide. Both carboplatin and etoposide had significantly higher IC₅₀ values when FOXP2 was over-expressed, and although the effect size was relatively modest (1.6-1.8 fold change), we believe these changes to be highly reproducible.

16. Figure S5f and S5h, the knockdown of FOXP2 does not correlate with spheroid numbers. Also, the knockdown efficiency was quite low.

FOXP2 knock-down correlated with reduced spheroid numbers in *shFOXP2.1* but not *shFOXP2.2*. We did not include these data in the revised manuscript due to the lack of significance with the second hairpin and space considerations. We believe the knock-down efficiency is low because *FOXP2* knock-down strongly affected the cell growth, and we observed that after 2 passages, FOXP2 returned to basal levels and growth reduction was no longer evident. This phenotype suggests that FOXP2 plays an important role in oncogenic property of lung cancer cells, even when it is expressed at low levels. However, we believe that it is higher expression of FOXP2, likely through promoter acetylation, that may make KRAS+ lung cancer cells more susceptible to JQ1 treatment.

17. Results showed in Fig. S6b does not match those in Fig. 6d. Contrary to the statement in results section, there is no difference in Foxp2 expression among the three conditions in 100 nM JQ1 treatment in Fig. S6h. It is surprising that EZH2 knockdown alone did not affect cell growth in Fig. S6.

For the original **Fig. S6b**, the data were averaged for all the donor mice for that genotype, which differed from **Fig. 6d** where every donor mouse was plotted relative to its own placebo control. Therefore, we replotted the tumor weights relative to the placebo controls for each donor mouse and then averaged those changes. The new visualizations of the data are in **Fig. S6b**.

With the JQ1 treated cells, *Foxp2* was significantly decreased in the *Ezh2* heterozygous and null 3D tumoroids, but not in *Ezh2* wild-type tumoroids (**Fig. S6e**). Similarly, in the H2030 2D cells, JQ1 treatment significantly

decreased *FOXP2* expression only in *shEZH2* cells but not in the *shGFP* cells (**Fig. S6i**). The *shEZH2* cell lines in **Fig. S6** were normalized to their vehicle controls, not to the *shGFP* line. If we normalize to the *shGFP* line, we do observe that *shEZH2.2* reduced cell growth in the H2030 cell line, but not in H460 cells. However, this was not a traditional multi time-point growth assay, so solid conclusions cannot be made.

0. Statements related to Fig. S7e and 7h are not accurate. For Fig. S7i-j, please specify the experiment was performed in 2D or 3D condition.

We have revised this section to focus only on the significantly different relapse-free survival of lung adenocarcinoma patients with high and low *FOXP2* expression. The original **Fig. S7i+j** were done in 2D condition, but this data has now been removed from the manuscript.

1. Please include statistical analysis in all the analysis where applicable.

We have performed statistics with our collaborating statistician Dr. Rani Jayswal for every graph. If statistics are not shown, it is because the appropriate test did not reach $p < 0.05$ for any comparison. For some analyses where differences are apparent but not statistically significant, we added "n.s." or the p value to the figure to make it clear that statistical analysis was performed.

Reviewer #2 (Remarks to the Author):

This manuscript describes the phenotypes of EZH2 heterozygous and homozygous deletions in non-small cell lung models, finding divergent effects across genotypes as well as differences across model systems. The demonstration that loss of one copy of EZH2 may be tumor suppressive while loss of both copies may be tumor promoting and increase aggressive phenotypes is of general interest. The authors take both a genetic and a pharmacological approach which make the study comprehensive and their conclusions generally accurate and relevant. However, the main message of the study is not presented clearly since although the divergent phenotypes of the EZH2 heterozygous vs homozygous gets lost in the multiple comparisons of 2D vs 3D vs in vivo models, which in most cases only investigate the wt vs homozygous deletion models. Within this context, it is unclear how authors rationalize the use of EZH2 inhibitors since this would imply a functional ‘total ablation’ of EZH2 function, similar to the homozygous deletion, which has a pro-cancer aggressive phenotype in their models. In addition, there are multiple instances of “data not shown” and even an instance of “data to be published in a separate manuscript” which are not acceptable if the missing data is to be considered toward the conclusions of the present work. Also, statistically non-significant data is mentioned in multiple cases as following a ‘trend’ and used to drive conclusions. Although some of these conclusions have other data to back it up, others do not and need to be addressed or conclusions altered. All these general concerns should be addressed. In addition, in more specific terms authors should comment on/address the following points:

We thank the reviewer for this constructive feedback. We have extensively edited the manuscript based on these comments. In the revised manuscript, we performed every experiment in all three *Ezh2* genotypes, with the exceptions of the migration assay in **Fig. S2a** and ChIP-PCR in **Fig. S4b**. We did include the *Ezh2* heterozygous in all RNA-seq, ChIP-seq, western blot, *in vitro* drug response and subcutaneous graft assays. We also now include GSEA analysis of heterozygous vs wild-type and heterozygous vs. null in the different models. This GSEA analysis shows that *Ezh2* heterozygous cells, despite having some loss the H3K27me3 enrichment at Polycomb targets, have the lowest expression of Polycomb target genes. Instead of a haplo-insufficiency, this suggests that the heterozygous state has more canonical PRC2 gene repression and that EZH2 is acting as a tumor suppressor in both *Ezh2* heterozygous and *Ezh2* null cells. Importantly, we also analyzed the immune microenvironment in the *Ezh2* heterozygous tumors that suggest the tumors contain more lymphoid cells and this could explain the lower tumor burden and longer survival when comparing the *Ezh2* null phenotypes (**Fig. S1f-k**).

We also made changes to ensure that all data are fully represented and that statistical analyses are indicated for every apparent difference. We agree that EZH2 inhibitors could in theory drive more aggressive phenotypes, particularly for KRAS-driven tumors. However, the use of the inhibitors could also modulate the immune microenvironment in ways not demonstrated by our mouse model, and could also make cells extremely sensitive to second agents. Therefore, we believe that finding the correct contexts for EZH2 inhibition is key to being able to leverage this FDA approved therapy moving forward. We hope that you agree that our data adds some important points to this discussion and warrants publication in *Nature Communications*.

1. What is the direct evidence of cell cycle ‘arrest’ in the models presented in Fig 2 and Supp Fig 2 vs a difference in cell cycle distribution/timing of cell division?

We rephrased the conclusion as differences in cell cycle ‘distribution’ in replacement of ‘arrest’.

2. It is concerning that only one gene was shared across the three models systems in the EZH2 deleted models of Fig 3/Supp Fig 3, EZH2 itself. Can authors address this at some level?

To address this point, we asked our bioinformatics team if there was any way to better normalize the data to uncover more differentially expressed genes shared between the models. Their answer was that there are substantial differences between the 2D, 3D and *in vivo* models that are likely the reason for few overlapping differentially expressed genes. These differences can be visualized in **Fig. 2b** and **Fig. S2e+f**. Given that drug sensitivities were similar between *in vivo* and 3D systems, we feel that using the method of overlapping differentially expressed genes in the two systems helped us to narrow down to the genes driving the response. Most other published studies of genotype comparisons use only 2D cultures for gene expression assays. If we had followed that approach, we would have more significantly up- and down-regulated genes, but many of them are 2D specific, including *Hox* genes, EMT genes, and some other 'canonical' Polycomb targets. Our approach helped to uncover *Foxp2* as a novel target of PRC2 repression in lung adenocarcinoma, and also highlights the differences in model systems that should be considered when examining targets of epigenetic complexes.

3. Of the eleven genes shared by the 3D and the sorted tumors, what is the rationale for focusing mainly on FOXP2 other than the stated observations? The other 10 genes may have functional consequences as well. Please address.

Among the 11 genes, 7 of them are upregulated and 4 genes are downregulated. We focused on the upregulated genes since they might be directly repressed by PRC2. FOXP2 is a transcription factor and has been shown very important in lung development (Shu et al., 2007; Shu, Yang, Zhang, Lu, & Morrisey, 2001) and that is why we examined its role more closely. In the developing lung, it is suggested that FOXP2 could be upstream of *Mycn* expression (Shu et al., 2007). *Ndr4* is another interesting gene that we believed could be downstream of FOXP2 and may be part of a 'stemness program' (Kotipatruni et al., 2012). In support of this idea, we demonstrate in **Fig. S5d+e** that *MYCN* and *NDRG4* could be upregulated by FOXP2 overexpression in the presence of EZH2 inhibitor in the 3D H460 and H2030 cells. *Sftmbt2* is also a very interesting gene target given that it is a Polycomb gene, but it is also imprinted in the mouse (Q. Wang et al., 2011), and we have previously identified imprinted genes as a major group of de-repressed genes upon Polycomb ablation (Zacharek et al., 2010). *Sftmbt2* is not imprinted in the human, so this result may be mouse specific. *Arc* is a cytoskeletal gene important in neuronal cells (Bramham et al., 2010), *Susd2* is expressed in myofibroblasts (Schmuck et al., 2021), and *Krt79* marks a migratory population of cells during ear development (Veniaminova et al., 2013), suggesting these genes may contribute to the migration phenotype of *Ezh2* null cells. We added discussion of these genes to the text. In summary, FOXP2 appeared to be the most likely candidate for driving phenotypes observed in the *Ezh2* null cells. FOXP2 had also not been explored previously in lung cancer, possibly due to its low expression in 2D cells, and also the lack of a probe on the U133A microarrays that dominated the patient sample landscape prior to RNA-seq advances around 2015.

4. The differences seen in immune cell infiltration and signatures seem key. Can authors expand on the differences across all three phenotypes seen *in vivo* and in 3D models? This seems important for really understanding the tumor suppressive vs promoting effects of the heterozygous vs homozygous tumors, respectively. More of this data could perhaps be included in main Fig 3 rather than supplemental. Similarly, the finding that heterozygous tumors were enriched in histone methylation signatures should be functionally

explored at some level to make a connection to the phenotype of these tumors vs the homozygous deleted tumors.

Absolutely, we concur that the immune cell differences were likely the key to the *in vivo* differences observed in Fig. 1. To further explore this, we performed three analyses. The first was to use the bulk tumor RNA-seq data and the TIMER2.0 algorithm to estimate immune cell infiltration. Please note that we had originally run each set of tumors (*Ezh2* wild-type, heterozygous and null) separately in order to be able to see the graphical visualizations because with $n > 10$, the visualizations are no longer shown. However, we recently re-ran the analysis with all 17 total tumor samples together. We observed, unexpectedly, that the numbers changed, suggesting that there is a normalization inside the TIMER algorithm that we did not previously appreciate. The new data show that only monocytes and CD8 T cells appear different among the genotypes. Next, we confirmed these differences using histological cross-sections of tumors (all tumors from 8 animals per genotype), and performed machine learning to profile the nuclear phenotypes of the cells. The third approach was to use immuno-histochemistry for CD3 that our core facility had previously validated to work very well on murine samples. All of these assays support that *Ezh2* heterozygous tumors have more lymphocytes than *Ezh2* wild-type and null tumors. These results are shown in **Fig. S1**. As for the mechanisms across the systems, please see the next question.

5. What is the mechanism behind the immune repertoire differences observed? Is this transcriptionally driven by EZH2/PRC2 or non-transcriptional? Can any further insights be added?

Given the low number of significantly differentially expressed genes between the *Ezh2* heterozygous and the other genotypes, we explored this idea with Gene Set Enrichment Analysis. As we discussed above, the *Ezh2* heterozygous tumors were enriched for lymphoid signatures, while the *Ezh2* wild-type and null tumors were enriched for myeloid and immunosuppressive signatures. In 3D tumoroid culture however, *Ezh2* heterozygous cells had more active interferon response pathways than *Ezh2* null tumoroids, suggesting that they may have a hyper-activation of viral mimicry and STING signaling that could also result in these cells being more likely to be targeted by T cells.

6. Have authors truly established that FOXP2 is a “direct target” of EZH2/PRC2? The data presented is consistent with this possibility but no PRC2 component has been ChIPed on FOXP2 gene regulatory regions as far as I could tell.

Yes, the ChIP-sequencing that we did on EZH2 in the *Ezh2* wild-type and heterozygous tumors did call a peak for EZH2 in the *Ezh2* wild-type cells at FDR=0.018. Please see the ChIP peaks in **Fig. S4c**.

7. What are the effects of FOXP2 k/o or o/e in the heterozygous setting?

We did attempt to knock-down *Foxp2* in the null and heterozygous organoids, but we failed at achieving suitable puromycin selection. We also reasoned the siRNAs we had would be too transient to effectively test for inhibition of growth. We did see de-repression of FOXP2 in both 2D and 3D cultures in the heterozygous cells, though the expression was lower than in the 3D *Ezh2* null cells. Our current hypothesis is that FOXP2 may have different functions when PRC2 activity is at normal homeostatic levels than in the PRC2-deficient setting. This idea is supported by the data in **Fig. S5d+e** showing that although *NDRG4* and *MYCN* increase with FOXP2 over-expression, they are the most highly expressed with EPZ-6438 inhibitor. However, given the complexity of this question, we are planning a new set of experiments in an R01 grant to address the interactions between FOXP2 and H3K27me3.

8. Fig 7 and Supp Fig 7 appear weak as correlations are not robust and some are not even significant. Authors should carefully review all this data and only present conclusions that are fully warranted and significant. In addition, how each group was defined (ie, well, moderately or poorly differentiated or expression quartiles) should be more clearly defined in the fig and legends. Finally, it is hard to really related the genotypes studied in mice with the levels of expression in human tumors. Again, a more careful and conservative wording of conclusions is warranted in this regard.

For the revised manuscript, we did not use the binning approach for the patient tumors, and instead only show the average percent of positive nuclei for each tumor differentiation state (**Fig. 7b**). These differentiation states were defined by a board-certified pathologist and are part of the meta-data given to us by the Markey Cancer Center Biospecimens SRF with the tissue micro-array. In addition, we have carefully worded the conclusions to focus on only significant results.

More minor points:

1. EZH2 levels in the heterozygous deleted tumors appear lower than expected (Fig 1/Suppl Fig 1). Have Western blots been done on tumor tissue? Any comments?

The image used for original **Fig. 1** had lower *Ezh2* levels, but the overall expression is very heterogeneous. An image from the same area of the same tumor, showing more EZH2 positive nuclei, has now been placed there. According to the software assisted quantifications, EZH2 expression in *Ezh2* heterozygous tumors was significantly lower than in the wild-type tumors, but ranged significantly (**Fig. S1b**). Western blots were not done on tumors since the tissues were used for sorting and histology and we did not have additional frozen samples for protein isolation.

2. How authors define “more poorly differentiated” tumors in both animal models and human data should be explained in more detail

The differentiation state of each tumor was defined by the pathologists from the Markey Cancer Center. In the mouse models, the *Ezh2* null tumors had higher nuclear grades, which we believe is synonymous with more poorly differentiated tumors (Jackson et al., 2005). It is consistent with the patient samples that poorly differentiated tumors had lower H3K27me3.

3. Throughout the manuscript please add “n.s” for all non significant statistical differences and alter text so as not to include small non-significant small changes as ‘trends’ supporting any conclusion.

We have edited the manuscript accordingly. For analyses where differences are apparent but not statistically significant, we added “n.s.” or the p value to the figure or the legend.

4. Authors may consider using more obvious differences in color schemes to distinguish wt vs heterozygous models throughout the figures. The two gray scales used are too close.

Thank you for your comment. We changed the heterozygous to a darker color and we hope that this is now clearer.

5. The SAM experiment is speculative. Has it been repeated and confirmed in some other independent way?

We have now performed the same experiment in three different human cell lines and in mouse *KRAS/p53-null* cells. These data are now shown in new **Fig. 7f-h + S7d+e**. In addition, we would like to mention that there are mice that have a deficiency in the enzyme cystathionine beta synthase (CBS), and have a marked accumulation of SAM. However, the inventors of the mice proposed that because SAH levels were also higher, methylation activities may decrease. They instead observed no differences in global methylation and what appeared to be a slight increase in H3K27me3 by ELISA (Lee et al., 2017). How CBS, SAM, and SAH work together to influence methyltransferase function is a current research area in our laboratory. We have observed that over-expression of CBS in HBECs reduces H3K27me3, but we need to confirm that SAM levels have changed.

Reviewer #3 (Remarks to the Author):

The authors investigated the role of haplo- and full-insufficiency of EZH2 in a GEMM of lung adenocarcinoma (KRASG12D/+, Tp53-/-; herein referred to KP mice). EZH2 is the functional enzymatic component of the PRC2 complex. The PRC2 complex is a histone methyltransferase particularly at H3K27 to end up with tri-methylation of H3K27 (H3K27Me3), which acts as a gene silencer. Mutant EZH2 can render PRC2 inactive and lead to acetylation of H3K27, which activates gene transcription. While it would make sense if increased EZH2 expression leads to increased H3K27 methylation, it has been observed to have a reciprocal relationship in this manuscript and in the literature cited. Thus, the rationale for this study ultimately investigates the cellular conditions as to whether EZH2 acts as a tumour suppressor or oncogene. The authors found significant differences in culture methods, where 3D organoid culture, but not 2D, recapitulated gene expression patterns found in in vivo tumours better.

FOXP2 was identified as a direct target to de-repression by non-functional PRC2, assessed by ChIP-qPCR. High FOXP2 expression was seen with EZH2 null murine cancer cell lines, which showed “stemness” and capacity to migrate. These high FOXP2 cells were also resistant to carboplatin treatment. Inhibitors against histone demethylase (GSK-J4) and BET inhibitor (JQ1) showed the best efficacy in EZH2 null murine cancer cell lines, with JQ1 being the best at attenuating allograft tumours. In the clinic, a decoupling of EZH2 and H3K27Me3 levels are seen. The authors hypothesise that EZH2 proficient cells are highly proliferative non-differentiated cells, which contribute to low H3K27Me3 because of the non-abundance of SAM, a substrate for tri-methylation of H3K27. Supplementation of SAM to a human cell line reversed this decoupling, however, this effect was less evident in EZH2 proficient murine cell lines. Overall, the authors present a very interesting manuscript with different technical and theoretical ideas, however, some points need to be elaborated further.

We thank the reviewer for the constructive feedback. As you will see in the revised manuscript and rebuttal, we have attempted to address all of your questions. We hope you agree that the additional data and improved descriptions of the rationales and conclusions have strengthened the manuscript and make it suitable for publication in *Nature Communications*.

Major comments:

1. The authors conclude that EZH2 acts as an oncogene based on analysis of KP mice proficient for *Ezh2* or harboring loss of either one (D/+) of two (D/D) *Ezh2* conditional alleles (Figure 1). Comparisons of survival, tumor burden and tumor grade, should be examined amongst all genotypes (+/+ vs D/+ vs D/D) for a fair conclusion to be drawn to support this statement. Specifically:

a. Is the survival of *Ezh2*+/+ mice vs *Ezh2* D/+ mice significant?

Yes, the p-value of mice survival was $p=0.0376$ *Ezh2* wild-type vs *Ezh2* heterozygous, $p=0.0397$ *Ezh2* wild-type vs *Ezh2* null, and $p=0.0003$ *Ezh2* heterozygous vs *Ezh2* null by Mantel-Cox log-rank test. Please note that we updated this data due to a mistake in using Days360 rather than Days to calculate the graphs, although the data in SPSS were correct. We also reviewed inclusion criteria and found errors that were corrected prior to knowing how these corrections would influence the p values. The updates change the graph and p values slightly from the original submission, but the conclusions remain the same.

b. In Figure 1e, the nuclear grade of *Ezh2*^{+/+} and *Ezh2*^{D/+} mice are largely comparable, albeit, Grade 4 lesions appear to be decreased in *Ezh2*^{D/+} mice, but comparable between *Ezh2*^{+/+} and *Ezh2*^{D/D} mice. Statistics for all genotype comparisons should be performed.

With our collaborating statistician Rani Jayswal, we did comparisons between each set of genotypes for each nuclear grade using ANOVA with pairwise comparisons. The comparisons that starred in the figure are the statistically significant comparisons. Other comparisons were not statistically significant and we believed it would detract from the figure to indicate all the non-significant comparisons. In the revised version, we have indicated the p value (0.051) for the grade 3 *Ezh2* null vs *Ezh2* heterozygous and n.s. for other non-significant comparisons where differences were apparent but not statistically significant.

c. The time-point analysis of tumor burden, as shown in Figure S1, is a more accurate representation of tumor burden at defined time-points, than what is shown in Figure 1c, which appears to be a combination of all time-points. The increased burden seen in *Ezh2*^{D/D} mice and increased metastasis is consistent with *Ezh2* acting as a tumor suppressor gene, rather than an oncogene.

We have now switched the figures so that the binned time-point analysis is in the main figure (**Fig. 2c**) and the overall analysis is in the supplemental figure (**Fig. S2a**). Absolutely, we agree with your assessment that in the KRAS/p53 model, full deletion of *Ezh2* uncovers a role of EZH2 as a tumor suppressor, and we revised the text on page 4 to suggest this.

d. Related to the above point, in Figure S1, the authors demonstrate that *Ezh2* expression is still observed in some *Ezh2*^{D/+} and *Ezh2*^{D/D} tumours. Are these “positive” tumours also included in the quantification of tumor burden (particularly important in the context of *Ezh2*^{D/D} mice).

We did not intend to distract from the data with this point. The *Ezh2* heterozygous tumors do have EZH2 expression as expected and the quantification can be seen in **Fig. S1B**. The EZH2 positive areas in the *Ezh2*^{-fl/fl} tumors could be found in 21/27 (78%) of samples. However, the areas were small, and accounted for an

average of only 2.75% of the tumor area.

Often, it appeared that EZH2-positive areas may be normal epithelium that has been incorporated into the tumor, as can be seen in **Reviewer Fig. 3**. However, we agree that these areas may artificially inflate the tumor burden measurements and so for each *Ezh2* null tumor slide that had

Reviewer Figure 3: Example of KRAS/*Trp-53* null, *Ezh2* *fl/fl* tumor stained with EZH2 immunohistochemistry. Scale bars indicated on images. Arrows point to areas that may be EZH2 positive due to normal bronchiolar cell hyperplasia in heavily tumor burdened areas. These EZH2 positive areas were removed from the tumor burden analysis.

any EZH2-positive areas, we repeated the tumor burden analysis with the EZH2 IHC image instead of the H&E, and removed the EZH2-positive area from the total tumor burden. The updated data are shown in **Fig. 1c** and **S1a**. We also removed one lymph node metastasis in an *Ezh2-fl/fl* mouse that appeared to retain some EZH2 from the analysis. We replaced the images in **Fig. S1c** with more representative images. We explain in the methods that there were some EZH2-positive areas so that others can be aware of the potential for floxed systems to not be 100% efficient. Despite that caveat, we did verify full excision of the floxed alleles in our 3D tumoroids by both end-point PCR and RNA-seq (**Fig. S2c+d**).

e. IHC staining pattern for *Ezh2*^{D/+} mice shown in Figure 1g appears more similar to levels of positive cells observed in *Ezh2*^{D/D} mice. How representative is this data, also given the results shown in Figure S1c, where almost all tumor nodules observed in *Ezh2*^{D/+} mice appear to show expression? Have the authors demonstrated loss of only one *Ezh2* allele in heterozygous tumours i.e. by recombination PCR?

The tumors in *Ezh2* heterozygous mice do show expression, often towards the edges of the tumor (**Fig. S1c**). The panel in the main figure was randomly selected from a larger tumor section and had fewer EZH2 positive cells. We have replaced this panel with an image from the same mouse that shows a region with slightly higher EZH2 and it is more consistent with the overall intermediate level of EZH2 in these tumors shown in the quantification in **Figure S1b**.

We also demonstrated loss of *Ezh2* one allele by recombination PCR in 3D organoids in the new **Fig. S2d**. In this figure, the heterozygous cells have one wild-type allele and the floxed allele appears fully excised in each sample. Given that we do not expose the cultures to Cre after tumor cell isolation, these results imply that excision of the floxed alleles was complete in the majority of both the *Ezh2* heterozygous and null tumor cells *in vivo*.

f. when comparing EZH2^{+/+} KP mice to EZH2^{+/-} KP mice (Page 4)?

There are significant differences in overall tumor burden (**Fig. S1a**), survival (**Fig. 1d**), EZH2 levels (**Fig. S1b**), and immune cell populations (**Fig. S1f-k**) between the *Ezh2* wild-type and *Ezh2* heterozygous mice. We highlight these differences in the text and figures.

2. The authors utilised 2D and 3D culture assays as a tool to functionally evaluate the mechanisms underlying hetero- vs homozygous loss of *Ezh2* on tumorigenesis. While the authors beautifully confirm the absence of *Ezh2* protein expression in 2D and 3D-derived cultures (Figure 2b and Figure S2b) Additional quantification of *Ezh2* levels should be included to support haplo-insufficiency of *Ezh2* in D/+ tumor cells.

a. Quantification of protein levels in Western blots shown in Figure 2b and Figure S2b. and/OR

We have now used ImageJ software to quantify western blotting results. These are shown in **Fig. 2e, 4f+h, S4g, S5g, 7h, S7d+e**. On the majority of samples that had more than two replicates, we performed statistical analysis.

b. PCR to confirm recombination of the *Ezh2* allele(s).

This is also required as mRNA expression is most variable in *Ezh2*^{D/+} cells propagated in 3D cultures (Figure S2a).

We also demonstrated loss of *Ezh2* one allele by recombination PCR in 3D organoids in new **Fig. S2d**. In this figure, the heterozygous cells have one wild-type allele and the floxed allele appears fully excised in each

sample. Given that we do not expose the cultures to Cre after tumor cell isolation, these results imply that excision of the floxed alleles was complete in the majority of both the *Ezh2* heterozygous and null tumor cells *in vivo*.

3. Did the authors evaluate the expression of EMT markers in 2D culture assays, given the published work of Serresi et al. Cancer Cell 2016. Moreover, how does this compare with 3D cultures. IHC/IF staining using markers of EMT + polarity/tight junctions e.g. E-cadherin would provide evidence to support the statement “3D tumor spheroids...pseudo stratification and apical-basal polarity of the epithelial layers” (pg. 5).

We used Gene Set Enrichment Analysis on the RNA-seq data of 2D and 3D cells and showed that 2D *Ezh2* null cells had higher EMT signatures than *Ezh2* wild-type, consistent with previous reports on 2D cultures (Serresi et al., 2016). However, using the same signatures, 3D *Ezh2* null tumoroids had lower EMT signatures than the *Ezh2* wild-type tumoroids (**Fig. 3a**). The E-cadherin staining we attempted was not convincing, and other than highlighting the differences between 2D and 3D transcriptional signatures, we did not intend this to be a major point. Therefore, we revised the wording to not mention any apical-basal polarity in the 3D tumoroid cultures.

4. On the hypothesis that higher proliferation rates lead to the decoupling of EZH2 and H3K27Me3, the data in Figure 2c shows higher proliferation (measured by increased in S-phase) in 2D cultures compared to 3D cultures. Both immunoblots in Figure 2B (2D culture) and Figure S2B (3D culture) show low EZH2 correlating with low H3K27Me3 in EZH2 null cultures. Can the authors please explain how these results relate to your hypothesis of decoupling of EZH2 and H3K27Me3 through high proliferation rates?

Yes, we agree that this point can use clarification. We are using the EZH2 knock-out mouse model as a paradigm for modeling PRC2 dysfunction, where we are causing loss of PRC2 activity by genetic manipulation. In that case, yes, EZH2 levels do correlate with PRC2 levels. However, in patient samples and certain mouse models, PRC2 dysfunction, as measured through loss of H3K27me3, is often accompanied by an increase in EZH2 (Chen et al., 2013; Holm et al., 2012; Pellakuru et al., 2012; H. Zhang et al., 2017) (**Fig. 7a**). In these *in vivo* settings, EZH2 is not mutated, and so we reasoned that there must be another reason why high EZH2 does not equal high PRC2 activity. Therefore, we explore SAM availability as one of the possible mechanisms for EZH2/PRC2 de-coupling. We tested this idea in cell lines, but we also plan to further explore this idea in mouse models.

5. Figure S3C, what are the units for Immune Cell Infiltration on the y-axis? The observation of low macrophage infiltration (even in the setting of *Ezh2* proficiency) is baffling, given that it is well recognised in the literature that Kras-mutant GEMMs exhibit high level of macrophage infiltration (Ji et al. Oncogene 2005; Best et al. Nature Communications 2019, among others). Moreover, how correlative were the RNA-seq findings related to immune infiltration to IHC staining of CD8 T cells, B cells and macrophages?

The TIMER2.0 data shown were almost all from CIBERSORT(Newman et al., 2015), which reports “Immune cell fractions, relative to total immune cell content” (Sturm et al., 2019). The original macrophage result was only for M1 macrophages, which is why it was so low. Indeed, the predominate immune cell type in these tumors is macrophages, as we demonstrate by machine learning nuclear phenotyping (**Fig. S1h-i**), and by grouping the results of TIMER2.0/CIBERSORT based on cell type (**Reviewer Figure 4**). It is important to note that CIBERSORT classifies the major immune cell as ‘dendritic’, but that given the expression of *Cd11c* by alveolar macrophages, the ‘dendritic’ cells are likely the macrophages.

Reviewer Figure 4: CIBERSORT results from TIMER2.0 analysis demonstrate that monocytic myeloid (Myeloid M) cells predominate these tumors, as expected. Myeloid G refers to granulocytic myeloid cells.

We had originally run each set of tumors (*Ezh2* wild-type, heterozygous and null) separately in order to be able to see the graphical visualizations, because with $n > 10$, the visualizations are no longer shown. However, we recently re-ran the analysis with all 17 total tumor samples together. We observed, unexpectedly, that the numbers changed, suggesting that there is a normalization inside the algorithm that we did not previously appreciate. The new data show that only monocytes and CD8 T

cells appear different among the genotypes. We were able to confirm that CD3 T cells are more predominant in *Ezh2* heterozygous tumor by immunohistochemistry (**Fig. S1j-k**), and that *Ezh2* heterozygous tumors had enrichment of lymphoid signatures instead of suppressive myeloid signatures (**Fig. S1g**). Together, these results suggest an immune landscape difference with the *Ezh2* heterozygous tumors that could explain the lower tumor burden observed.

5. Examination of FoxP3 protein expression on spontaneous GEMMs tumors vs 3D tumor spheroids is recommended.

For this point, we performed immunohistochemistry for FOXP2 in the murine tumor samples. This analysis showed that similar to the 3D tumoroids, FOXP2 is significantly higher in only the *Ezh2* null tumors (**Fig. 4e**). This matches the western blot result with 2D and 3D cells in **Fig. 4b**. Given that we did not have banked frozen tumor samples or more mice, it was not possible for us to do a side-by-side comparison of FOXP2 levels in tumors and tumoroids.

6. It would be nice to confirm FOXP2 expression in human cell lines treated with EPZ6438 to complement Figure 4h. Does a negative normalized FOXP2 mRNA expression value still reflect protein abundance?

To address this question, we performed western blotting on EPZ6438 treated human cell lines using two different antibodies (SIGMA and CST). These results are shown and quantified in **Fig. 4e+f** and demonstrate a robust increase in FOXP2 protein levels in 4/5 tumor cell lines tested. The RT-qPCR (now **Fig. 4d**) was plotted as relative to the average of all cell lines and treatments, so yes, a negative is indicative of very low expression. Please note there was an error in the original RT-qPCR figure where some of the FOXP2 data were normalized to the *GAPDH* reference value, rather than to the FOXP2 reference value. This update did not effect the relative changes between vehicle and EPZ6438 or the statistics, but does change the appearance of the graph.

7. Could the authors comment on whether the band at 75 kDa or 100 kDa is specific to FOXP2 (Figure 4b), and why this double band signature is not seen in the empty vector human cell lines (Figure 5a). High EZH2 protein expression in 2D EZH2 WT cultures produces a band at 100 kDa (Figure 4b), while high EZH2 protein expression in HBEC did not have this 100 kDa band.

The band between 75 kDa and 100 kDa is the endogenous FOXP2. The band at 100 kDa is the overexpressed FOXP2, which is consistent with an over-expressed FOXP2 construct (Moparthi & Koch, 2020). The construct

we used is 2148bp plus an HA tag, while the construct used in the cited reference was 2220 including a flag tag. Therefore, our version runs a bit lower on the gel as expected. As for the band above EZH2 (at about 100 kDa), we see this band sometimes in western blots with our current protocol of whole cell extracts and the Cell Signaling antibody, in both mouse and human lysates. It appears to be present in 2D and 3D *Ezh2* null lines (**Fig. 2d**). It is also often unchanged in the *shEZH2* human cell lines and therefore, we believe this is a non-specific band (**Reviewer Fig. 5**).

9. The authors demonstrate the FoxP3 is more highly expressed in Ezh2-null tumor cells. If the authors were interested in determining whether FoxP3 played a role in the metastatic phenotype observed in Ezh2-KO tumor cells, have the authors KD FoxP3 in the murine tumor cell lines / spheroids and evaluated whether FoxP3 KD augmented the migration observed in Figure 5c? That would be definitive evidence of a role of FoxP3 in this process.

We did attempt to do this experiment with the siRNAs, but were unable to determine the correct timing for transfection and splitting so that the cells reliably migrated. We also purchased *shFoxp2* vectors from SIGMA, but were unable to get the puromycin selection to work well on the murine cultures. While we agree that this is an interesting experiment, given these technical difficulties we do not have the data to share with you at this time.

10. Have the authors tried other first-line chemotherapy agents besides carboplatin (e.g. cisplatin, etoposide)? Using more than one agent would be more compelling to the authors' idea of poorly differentiated cells with high FOXP2 being more chemoresistant.

Yes, we repeated this assay using etoposide and now show both carboplatin and etoposide in **Fig. S5b-c**. Both drugs had a significantly higher IC₅₀ values when FOXP2 was over-expressed.

11. Can the authors comment on the 3d organoid sphere sizes? There appears to be variability in 3D organoid sizes across the three EZH2 genotypes in Figure S6. Could the size of the 3D organoid structure have a confounding effect on JQ1 / GSK-J4 efficacy due to the possibility of the lack of drug penetrance into the centre of bigger organoid structures.

The organoids are hollow (see histology in **Fig. 2a**), so we do not believe that drug penetrance into the organoids is an issue. This idea is supported by the almost complete loss of H3K27me3 in EPZ6438 treated organoids by ChIP-seq (**Fig. S3b+d**). We did not observe any apparent differences in organoid sizes between genotypes.

12. Can the authors comment on the observed decrease in survival of 3D EZH2 +/+ organoids upon treatment with JQ1 (Figure 6C)?

JQ1 is known for its ability to target and decrease *c-Myc*. Therefore, in the 3D *Ezh2* wild-type organoids, JQ1 is likely decreasing growth through a decrease in *c-Myc* expression (**Fig. S6d**). However, the *c-Myc* was not differentially expressed in different genotypes, so we think *c-Myc* is not the reason of the differential sensitivity.

18. Statistics on GSK-J4 day 10 values of EZH2 null compared to EZH2 haplo-insufficient and EZH2 proficient genotypes needed (Figure 6D). Please specify what allograft tumour volume was reached before drug treatment commenced, and at what range the tumour volumes were.

The statistics on GSK-J4 treated *Ezh2* null allograft relative volumes compared to *Ezh2* heterozygous and *Ezh2* wild-type were not statistically significant using repeated measures ANOVA by our collaborating statistician; we added this point the figure legend and a n.s. indication on the graph. Please note that this metric was testing a difference between *Ezh2* wild-type and *Ezh2* null response to the drug rather than comparing placebo to treated. We performed this analysis to more closely mimic the 2D and 3D results, and also to be able to plot more data on one graph. The treatments started when the graft volumes reached an average of 60mm³ and the range of tumor volumes during the 10 days were 29.5 mm³-281.6 mm³. The reason why we chose JQ1 for further study is that GSK-J4 did not work well in *Ezh2* heterozygous organoids, which suggested that GSK-J4 works best in the cells with the complete loss of EZH2 function, and this state may be difficult to achieve in patients.

19. If high proliferation is the cause of decoupling EZH2 and H3K27Me3, have the authors considered using small molecular inhibitors or knock-down experiments targeting cell cycle. This would be additional compelling data alongside supplementation with SAM.

Yes, that theory makes sense. However, when we tried to induce cell cycle arrest in 2D cultures using a CDK1 inhibitor or nocodazole, we observed only an increase in H3K27me3 but not a decrease in EZH2. We chose these drugs due to their frequent use, though they may not be ideal for our purposes. Both CDK1 inhibitor and nocodazole will freeze cells in G2-M, where EZH2 levels may be kept high. We may in the future try ionizing radiation or hydroxy-urea, though both also have G2-M accumulation and DNA damage repair up-regulation and therefore, stabilize EZH2, but at this time we do not have those data to share.

20. In the patient IHC, decoupling EZH2 loss and H3K27Me3 is observed. However, in many in vitro and in vivo systems presented in this manuscript, we observe high EZH2 being associated with high H3K27Me3. This decoupling hypothesis is only explored with one human cell line H460 (Figure 7F) and less compellingly with a EZH2 WT murine line (Figure S7D). This begs the question as to whether this particular murine GEMM model is an appropriate model for decoupling.

To address this, we performed the SAM experiment in two additional human lung cancer cell lines and reproducibly observed a stabilization of H3K27me3 and a decrease in EZH2, which we quantified and found to be statistically significant (**Fig. 7f+h**). We did not intend the mouse model to be a model of EZH2-PRC2 activity de-coupling. Instead, we used the mouse model as a paradigm to study PRC2 deficiency achieved through deletion of EZH2.

Minor Comments:

1. Page 3 – spell out in full: MEK acronym.
2. Related to Figure 1b – no scale bar is included. Please add.
3. Figure 1G column 2 – missing label for EZH2 Δ /+
4. There is no panel Figure 3c and Figure 3d that relate to what is written in the Figure Legend.
5. Figure legend 5d and 5k are missing scale bar measurement in the Figure legend.
6. There are two figure legends for Figure S5c.
7. Measurement for scale bar in Figure S5c missing in figure legend.

All of these minor comments have been addressed.

References:

- Bramham, C. R., Alme, M. N., Bittins, M., Kuipers, S. D., Nair, R. R., Pai, B., . . . Wibbrand, K. (2010). The Arc of synaptic memory. *Exp Brain Res*, 200(2), 125-140. doi:10.1007/s00221-009-1959-2
- Chen, X., Song, N., Matsumoto, K., Nanashima, A., Nagayasu, T., Hayashi, T., . . . Koji, T. (2013). High expression of trimethylated histone H3 at lysine 27 predicts better prognosis in non-small cell lung cancer. *International Journal of Oncology*, 43(5), 1467-1480. doi:10.3892/ijo.2013.2062
- Holm, K., Grabau, D., Lövgren, K., Aradottir, S., Gruvberger-Saal, S., Howlin, J., . . . Ringnér, M. (2012). Global H3K27 trimethylation and EZH2 abundance in breast tumor subtypes. *Molecular Oncology*, 6(5), 494-506. doi:10.1016/j.molonc.2012.06.002
- Huang, X., Yan, J., Zhang, M., Wang, Y., Chen, Y., Fu, X., . . . Geng, M. (2018). Targeting Epigenetic Crosstalk as a Therapeutic Strategy for EZH2-Aberrant Solid Tumors. *Cell*, 175(1), 186-199.e119. doi:<https://doi.org/10.1016/j.cell.2018.08.058>
- Jackson, E. L., Olive, K. P., Tuveson, D. A., Bronson, R., Crowley, D., Brown, M., & Jacks, T. (2005). The Differential Effects of Mutant p53 Alleles on Advanced Murine Lung Cancer. *Cancer Research*, 65(22), 10280-10288. doi:10.1158/0008-5472.can-05-2193
- Kotipatruni, R. P., Ferraro, D. J., Ren, X., Vanderwaal, R. P., Thotala, D. K., Hallahan, D. E., & Jaboin, J. J. (2012). NDRG4, the N-Myc downstream regulated gene, is important for cell survival, tumor invasion and angiogenesis in meningiomas. *Integrative Biology*, 4(10), 1185-1197. doi:10.1039/c2ib20168b
- Lee, H.-O., Wang, L., Kuo, Y.-M., Gupta, S., Slifker, M. J., Li, Y.-s., . . . Kruger, W. D. (2017). Lack of global epigenetic methylation defects in CBS deficient mice. *Journal of inherited metabolic disease*, 40(1), 113-120. doi:10.1007/s10545-016-9958-5
- Moparthi, L., & Koch, S. (2020). A uniform expression library for the exploration of FOX transcription factor biology. *Differentiation*, 115, 30-36. doi:<https://doi.org/10.1016/j.diff.2020.08.002>
- Newman, A. M., Liu, C. L., Green, M. R., Gentles, A. J., Feng, W., Xu, Y., . . . Alizadeh, A. A. (2015). Robust enumeration of cell subsets from tissue expression profiles. *Nat Methods*, 12(5), 453-457. doi:10.1038/nmeth.3337
- Pellakuru, L. G., Iwata, T., Gurel, B., Schultz, D., Hicks, J., Bethel, C., . . . De Marzo, A. M. (2012). Global Levels of H3K27me3 Track with Differentiation in Vivo and Are Deregulated by MYC in Prostate Cancer. *The American Journal of Pathology*, 181(2), 560-569. doi:10.1016/j.ajpath.2012.04.021
- Schmuck, E. G., Roy, S., Dhillon, A., Walker, S., Spinali, K., Colevas, S., . . . Raval, A. N. (2021). Cultured cardiac fibroblasts and myofibroblasts express Sushi Containing Domain 2 and assemble a unique fibronectin rich matrix. *Exp Cell Res*, 399(2), 112489. doi:10.1016/j.yexcr.2021.112489
- Serresi, M., Gargiulo, G., Proost, N., Siteur, B., Cesaroni, M., Koppens, M., . . . van Lohuizen, M. (2016). Polycomb Repressive Complex 2 Is a Barrier to KRAS-Driven Inflammation and Epithelial-Mesenchymal Transition in Non-Small-Cell Lung Cancer. *Cancer Cell*, 29(1), 17-31. doi:10.1016/j.ccell.2015.12.006
- Shu, W., Lu, M. M., Zhang, Y., Tucker, P. W., Zhou, D., & Morrissey, E. E. (2007). Foxp2 and Foxp1 cooperatively regulate lung and esophagus development. *Development*, 134(10), 1991-2000. doi:10.1242/dev.02846
- Shu, W., Yang, H., Zhang, L., Lu, M. M., & Morrissey, E. E. (2001). Characterization of a New Subfamily of Winged-helix/Forkhead (Fox) Genes That Are Expressed in the Lung and Act as Transcriptional Repressors. *Journal of Biological Chemistry*, 276(29), 27488-27497.
- Sturm, G., Finotello, F., Petitprez, F., Zhang, J. D., Baumbach, J., Fridman, W. H., . . . Aneichyk, T. (2019). Comprehensive evaluation of transcriptome-based cell-type quantification methods for immunology. *Bioinformatics*, 35(14), i436-i445. doi:10.1093/bioinformatics/btz363
- Veniaminova, N. A., Vagnozzi, A. N., Kopinke, D., Do, T. T., Murtaugh, L. C., Maillard, I., . . . Wong, S. Y. (2013). Keratin 79 identifies a novel population of migratory epithelial cells that initiates hair canal morphogenesis and regeneration. *Development (Cambridge, England)*, 140(24), 4870-4880. doi:10.1242/dev.101725

- Wang, L., Jin, Q., Lee, J. E., Su, I. H., & Ge, K. (2010). Histone H3K27 methyltransferase Ezh2 represses Wnt genes to facilitate adipogenesis. *Proc Natl Acad Sci U S A*, 107(16), 7317-7322. doi:10.1073/pnas.1000031107
- Wang, Q., Chow, J., Hong, J., Smith, A. F., Moreno, C., Seaby, P., . . . Varmuza, S. (2011). Recent acquisition of imprinting at the rodent Sfbmt2 locus correlates with insertion of a large block of miRNAs. *BMC Genomics*, 12(1), 204. doi:10.1186/1471-2164-12-204
- Zacharek, S. J., Fillmore, C. M., Lau, A. N., Gludish, D. W., Chou, A., Ho, J. W. K., . . . Kim, C. F. (2010). Lung Stem Cell Self-Renewal Relies on BMI1-Dependent Control of Expression at Imprinted Loci. *Cell Stem Cell*, 9(3), 272-281. doi:10.1016/j.stem.2011.07.007
- Zhang, H., Fillmore Brainson, C., Koyama, S., Redig, A. J., Chen, T., Li, S., . . . Wong, K.-K. (2017). Lkb1 inactivation drives lung cancer lineage switching governed by Polycomb Repressive Complex 2. *Nature Communications*, 8, 14922-14922. doi:10.1038/ncomms14922
- Zhang, Y., Dong, W., Zhu, J., Wang, L., Wu, X., & Shan, H. (2017). Combination of EZH2 inhibitor and BET inhibitor for treatment of diffuse intrinsic pontine glioma. *Cell Biosci*, 7(1), 56. doi:10.1186/s13578-017-0184-0

REVIEWER COMMENTS

Reviewer #1 (Remarks to the Author):

This reviewer appreciates the efforts by the authors in addressing the comments raised in the initial review. The manuscript has been substantially improved by the additional experiments and clarifications. However, several concerns related to the conceptual disconnections and scientific rigor remain to be addressed.

1. The manuscript remains to be disconnected between the mouse model/mechanistic studies and correlative studies in human patient samples. Chiefly among them, EZH2 loss increases FOXP2 in mouse models and EZH2 positively correlates with FOXP2 in human specimens. The authors may consider to only focus those tumors with high EZH2 and low H3K27me3 and examine whether there is a negative correlation between EZH2 and FOXP2.
2. Scientific rigor remains to be a major concern because some of the data simply do not support the conclusion. For example, in Figure S5, MYCN expression in S5e and shFOXP2.1 does not reduce its expression in H2009 cells, but suppressed cell growth, indicating off-target effects. In addition, several results are either marginal or do not support the conclusion (for example, Reviewer Fig. 2).
3. Please simplify the description and text. It is very dense and hard to follow. In addition, please carefully check the Figure citations in the text as quite a few of them do not match the description in the text.

Reviewer #2 (Remarks to the Author):

In the revised version of the manuscript, the authors have addressed most of the major points mentioned during the first review cycle and have markedly improved the study, including new experimental data, new analysis, and more careful and less broad conclusions. In addition, the authors have improved transparency and statistical significance. The work as revised warrants publication and should be of interest to the readership of Nature Communications.

Reviewer #3 (Remarks to the Author):

I would like to thank the authors for taking on-board the comments I raised. They have undertaken a significant amount of additional experiments and re-analysis of data to fully address my concerns. I therefore believe that this manuscript is now fit for publication in Nature Communications.

Rebuttal for *Nature Communications* manuscript “Polycomb Deficiency Drives a FOXP2-high Aggressive State Targetable by Epigenetic Inhibitors” by Chen et al.

We thank the reviewers for their comprehensive review of our manuscript. We have taken significant time and effort to address the concerns raised. Below are each of the reviewer’s comments followed by our responses in blue.

Reviewers' comments:

Reviewer #1 (Remarks to the Author):

This reviewer appreciates the efforts by the authors in addressing the comments raised in the initial review. The manuscript has been substantially improved by the additional experiments and clarifications. However, several concerns related to the conceptual disconnections and scientific rigor remain to be addressed.

1. The manuscript remains to be disconnected between the mouse model/mechanistic studies and correlative studies in human patient samples. Chiefly among them, EZH2 loss increases FOXP2 in mouse models and EZH2 positively correlates with FOXP2 in human specimens. The authors may consider to only focus those tumors with high EZH2 and low H3K27me3 and examine whether there is a negative correlation between EZH2 and FOXP2.

As the reviewer suggested, we focused on tumors with high EZH2 (>10% of cells) and low H3K27me3 (<40% of cells). In these tumors, there remains a positive correlation between EZH2 and FOXP2 (**Reviewer Figure 1a**).

However, this result is expected based on our data and model. In patient samples, EZH2 expression is likely a reflection of increased proliferation, given that EZH2 is a direct target of E2F transcription factors, highly co-expressed with cell cycle and mitosis genes in lung cancer, and correlates with poor prognosis (Bracken et al., 2003; Fillmore et al., 2015) (**Figure S7c+d**). Furthermore, several papers, including our own, have shown an *inverse* correlation between EZH2 and its catalytic mark, H3K27me3 *in vivo* (Bae et al., 2015; Chen et al., 2013; Holm et al., 2012; Onishi, Takashima, Kurashige, Ohshima, & Morii, 2022; Pellakuru et al., 2012; Zhang et al., 2017; Zhu et al., 2018). This abundant EZH2 may be part of PRC2-independent complexes (J. Kim et al., 2018; K. H. Kim et al., 2015; Xu et al., 2012), or unable to complete histone methylation for a variety of reasons (one reason postulated is lack of SAM). If one follows this logic, then high EZH2 may be a marker of lower PRC2 function, and our mouse model and *in vitro* data clearly show that one result of lowering PRC2 function in KRAS-driven adenocarcinomas is to de-repress *FOXP2*. In this way, the mouse model, with EZH2 deletion to drive PRC2 dysfunction, and the human samples, with high EZH2 indicating PRC2 dysfunction, do match. We now clearly state this interpretation in the discussion.

To link this finding to drug sensitivity, we now show that when *Ezh2* null cells are reconstituted with WT EZH2, their sensitivity to JQ1 is *much lower* (2.4-fold higher IC₅₀) than when methyltransferase dead EZH2 is present (**Figure S6j**). These data support a model in which EZH2 can be present at high levels in tumor cells, but when it is not part of a functional PRC2 complex, then FOXP2 levels and sensitivity to JQ1 can be high. In our experimental models, this ‘de-coupling’ was achieved through EPZ6438 treatment and reconstitution with methyltransferase-dead EZH2. Furthermore, we hypothesize that *in vivo*, low SAM availability may be a reason for high EZH2 and low H3K27me3. To support this theory, in mouse and human lines *in vitro*, we observed that H3K27me3 decoupled from EZH2 and FOXP2 expression after 500µM SAM treatment for 6 days (**Figure 7f-h, S7g+h**). This suggests that a mouse model with metabolic rewiring/dysregulation may be

suitable to study the mechanism of EZH2/H3K27me3 decoupling in human patient samples, which needs further research in the future.

What we might expect, and were unable to detect with our methods, was an inverse correlation between FOXP2 and H3K27me3. The reasons for this could include: 1) tissue heterogeneity in terms of cell-of-origin, 2) lack of abundant KRAS-driven samples (we do not have genotyping for this TMA), or 3) de-repression of *FOXP2* at a locus level without a global decrease in H3K27me3. We observed this phenomenon in some of the *Ezh2* heterozygous tumors and tumoroids – they retained high levels of H3K27me3, but also had FOXP2 expression. We have recently submitted an R01 proposal that will further examine the interactions between FOXP2 and the PRC2-dependent chromatin landscape in lung cells and this future work will help us to better understand these complexities.

2. Scientific rigor remains to be a major concern because some of the data simply do not support the conclusion. For example, in Figure S5, MYCN expression in S5e and shFOXP2.1 does not reduce its expression in H2009 cells, but suppressed cell growth, indicating off-target effects. In addition, several results are either marginal or do not support the conclusion (for example, Reviewer Fig. 2).

We had previously re-worded our conclusions surrounding the hypothesis that FOXP2 drives JQ1 sensitivity (previous reviewer Figure 2). To further test this point, we performed JQ1 dose response assays using H2030 and H460 cells over-expressing FOXP2 in 3D tumoroids instead of 2D cell lines. We found that H460 cells

Reviewer Figure 1:

A) Correlation of EZH2 and FOXP2 in lung adenocarcinomas and poorly differentiated tumors with High EZH2 and Low H3K27me3 (n=29). **B)** JQ1 dose response on H2030 FOXP2 over-expressing and control tumoroids, n=4. **C)** RT-qPCR in H2009 cultures showed a significant decrease in *FOXP2* gene expression with *shFOXP2.1*. Due to knock-down variability, the result with *shFOXP2.2* was not significant (p=0.069), ** indicates p=0.0004, n=5.

have a *dramatically increased* sensitivity to JQ1 (2.4-fold lower IC₅₀ p<0.0001) when grown in 3D cultures (**Figure S6l**), but H2030 cells did not show this sensitivity (**Reviewer Figure 1b**). These results suggest that in addition to FOXP2 expression, the epigenetic state of the cells, or the other genetic mutations that they harbor in addition to KRAS activation,

determine sensitivity to JQ1. Given that FOXP2 is a transcription factor, this result is somewhat expected – a differing chromatin state, governed in part through PRC2 activity, will determine where in the genome FOXP2 is able to bind, and could determine downstream gene repression/activation and drug responses. We have again carefully worded the conclusions around this point, stating that FOXP2 *can* induce JQ1 sensitivity in certain conditions, but not implying that those conditions are always met.

Similarly, in **Figure S5e**, H2030 cells appear to have a slightly additive effect of over-expression of FOPX2 and treatment with EPZ6438 for *MYCN* and *NDRG4* expression, while in **Figure S5d**, H460 cells show that *NDRG4* is increased with EPZ6438 treatment and slightly down-regulated from that high level when FOXP2 is expressed and EPZ6438 is used. There are numerous possible explanations for this outcome, including the presence of a negative feedback loop – previous literature has shown that over-expression of NDRG1 decreases c-MYC (Xi et al., 2017), and a similar phenomenon could be in place with NDRG4 and n-MYC. For clarity, we now add the ANOVA statistics for the comparison of FOXP2 to FOXP2+EPZ6438. For H460 cells,

both genes are increased when FOXP2 is over-expressed and EPZ6438 is used, suggesting that FOXP2 requires a PRC2-depleted epigenetic state to exert up-regulation. Our conclusion here was reworded to state that combination of FOXP2 and EPZ6438 (veh vs combo) can induce genes that we observed in *Ezh2* null tumoroids. Given that these data are peripheral to other major findings, we do not speculate in the text exactly why *MYCN* does not follow a stepwise increase pattern.

For the FOXP2 knock-down experiment, we now show a new blot for H2009 cells from different lysates, and the RT-qPCR results for 5 passages (**Figure S5f, Reviewer Figure 1c**). For all of the *FOXP2* knock-down quantifications, the results represent an average of 4 western blots per cell line (**Figure S5g**). Off-target effects of short-hairpins can never be fully ruled out, but given that we observed similar growth defects with 2 different small hairpins in 3 different NSCLC cell lines, we believe it highly unlikely that the effect is due to any other perturbation than that of *FOXP2*.

3. Please simplify the description and text. It is very dense and hard to follow. In addition, please carefully check the Figure citations in the text as quite a few of them do not match the description in the text.

We verified all the figure citations, and made efforts to streamline and simplify the text together with our research communications office. Given the large number of edits we made, we do not provide tracked changes. We hope that you agree that these changes improve the reading experience.

Reviewer #2 (Remarks to the Author):

In the revised version of the manuscript, the authors have addressed most of the major points mentioned during the first review cycle and have markedly improved the study, including new experimental data, new analysis, and more careful and less broad conclusions. In addition, the authors have improved transparency and statistical significance. The work as revised warrants publication and should be of interest to the readership of Nature Communications.

We thank you very much for your comprehensive review and approval of our work.

Reviewer #3 (Remarks to the Author):

I would like to thank the authors for taking on-board the comments I raised. They have undertaken a significant amount of additional experiments and re-analysis of data to fully address my concerns. I therefore believe that this manuscript is now fit for publication in Nature Communications.

We thank you very much for your comprehensive review and approval of our work.

Bae, W. K., Yoo, K. H., Lee, J. S., Kim, Y., Chung, I.-J., Park, M. H., . . . Hennighausen, L. (2015). The methyltransferase EZH2 is not required for mammary cancer development, although high EZH2 and low H3K27me3 correlate with poor prognosis of ER-positive breast cancers. *Molecular carcinogenesis*, 54(10), 1172-1180. doi:10.1002/mc.22188

Bracken, A. P., Pasini, D., Capra, M., Prosperini, E., Colli, E., & Helin, K. (2003). EZH2 is downstream of the pRB-E2F pathway, essential for proliferation and amplified in cancer. *Embo j*, 22(20), 5323-5335. doi:10.1093/emboj/cdg542

Chen, X., Song, N., Matsumoto, K., Nanashima, A., Nagayasu, T., Hayashi, T., . . . Koji, T. (2013). High expression of trimethylated histone H3 at lysine 27 predicts better prognosis in non-small cell lung cancer. *International Journal of Oncology*, 43(5), 1467-1480. doi:10.3892/ijo.2013.2062

- Fillmore, C. M., Xu, C., Desai, P. T., Berry, J. M., Rowbotham, S. P., Lin, Y.-J., . . . Kim, C. F. (2015). EZH2 inhibition sensitizes BRG1 and EGFR mutant lung tumours to TopoII inhibitors. *Nature*, *520*(7546), 239-242. doi:10.1038/nature14122
- Holm, K., Grabau, D., Lövgren, K., Aradottir, S., Gruvberger-Saal, S., Howlin, J., . . . Ringnér, M. (2012). Global H3K27 trimethylation and EZH2 abundance in breast tumor subtypes. *Molecular Oncology*, *6*(5), 494-506. doi:10.1016/j.molonc.2012.06.002
- Kim, J., Lee, Y., Lu, X., Song, B., Fong, K. W., Cao, Q., . . . Yu, J. (2018). Polycomb- and Methylation-Independent Roles of EZH2 as a Transcription Activator. *Cell Rep*, *25*(10), 2808-2820.e2804. doi:10.1016/j.celrep.2018.11.035
- Kim, K. H., Kim, W., Howard, T. P., Vazquez, F., Tsherniak, A., Wu, J. N., . . . Roberts, C. W. (2015). SWI/SNF-mutant cancers depend on catalytic and non-catalytic activity of EZH2. *Nat Med*, *21*(12), 1491-1496. doi:10.1038/nm.3968
- Onishi, T., Takashima, T., Kurashige, M., Ohshima, K., & Morii, E. (2022). Mutually exclusive expression of EZH2 and H3K27me3 in non-small cell lung carcinoma. *Pathol Res Pract*, *238*, 154071. doi:10.1016/j.prp.2022.154071
- Pellakuru, L. G., Iwata, T., Gurel, B., Schultz, D., Hicks, J., Bethel, C., . . . De Marzo, A. M. (2012). Global Levels of H3K27me3 Track with Differentiation in Vivo and Are Deregulated by MYC in Prostate Cancer. *The American Journal of Pathology*, *181*(2), 560-569. doi:10.1016/j.ajpath.2012.04.021
- Xi, R., Pun, I. H. Y., Menezes, S. V., Fouani, L., Kalinowski, D. S., Huang, M. L. H., . . . Kovacevic, Z. (2017). Novel Thiosemicarbazones Inhibit Lysine-Rich Carcinoembryonic Antigen-Related Cell Adhesion Molecule 1 (CEACAM1) Coisolated (LYRIC) and the LYRIC-Induced Epithelial-Mesenchymal Transition via Upregulation of N-Myc Downstream-Regulated Gene 1 (NDRG1). *Molecular Pharmacology*, *91*(5), 499-517. doi:10.1124/mol.116.107870
- Xu, K., Wu, Z. J., Groner, A. C., He, H. H., Cai, C., Lis, R. T., . . . Brown, M. (2012). EZH2 Oncogenic Activity in Castration-Resistant Prostate Cancer Cells Is Polycomb-Independent. *Science*, *338*(6113), 1465-1469. doi:10.1126/science.1227604
- Zhang, H., Fillmore Brainson, C., Koyama, S., Redig, A. J., Chen, T., Li, S., . . . Wong, K.-K. (2017). Lkb1 inactivation drives lung cancer lineage switching governed by Polycomb Repressive Complex 2. *Nature Communications*, *8*, 14922-14922. doi:10.1038/ncomms14922
- Zhu, K., Deng, Y., Weng, G., Hu, D., Huang, C., Matsumoto, K., . . . Chen, X. (2018). Analysis of H3K27me3 expression and DNA methylation at CCGG sites in smoking and non-smoking patients with non-small cell lung cancer and their clinical significance. *Oncology Letters*, *15*(5), 6179-6188. doi:10.3892/ol.2018.8100

REVIEWERS' COMMENTS

Reviewer #1 (Remarks to the Author):

The authors answered my remaining concerns with clarifications and simplified the manuscript as suggested. The manuscript is now acceptable for publication.